# Simplifying, Stabilizing & Scaling Continuous-Time Consistency Models

**Cheng Lu & Yang Song**
OpenAI

## Abstract

Consistency models (CMs) are a powerful class of diffusion-based generative models optimized for fast sampling. Most existing CMs are trained using discretized timesteps, which introduce additional hyperparameters and are prone to discretization errors. While continuous-time formulations can mitigate these issues, their success has been limited by training instability. To address this, we propose a simplified theoretical framework that unifies previous parameterizations of diffusion models and CMs, identifying the root causes of instability. Based on this analysis, we introduce key improvements in diffusion process parameterization, network architecture, and training objectives. These changes enable us to train continuous-time CMs at an unprecedented scale, reaching 1.5B parameters on ImageNet 512×512. Our proposed training algorithm, using only two sampling steps, achieves FID scores of 2.06 on CIFAR-10, 1.48 on ImageNet 64×64, and 1.88 on ImageNet 512×512, narrowing the gap in FID scores with the best existing diffusion models to within 10%.

## 1 Introduction

Diffusion models (Sohl-Dickstein et al., 2015; Song & Ermon, 2019; Ho et al., 2020; Song et al., 2021b) have revolutionized generative AI, achieving remarkable results in image (Rombach et al., 2022; Ramesh et al., 2022; Ho et al., 2022), 3D (Poole et al., 2022; Wang et al., 2024; Liu et al., 2023b), audio (Liu et al., 2023a; Evans et al., 2024), and video generation (Blattmann et al., 2023; Brooks et al., 2024). Despite their success, a significant drawback is their slow sampling speed, often requiring dozens to hundreds of steps to generate a single sample. Various diffusion distillation techniques have been proposed, including direct distillation (Luhman & Luhman, 2021; Zheng et al., 2023b), adversarial distillation (Wang et al., 2022; Sauer et al., 2023), progressive distillation (Salimans & Ho, 2022), and variational score distillation (VSD) (Wang et al., 2024; Yin et al., 2024b;a; Luo et al., 2024; Xie et al., 2024b; Salimans et al., 2024). However, these methods come with challenges: direct distillation incurs extensive computational cost due to the need for numerous diffusion model samples; adversarial distillation introduces complexities associated with GAN training; progressive

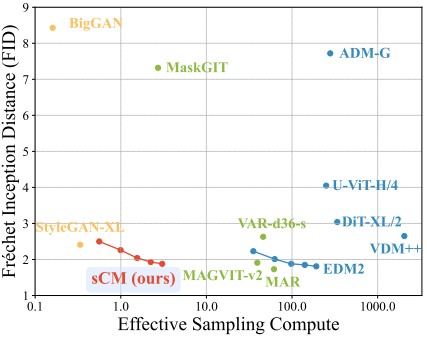

Figure 1: Sample quality vs. effective sampling compute (billion parameters × number of function evaluations during sampling). We compare the sample quality of different models on ImageNet 512×512, measured by FID (↓). Our 2-step sCM achieves sample quality comparable to the best previous generative models while using less than 10% of the effective sampling compute.

distillation requires multiple training stages and is less effective for one or two-step generation; and VSD can produce overly smooth samples with limited diversity and struggles at high guidance levels.

Consistency models (CMs) (Song et al., 2023; Song & Dhariwal, 2023) offer significant advantages in addressing these issues. They eliminate the need for supervision from diffusion model samples, avoiding the computational cost of generating synthetic datasets. CMs also bypass adversarial training, sidestepping its inherent difficulties. Aside from distillation, CMs can be trained from scratch with consistency training (CT), without relying on pre-trained diffusion models. Previous work (Song & Dhariwal, 2023; Geng et al., 2024; Luo et al., 2023; Xie et al., 2024a) has demonstrated the

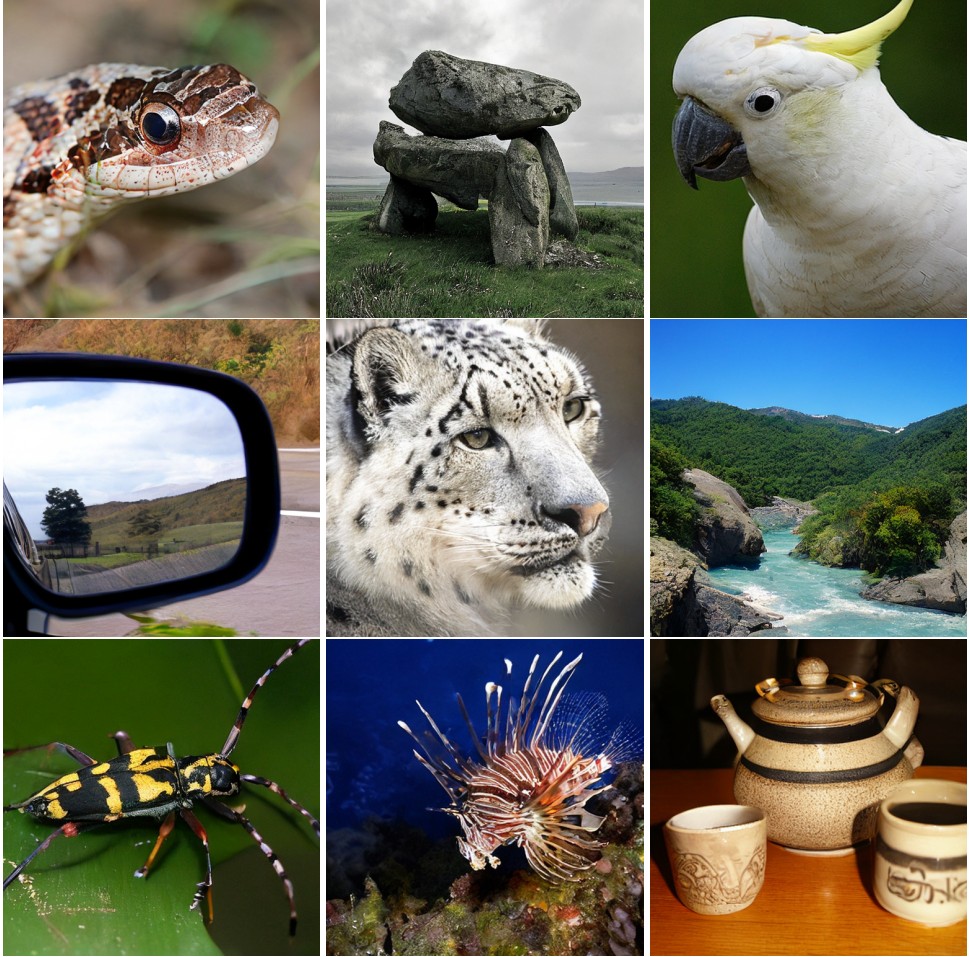

Figure 2: Selected 2-step samples from a continuous-time consistency model trained on ImageNet $512\times512$.

effectiveness of CMs in few-step generation, especially in one or two steps. However, these results are all based on discrete-time CMs, which introduces discretization errors and requires careful scheduling of the timestep grid, potentially leading to suboptimal sample quality. In contrast, continuous-time CMs avoid these issues but have faced challenges with training instability (Song et al., 2023; Song & Dhariwal, 2023; Geng et al., 2024).

In this work, we introduce techniques to simplify, stabilize, and scale up the training of continuous-time CMs. Our first contribution is TrigFlow, a new formulation that unifies EDM (Karras et al., 2022; 2024) and Flow Matching (Peluchetti, 2022; Lipman et al., 2022; Liu et al., 2022; Albergo et al., 2023; Heitz et al., 2023), significantly simplifying the formulation of diffusion models, the associated probability flow ODE and CMs. Building on this foundation, we analyze the root causes of instability in CM training and propose a complete recipe for mitigation. Our approach includes improved time-conditioning and adaptive group normalization within the network architecture. Additionally, we re-formulate the training objective for continuous-time CMs, incorporating adaptive weighting and normalization of key terms, and progressive annealing for stable and scalable training.

With these improvements, we elevate the performance of consistency models in both consistency training and distillation, achieving comparable or better results compared to previous discrete-time formulations. Our models, referred to as sCMs, demonstrate success across various datasets and model sizes. We train sCMs on CIFAR-10, ImageNet $64\times64$, and ImageNet $512\times512$, reaching an unprecedented scale with 1.5 billion parameters—the largest CMs trained to date (samples in Figure 2). We show that sCMs scale effectively with increased compute, achieving better sample quality in a predictable way. Moreover, when measured against state-of-the-art diffusion models, which require significantly more sampling compute, sCMs narrow the FID gap to within 10% using two-step generation. In addition, we provide a rigorous justification for the advantages of continuous-

time CMs over discrete-time variants by demonstrating that sample quality improves as the gap between adjacent timesteps narrows to approach the continuous-time limit. Furthermore, we examine the differences between sCMs and VSD, finding that sCMs produce more diverse samples and are more compatible with guidance, whereas VSD tends to struggle at higher guidance levels.

## 2 PRELIMINARIES

### 2.1 DIFFUSION MODELS

Given a training dataset, let $p_d$ denote its underlying data distribution and $\sigma_d$ its standard deviation. Diffusion models generate samples by learning to reverse a noising process that progressively perturbs a data sample $\boldsymbol{x}_0 \sim p_d$ into a noisy version $\boldsymbol{x}_t = \alpha_t \boldsymbol{x}_0 + \sigma_t \boldsymbol{z}$, where $\boldsymbol{z} \sim \mathcal{N}(\boldsymbol{0}, \boldsymbol{I})$ is standard Gaussian noise. This perturbation increases with $t \in [0, T]$, where larger $t$ indicates greater noise.

We consider two recent formulations for diffusion models.

**EDM** (Karras et al., 2022; 2024). The noising process simply sets $\alpha_t = 1$ and $\sigma_t = t$, with the training objective given by $\mathbb{E}_{\boldsymbol{x}_0, \boldsymbol{z}, t} \left[ w(t) \left\| \boldsymbol{f}_\theta^{\text{DM}}(\boldsymbol{x}_t, t) - \boldsymbol{x}_0 \right\|_2^2 \right]$, where $w(t)$ is a weighting function. The diffusion model is parameterized as $\boldsymbol{f}_\theta^{\text{DM}}(\boldsymbol{x}_t, t) = c_{\text{skip}}(t)\boldsymbol{x}_t + c_{\text{out}}(t)\boldsymbol{F}_\theta(c_{\text{in}}(t)\boldsymbol{x}_t, c_{\text{noise}}(t))$, where $\boldsymbol{F}_\theta$ is a neural network with parameters $\theta$, and $c_{\text{skip}}$, $c_{\text{out}}$, $c_{\text{in}}$, and $c_{\text{noise}}$ are manually designed coefficients that ensure the training objective has the unit variance across timesteps at initialization. For sampling, EDM solves the *probability flow ODE (PF-ODE)* (Song et al., 2021b), defined by $\frac{\mathrm{d}\boldsymbol{x}_t}{\mathrm{d}t} = [\boldsymbol{x}_t - \boldsymbol{f}_\theta^{\text{DM}}(\boldsymbol{x}_t, t)]/t$, starting from $\boldsymbol{x}_T \sim \mathcal{N}(\boldsymbol{0}, T^2\boldsymbol{I})$ and stopping at $\boldsymbol{x}_0$.

**Flow Matching**. The noising process uses differentiable coefficients $\alpha_t$ and $\sigma_t$, with time derivatives denoted by $\alpha_t'$ and $\sigma_t'$ (typically, $\alpha_t = 1 - t$ and $\sigma_t = t$). The training objective is given by $\mathbb{E}_{\boldsymbol{x}_0, \boldsymbol{z}, t} \left[ w(t) \left\| \boldsymbol{F}_\theta(\boldsymbol{x}_t, t) - (\alpha_t' \boldsymbol{x}_0 + \sigma_t' \boldsymbol{z}) \right\|_2^2 \right]$, where $w(t)$ is a weighting function and $\boldsymbol{F}_\theta$ is a neural network parameterized by $\theta$. The sampling procedure begins at $t = 1$ with $\boldsymbol{x}_1 \sim \mathcal{N}(\boldsymbol{0}, \boldsymbol{I})$ and solves the probability flow ODE (PF-ODE), defined by $\frac{\mathrm{d}\boldsymbol{x}_t}{\mathrm{d}t} = \boldsymbol{F}_\theta(\boldsymbol{x}_t, t)$, from $t = 1$ to $t = 0$.

### 2.2 CONSISTENCY MODELS

A consistency model (CM) (Song et al., 2023; Song & Dhariwal, 2023) is a neural network $\boldsymbol{f}_\theta(\boldsymbol{x}_t, t)$ trained to map the noisy input $\boldsymbol{x}_t$ directly to the corresponding clean data $\boldsymbol{x}_0$ in one step, by following the sampling trajectory of the PF-ODE starting at $\boldsymbol{x}_t$. A valid $\boldsymbol{f}_\theta$ must satisfy the *boundary condition*, $\boldsymbol{f}_\theta(\boldsymbol{x}, 0) \equiv \boldsymbol{x}$. One way to meet this condition is to parameterize the consistency model as $\boldsymbol{f}_\theta(\boldsymbol{x}_t, t) = c_{\text{skip}}(t)\boldsymbol{x}_t + c_{\text{out}}(t)\boldsymbol{F}_\theta(c_{\text{in}}(t)\boldsymbol{x}_t, c_{\text{noise}}(t))$ with $c_{\text{skip}}(0) = 1$ and $c_{\text{out}}(0) = 0$. CMs are trained to have consistent outputs at adjacent time steps. Depending on how nearby time steps are selected, there are two categories of consistency models, as described below.

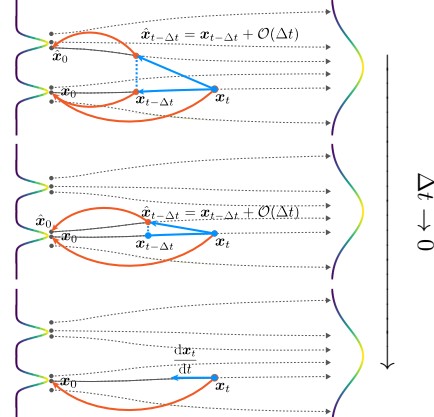

**Discrete-time CMs.** The training objective is defined at two adjacent time steps with finite distance:

$$\mathbb{E}_{\boldsymbol{x}_t, t} \left[ w(t) d(\boldsymbol{f}_\theta(\boldsymbol{x}_t, t), \boldsymbol{f}_{\theta^-}(\boldsymbol{x}_{t-\Delta t}, t - \Delta t)) \right], \quad (1)$$

where $\theta^-$ denotes stopgrad($\theta$), $w(t)$ is the weighting function, $\Delta t > 0$ is the distance between adjacent time steps, and $d(\cdot, \cdot)$ is a metric function; common choices are $\ell_2$ loss $d(\boldsymbol{x}, \boldsymbol{y}) = \|\boldsymbol{x} - \boldsymbol{y}\|_2^2$, Pseudo-Huber loss $d(\boldsymbol{x}, \boldsymbol{y}) = \sqrt{\|\boldsymbol{x} - \boldsymbol{y}\|_2^2 + c^2} - c$ for $c > 0$ (Song & Dhariwal, 2023), and LPIPS loss (Zhang et al., 2018). Discrete-time CMs are sensitive to the choice of $\Delta t$, and therefore require manually designed annealing schedules (Song & Dhariwal, 2023; Geng et al., 2024) for fast convergence. The noisy sample $\boldsymbol{x}_{t-\Delta t}$ at the preceding time step $t - \Delta t$ is often obtained from $\boldsymbol{x}_t$

Figure 3: Discrete-time CMs (top & middle) vs. continuous-time CMs (bottom). Discrete-time CMs suffer from discretization errors from numerical ODE solvers, causing imprecise predictions during training. In contrast, continuous-time CMs stay on the ODE trajectory by following its tangent direction with infinitesimal steps.

by solving the PF-ODE with numerical ODE solvers using step size $\Delta t$, which can cause additional discretization errors.

**Continuous-time CMs.** When using $d(\boldsymbol{x}, \boldsymbol{y}) = \|\boldsymbol{x} - \boldsymbol{y}\|_2^2$ and taking the limit $\Delta t \to 0$, Song et al. (2023, *Remark 10*) show that the gradient of Eq. (1) with respect to $\theta$ converges to

$$\nabla_\theta \mathbb{E}_{\boldsymbol{x}_t,t} \left[ w(t) \boldsymbol{f}_\theta^\top (\boldsymbol{x}_t, t) \frac{\mathrm{d}\boldsymbol{f}_{\theta^-}(\boldsymbol{x}_t, t)}{\mathrm{d}t} \right], \tag{2}$$

where $\frac{\mathrm{d}\boldsymbol{f}_{\theta^-}(\boldsymbol{x}_t,t)}{\mathrm{d}t} = \nabla_{\boldsymbol{x}_t} \boldsymbol{f}_{\theta^-}(\boldsymbol{x}_t, t)\frac{\mathrm{d}\boldsymbol{x}_t}{\mathrm{d}t} + \partial_t \boldsymbol{f}_{\theta^-}(\boldsymbol{x}_t, t)$ is the *tangent* of $\boldsymbol{f}_{\theta^-}$ at $(\boldsymbol{x}_t, t)$ along the trajectory of the PF-ODE $\frac{\mathrm{d}\boldsymbol{x}_t}{\mathrm{d}t}$. Notably, continuous-time CMs do not rely on ODE solvers, which avoids discretization errors and offers more accurate supervision signals during training. However, previous work (Song et al., 2023; Geng et al., 2024) found that training continuous-time CMs, or even discrete-time CMs with an extremely small $\Delta t$, suffers from severe instability in optimization. This greatly limits the empirical performance and adoption of continuous-time CMs.

**Consistency Distillation and Consistency Training.** Both discrete-time and continuous-time CMs can be trained using either *consistency distillation (CD)* or *consistency training (CT)*. In consistency distillation, a CM is trained by distilling knowledge from a pretrained diffusion model. This diffusion model provides the PF-ODE, which can be directly plugged into Eq. (2) for training continuous-time CMs. Furthermore, by numerically solving the PF-ODE to obtain $\boldsymbol{x}_{t-\Delta t}$ from $\boldsymbol{x}_t$, one can also train discrete-time CMs via Eq. (1). Consistency training (CT), by contrast, trains CMs from scratch without the need for pretrained diffusion models, which establishes CMs as a standalone family of generative models in their own right. Specifically, CT approximates $\boldsymbol{x}_{t-\Delta t}$ in discrete-time CMs as $\boldsymbol{x}_{t-\Delta t} = \alpha_{t-\Delta t}\boldsymbol{x}_0 + \sigma_{t-\Delta t}\boldsymbol{z}$, reusing the same data $\boldsymbol{x}_0$ and noise $\boldsymbol{z}$ when sampling $\boldsymbol{x}_t = \alpha_t\boldsymbol{x}_0 + \sigma_t\boldsymbol{z}$. In the continuous-time limit, as $\Delta t \to 0$, this approach yields an unbiased estimate of the PF-ODE $\frac{\mathrm{d}\boldsymbol{x}_t}{\mathrm{d}t} \to \alpha_t'\boldsymbol{x}_0 + \sigma_t'\boldsymbol{z}$, leading to an unbiased estimate of Eq. (2) for training continuous-time CMs.

## 3 SIMPLIFYING CONTINUOUS-TIME CONSISTENCY MODELS

Previous consistency models (CMs) adopt the model parameterization and diffusion process formulation in EDM (Karras et al., 2022). Specifically, the CM is parameterized as $\boldsymbol{f}_\theta(\boldsymbol{x}_t, t) = c_{\text{skip}}(t)\boldsymbol{x}_t + c_{\text{out}}(t)\boldsymbol{F}_\theta(c_{\text{in}}(t)\boldsymbol{x}_t, c_{\text{noise}}(t))$, where $\boldsymbol{F}_\theta$ is a neural network with parameters $\theta$. The coefficients $c_{\text{skip}}(t)$, $c_{\text{out}}(t)$, $c_{\text{in}}(t)$ are fixed to ensure that the variance of the diffusion objective is equalized across all time steps at initialization, and $c_{\text{noise}}(t)$ is a transformation of $t$ for better time conditioning. Since EDM diffusion process is variance-exploding (Song et al., 2021b), meaning that $\boldsymbol{x}_t = \boldsymbol{x}_0 + t\boldsymbol{z}$, we can derive that $c_{\text{skip}}(t) = \sigma_d^2/(t^2 + \sigma_d^2)$, $c_{\text{out}}(t) = \sigma_d \cdot t/\sqrt{\sigma_d^2 + t^2}$, and $c_{\text{in}}(t) = 1/\sqrt{t^2 + \sigma_d^2}$ (see Appendix B.6 in Karras et al. (2022)). Although these coefficients are important for training efficiency, their complex arithmetic relationships with $t$ and $\sigma_d$ complicate theoretical analyses of CMs.

To simplify EDM and subsequently CMs, we propose *TrigFlow*, a formulation of diffusion models that keep the EDM properties but satisfy $c_{\text{skip}}(t) = \cos(t)$, $c_{\text{out}}(t) = -\sigma_d \sin(t)$, and $c_{\text{in}}(t) \equiv 1/\sigma_d$ (proof in Appendix B). TrigFlow is a special case of flow matching (also known as stochastic interpolants or rectified flows) and v-prediction parameterization (Salimans & Ho, 2022). It closely resembles the trigonometric interpolant proposed by Albergo & Vanden-Eijnden (2023); Albergo et al. (2023); Ma et al. (2024), but is modified to account for $\sigma_d$, the standard deviation of the data distribution $p_d$. Since TrigFlow is a special case of flow matching and simultaneously satisfies EDM principles, it combines the advantages of both formulations while allowing the diffusion process, diffusion model parameterization, the PF-ODE, the diffusion training objective, and the CM parameterization to all have simple expressions, as provided below.

**Diffusion Process.** Given $\boldsymbol{x}_0 \sim p_d(\boldsymbol{x}_0)$ and $\boldsymbol{z} \sim \mathcal{N}(\boldsymbol{0}, \sigma_d^2 \boldsymbol{I})$, the noisy sample is defined as $\boldsymbol{x}_t = \cos(t)\boldsymbol{x}_0 + \sin(t)\boldsymbol{z}$ for $t \in [0, \frac{\pi}{2}]$. As a special case, the prior sample $\boldsymbol{x}_{\frac{\pi}{2}} \sim \mathcal{N}(\boldsymbol{0}, \sigma_d^2 \boldsymbol{I})$.

**Diffusion Models and PF-ODE.** We parameterize the diffusion model as $\boldsymbol{f}_\theta^{\text{DM}}(\boldsymbol{x}_t, t) = \boldsymbol{F}_\theta(\boldsymbol{x}_t/\sigma_d, c_{\text{noise}}(t))$, where $\boldsymbol{F}_\theta$ is a neural network with parameters $\theta$, and $c_{\text{noise}}(t)$ is a transformation of $t$ to facilitate time conditioning. The corresponding PF-ODE is given by

$$\frac{\mathrm{d}\boldsymbol{x}_t}{\mathrm{d}t} = \sigma_d \boldsymbol{F}_\theta \left( \frac{\boldsymbol{x}_t}{\sigma_d}, c_{\text{noise}}(t) \right). \tag{3}$$

**Diffusion Objective.** In TrigFlow, the diffusion model is trained by minimizing

$$\mathcal{L}_{\text{Diff}}(\theta) = \mathbb{E}_{\boldsymbol{x}_0, \boldsymbol{z}, t} \left[ \left\| \sigma_d \boldsymbol{F}_\theta \left( \frac{\boldsymbol{x}_t}{\sigma_d}, c_{\text{noise}}(t) \right) - \boldsymbol{v}_t \right\|_2^2 \right], \tag{4}$$

where $\boldsymbol{v}_t = \cos(t)\boldsymbol{z} - \sin(t)\boldsymbol{x}_0$ is the training target.

**Consistency Models.** As mentioned in Sec. 2.2, a valid CM must satisfy the boundary condition $\boldsymbol{f}_\theta(\boldsymbol{x}, 0) \equiv \boldsymbol{x}$. To enforce this condition, we parameterize the CM as the single-step solution of the PF-ODE in Eq. (3) using the first-order ODE solver (see Appendix B.1 for derivations). Specifically, CMs in TrigFlow take the form of

$$\boldsymbol{f}_\theta(\boldsymbol{x}_t, t) = \cos(t)\boldsymbol{x}_t - \sin(t)\sigma_d \boldsymbol{F}_\theta \left( \frac{\boldsymbol{x}_t}{\sigma_d}, c_{\text{noise}}(t) \right), \tag{5}$$

where $c_{\text{noise}}(t)$ is a time transformation for which we defer the discussion to Sec. 4.1.

# 4 STABILIZING CONTINUOUS-TIME CONSISTENCY MODELS

Training continuous-time CMs has been highly unstable (Song et al., 2023; Geng et al., 2024). As a result, they perform significantly worse compared to discrete-time CMs in prior works. To address this issue, we build upon the TrigFlow framework and introduce several theoretically motivated improvements to stabilize continuous-time CMs, with a focus on parameterization, network architecture, and training objectives.

## 4.1 PARAMETERIZATION AND NETWORK ARCHITECTURE

Key to the training of continuous-time CMs is Eq. (2), which depends on the tangent function $\frac{\mathrm{d}\boldsymbol{f}_{\theta^-}(\boldsymbol{x}_t, t)}{\mathrm{d}t}$. Under the TrigFlow formulation, this tangent function is given by

$$\frac{\mathrm{d}\boldsymbol{f}_{\theta^-}(\boldsymbol{x}_t, t)}{\mathrm{d}t} = -\cos(t) \left( \sigma_d \boldsymbol{F}_{\theta^-} \left( \frac{\boldsymbol{x}_t}{\sigma_d}, t \right) - \frac{\mathrm{d}\boldsymbol{x}_t}{\mathrm{d}t} \right) - \sin(t) \left( \boldsymbol{x}_t + \sigma_d \frac{\mathrm{d}\boldsymbol{F}_{\theta^-} \left( \frac{\boldsymbol{x}_t}{\sigma_d}, t \right)}{\mathrm{d}t} \right), \tag{6}$$

where $\frac{\mathrm{d}\boldsymbol{x}_t}{\mathrm{d}t}$ represents the PF-ODE, which is either estimated using a pretrained diffusion model in consistency distillation, or using an unbiased estimator calculated from noise and clean samples in consistency training.

To stabilize training, it is necessary to ensure the tangent function in Eq. (6) is stable across different time steps. Empirically, we found that $\sigma_d \boldsymbol{F}_{\theta^-}$, the PF-ODE $\frac{\mathrm{d}\boldsymbol{x}_t}{\mathrm{d}t}$, and the noisy sample $\boldsymbol{x}_t$ are all relatively stable. The only term left in the tangent function now is $\sin(t)\frac{\mathrm{d}\boldsymbol{F}_{\theta^-}}{\mathrm{d}t} = \sin(t)\nabla_{\boldsymbol{x}_t} \boldsymbol{F}_{\theta^-} \frac{\mathrm{d}\boldsymbol{x}_t}{\mathrm{d}t} + \sin(t)\partial_t \boldsymbol{F}_{\theta^-}$. After further analysis, we found $\nabla_{\boldsymbol{x}_t} \boldsymbol{F}_{\theta^-} \frac{\mathrm{d}\boldsymbol{x}_t}{\mathrm{d}t}$ is typically well-conditioned, so instability originates from the time-derivative $\sin(t)\partial_t \boldsymbol{F}_{\theta^-}$, which can be decomposed according to

$$\sin(t)\partial_t \boldsymbol{F}_{\theta^-} = \sin(t) \frac{\partial c_{\text{noise}}(t)}{\partial t} \cdot \frac{\partial \text{emb}(c_{\text{noise}})}{\partial c_{\text{noise}}} \cdot \frac{\partial \boldsymbol{F}_{\theta^-}}{\partial \text{emb}(c_{\text{noise}})}, \tag{7}$$

where $\text{emb}(\cdot)$ refers to the time embeddings, typically in the form of either positional embeddings (Ho et al., 2020; Vaswani, 2017) or Fourier embeddings (Song et al., 2021b; Tancik et al., 2020) in the literature of diffusion models and CMs.

Below we describe improvements to stabilize each component from Eq. (7) in turns.

**Identity Time Transformation ($c_{\text{noise}}(t) = t$).** Most existing CMs use the EDM formulation, which can be directly translated to the TrigFlow formulation as described in Appendix B.2. In particular, the time transformation becomes $c_{\text{noise}}(t) \propto \log(\sigma_d \tan t)$. Straightforward derivation shows that with this $c_{\text{noise}}(t)$, $\sin(t) \cdot \partial_t c_{\text{noise}}(t) = 1/\cos(t)$ blows up whenever $t \to \frac{\pi}{2}$. To mitigate numerical instability, we propose to use $c_{\text{noise}}(t) = t$ as the default time transformation.

**Positional Time Embeddings.** For general time embeddings in the form of $\text{emb}(c) = \sin(s \cdot 2\pi\omega \cdot c + \phi)$, we have $\partial_c \text{emb}(c) = s \cdot 2\pi\omega \cos(s \cdot 2\pi\omega \cdot c + \phi)$. With larger Fourier scale $s$, this derivative

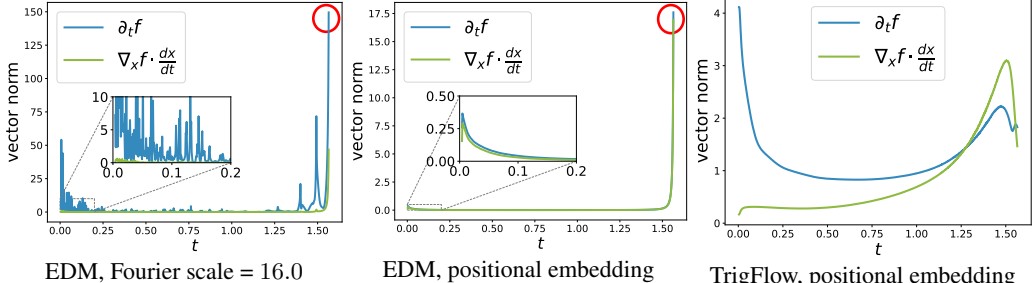

EDM, Fourier scale = 16.0          EDM, positional embedding          TrigFlow, positional embedding

Figure 4: **Stability of different formulations.** We show the norms of both terms in $\frac{\mathrm{d} \boldsymbol{f}_{\theta-}}{\mathrm{d}t} = \nabla_{\boldsymbol{x}} \boldsymbol{f}_{\theta-} \cdot \frac{\mathrm{d}\boldsymbol{x}_t}{\mathrm{d}t} + \partial_t \boldsymbol{f}_{\theta-}$ for diffusion models trained with the EDM ($c_{\text{noise}}(t) = \log(\sigma_d \tan(t))$) and TrigFlow ($c_{\text{noise}}(t) = t$) formulations using different time embeddings. We observe that large Fourier scales in Fourier embeddings cause instabilities. In addition, the EDM formulation suffers from numerical issues when $t \to \frac{\pi}{2}$, while TrigFlow (using positional embeddings) has stable partial derivatives for both $\boldsymbol{x}_t$ and $t$.

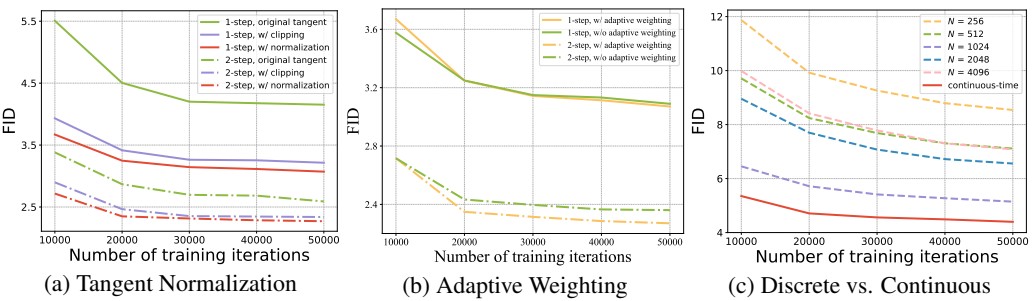

(a) Tangent Normalization          (b) Adaptive Weighting          (c) Discrete vs. Continuous

Figure 5: **Comparing different training objectives for consistency distillation.** The diffusion models are EDM2 (Karras et al., 2024) pretrained on ImageNet 512×512. (a) 1-step and 2-step sampling of continuous-time CMs trained by using raw tangents $\frac{\mathrm{d}\boldsymbol{f}_{\theta-}}{\mathrm{d}t}$, clipped tangents clip($\frac{\mathrm{d}\boldsymbol{f}_{\theta-}}{\mathrm{d}t}, -1, 1$) and normalized tangents ($\frac{\mathrm{d}\boldsymbol{f}_{\theta-}}{\mathrm{d}t}$)/($\|\frac{\mathrm{d}\boldsymbol{f}_{\theta-}}{\mathrm{d}t}\| + 0.1$). (b) Quality of 1-step and 2-step samples from continuous-time CMs trained w/ and w/o adaptive weighting, both are w/ tangent normalization. (c) Quality of 1-step samples from continuous-time CMs vs. discrete-time CMs using varying number of time steps ($N$), trained using all techniques in Sec. 4.

has greater magnitudes and oscillates more vibrantly, causing worse instability. To avoid this, we use positional embeddings, which amounts to $s \approx 0.02$ in Fourier embeddings. This analysis provides a principled explanation for the observations in Song & Dhariwal (2023).

**Adaptive Double Normalization.** Song & Dhariwal (2023) found that the AdaGN layer (Dhariwal & Nichol, 2021), defined as $\boldsymbol{y} = \text{norm}(\boldsymbol{x}) \odot \boldsymbol{s}(t) + \boldsymbol{b}(t)$, negatively causes CM training to diverge. Our modification is *adaptive double normalization*, defined as $\boldsymbol{y} = \text{norm}(\boldsymbol{x}) \odot \text{pnorm}(\boldsymbol{s}(t)) + \text{pnorm}(\boldsymbol{b}(t))$, where pnorm($\cdot$) denotes pixel normalization (Karras, 2017). Empirically we find it retains the expressive power of AdaGN for diffusion training but removes its instability in CM training.

As shown in Figure 4, we visualize how our techniques stabilize the time-derivates for CMs trained on CIFAR-10. Empirically, we find that these improvements help stabilize the training dynamics of CMs without hurting diffusion model training (see Appendix G).

## 4.2 TRAINING OBJECTIVES

Using the TrigFlow formulation in Sec. 3 and techniques proposed in Sec. 4.1, the gradient of continuous-time CM training in Eq. (2) becomes

$$\nabla_\theta \mathbb{E}_{\boldsymbol{x}_t, t} \left[ -w(t)\sigma_d \sin(t) \boldsymbol{F}_\theta^\top \left( \frac{\boldsymbol{x}_t}{\sigma_d}, t \right) \frac{\mathrm{d}\boldsymbol{f}_{\theta-}(\boldsymbol{x}_t, t)}{\mathrm{d}t} \right].$$

Below we propose additional techniques to explicitly control this gradient for improved stability.

**Tangent Normalization.** As discussed in Sec. 4.1, most gradient variance in CM training comes from the tangent function $\frac{\mathrm{d}\boldsymbol{f}_{\theta-}}{\mathrm{d}t}$. We propose to explicitly normalize the tangent function by replacing $\frac{\mathrm{d}\boldsymbol{f}_{\theta-}}{\mathrm{d}t}$ with $\frac{\mathrm{d}\boldsymbol{f}_{\theta-}}{\mathrm{d}t}/(\|\frac{\mathrm{d}\boldsymbol{f}_{\theta-}}{\mathrm{d}t}\| + c)$, where we empirically set $c = 0.1$. Alternatively, we can clip the

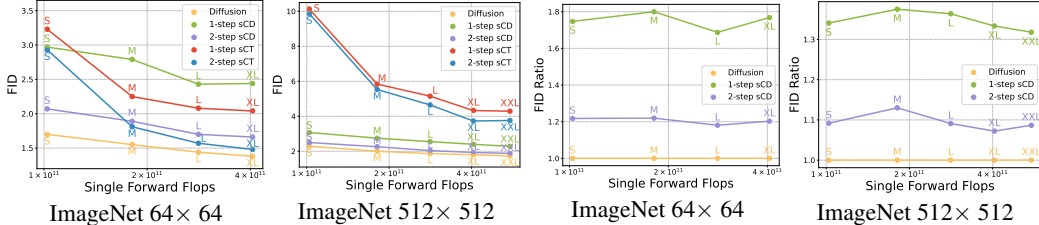

|  |  |
|---|---|
| ImageNet $64 \times 64$ | ImageNet $512 \times 512$ |

(a) FID ($\downarrow$) as a function of single forward flops.

|  |  |
|---|---|
| ImageNet $64 \times 64$ | ImageNet $512 \times 512$ |

(b) FID ratio ($\downarrow$) as a function of single forward flops.

Figure 6: **sCD scales commensurately with teacher diffusion models**. We plot the (a) FID and (b) FID ratio against the teacher diffusion model (at the same model size) on ImageNet $64 \times 64$ and $512 \times 512$. sCD scales better than sCT, and has a *constant offset* in the FID ratio across all model sizes, implying that sCD has the same scaling property of the teacher diffusion model. Furthermore, the offset diminishes with more sampling steps. tangent within $[-1, 1]$, which also caps its variance. Our results in Figure 5(a) demonstrate that either normalization or clipping leads to substantial improvements for the training of continuous-time CMs.

**Adaptive Weighting.** Previous works (Song & Dhariwal, 2023; Geng et al., 2024) design weighting functions $w(t)$ manually for CM training, which can be suboptimal for different data distributions and network architectures. Following EDM2 (Karras et al., 2024), we propose to train an adaptive weighting function alongside the CM, which not only eases the burden of hyperparameter tuning but also outperforms manually designed weighting functions with better empirical performance and negligible training overhead. Key to our approach is the observation that $\nabla_\theta \mathbb{E}[\boldsymbol{F}_\theta^\top \boldsymbol{y}] = \frac{1}{2} \nabla_\theta \mathbb{E}[\|\boldsymbol{F}_\theta - \boldsymbol{F}_{\theta^-} + \boldsymbol{y}\|_2^2]$, where $\boldsymbol{y}$ is an arbitrary vector independent of $\theta$. When training continuous-time CMs using Eq. (2), we have $\boldsymbol{y} = -w(t)\sigma_d \sin(t)\frac{\mathrm{d}\boldsymbol{f}_{\theta^-}}{\mathrm{d}t}$. This observation allows us to convert Eq. (2) into the gradient of an MSE objective. We can therefore use the same approach in Karras et al. (2024) to train an adaptive weighting function that minimizes the variance of MSE losses across time steps (details in Appendix D). In practice, we find that integrating a prior weighting $w(t) = \frac{1}{\sigma_d \tan(t)}$ further reduces training variance. By incorporating the prior weighting, we train both the network $\boldsymbol{F}_\theta$ and the adaptive weighting function $w_\phi(t)$ by minimizing

$$\mathcal{L}_{\text{sCM}}(\theta, \phi) := \mathbb{E}_{\boldsymbol{x}_t, t}\left[\frac{e^{w_\phi(t)}}{D}\left\|\boldsymbol{F}_\theta\left(\frac{\boldsymbol{x}_t}{\sigma_d}, t\right) - \boldsymbol{F}_{\theta^-}\left(\frac{\boldsymbol{x}_t}{\sigma_d}, t\right) - \cos(t)\frac{\mathrm{d}\boldsymbol{f}_{\theta^-}(\boldsymbol{x}_t, t)}{\mathrm{d}t}\right\|_2^2 - w_\phi(t)\right], \quad (8)$$

where $D$ is the dimensionality of $\boldsymbol{x}_0$, and we sample $\tan(t)$ from a log-Normal proposal distribution (Karras et al., 2022), that is, $e^{\sigma_d \tan(t)} \sim \mathcal{N}(P_{\text{mean}}, P_{\text{std}}^2)$ (details in Appendix G).

**Diffusion Finetuning and Tangent Warmup.** For consistency distillation, we find that finetuning the CM from a pretrained diffusion model can speed up convergence, which is consistent with Song et al. (2023); Geng et al. (2024). Recall that in Eq. (6), the tangent $\frac{\mathrm{d}\boldsymbol{f}_{\theta^-}}{\mathrm{d}t}$ can be decomposed into two parts: the first term $\cos(t)(\sigma_d \boldsymbol{F}_{\theta^-} - \frac{\mathrm{d}\boldsymbol{x}_t}{\mathrm{d}t})$ is relatively stable, whereas the second term $\sin(t)(\boldsymbol{x}_t + \sigma_d\frac{\mathrm{d}\boldsymbol{F}_{\theta^-}}{\mathrm{d}t})$ may cause instability. We introduce an optional technique named as *tangent warmup* by replacing the coefficient $\sin(t)$ with $r \cdot \sin(t)$, where $r$ linearly increases from 0 to 1 over the first 10k training iterations. We find that the tangent normalization does not affect sample quality but may reduce some gradient spikes during training.

With all techniques in place, the stability of both discrete-time and continuous-time CM training substantially improves. We provide detailed algorithms for discrete-time CMs in Appendix E, and train continuous-time CMs and discrete-time CMs with the same setting. As demonstrated in Figure 5(c), increasing the number of discretization steps $N$ in discrete-time CMs improves sample quality by reducing discretization errors, but degrades once $N$ becomes too large (after $N > 1024$) to suffer from numerical precision issues. By contrast, continuous-time CMs significantly outperform discrete-time CMs across all $N$'s which provides strong justification for choosing continuous-time CMs over discrete-time counterparts. We call our model **sCM** (s for *simple, stable, and scalable*), and provide detailed pseudo-code for sCM training in Appendix A.

## 5 SCALING UP CONTINUOUS-TIME CONSISTENCY MODELS

Below we test all the improvements proposed in previous sections by training large-scale sCMs on a variety of challenging datasets.

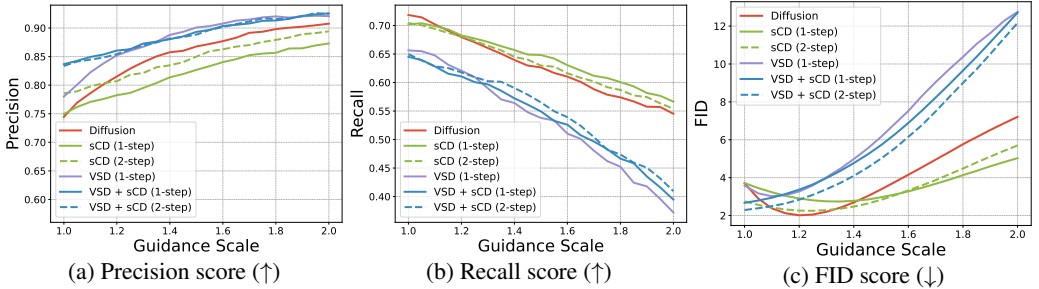

Figure 7: **sCD has higher diversity compared to VSD**: Sample quality comparison of the EDM2 (Karras et al., 2024) diffusion model, VSD (Wang et al., 2024; Yin et al., 2024b), sCD, and the combination of VSD and sCD, across varying guidance scales. All models are of EDM2-M size and trained on ImageNet 512×512.

Table 1: Sample quality on unconditional CIFAR-10 and class-conditional ImageNet 64× 64.

**Unconditional CIFAR-10**

| METHOD | NFE ($\downarrow$) | FID ($\downarrow$) |
| --- | --- | --- |
| **Diffusion models & Fast Samplers** | | |
| Score SDE (deep) (Song et al., 2021b) | 2000 | 2.20 |
| EDM (Karras et al., 2022) | 35 | 2.01 |
| Flow Matching (Lipman et al., 2022) | 142 | 6.35 |
| OT-CFM (Tong et al., 2023) | 1000 | 3.57 |
| DPM-Solver (Lu et al., 2022a) | 10 | 4.70 |
| DPM-Solver++ (Lu et al., 2022b) | 10 | 2.91 |
| DPM-Solver-v3 (Zheng et al., 2023c) | 10 | 2.51 |
| **Joint Training** | | |
| Diffusion GAN (Xiao et al., 2022) | 4 | 3.75 |
| Diffusion StyleGAN (Wang et al., 2022) | 1 | 3.19 |
| StyleGAN-XL (Sauer et al., 2022) | 1 | 1.52 |
| CTM (Kim et al., 2023) | 1 | **1.87** |
| Diff-Instruct (Luo et al., 2024) | 1 | 4.53 |
| DMD (Yin et al., 2024b) | 1 | 3.77 |
| SiD (Zhou et al., 2024) | 1 | 1.92 |
| **Diffusion Distillation** | | |
| DFNO (LPIPS) (Zheng et al., 2023b) | 1 | 3.78 |
| 2-Rectified Flow (Liu et al., 2022) | 1 | 4.85 |
| PID (LPIPS) (Tee et al., 2024) | 1 | 3.92 |
| Consistency-FM (Yang et al., 2024) | 2 | 5.34 |
| PD (Salimans & Ho, 2022) | 1 | 8.34 |
| | 2 | 5.58 |
| TRACT (Berthelot et al., 2023) | 1 | 3.78 |
| | 2 | 3.32 |
| CD (LPIPS) (Song et al., 2023) | 1 | **3.55** |
| | 2 | 2.93 |
| **sCD (ours)** | 1 | 3.66 |
| | 2 | **2.52** |
| **Consistency Training** | | |
| iCT (Song & Dhariwal, 2023) | 1 | 2.83 |
| | 2 | 2.46 |
| iCT-deep (Song & Dhariwal, 2023) | 1 | **2.51** |
| | 2 | 2.24 |
| ECT (Geng et al., 2024) | 1 | 3.60 |
| | 2 | 2.11 |
| **sCT (ours)** | 1 | 2.85 |
| | 2 | **2.06** |

**Class-Conditional ImageNet 64×64**

| METHOD | NFE ($\downarrow$) | FID ($\downarrow$) |
| --- | --- | --- |
| **Diffusion models & Fast Samplers** | | |
| ADM (Dhariwal & Nichol, 2021) | 250 | 2.07 |
| RIN (Jabri et al., 2022) | 1000 | 1.23 |
| DPM-Solver (Lu et al., 2022a) | 20 | 3.42 |
| EDM (Heun) (Karras et al., 2022) | 79 | 2.44 |
| EDM2 (Heun) (Karras et al., 2024) | 63 | 1.33 |
| **Joint Training** | | |
| StyleGAN-XL (Sauer et al., 2022) | 1 | 1.52 |
| Diff-Instruct (Luo et al., 2024) | 1 | 5.57 |
| EMD (Xie et al., 2024b) | 1 | 2.20 |
| DMD (Yin et al., 2024b) | 1 | 2.62 |
| DMD2 (Yin et al., 2024a) | 1 | **1.28** |
| SiD (Zhou et al., 2024) | 1 | 1.52 |
| CTM (Kim et al., 2023) | 1 | 1.92 |
| | 2 | 1.73 |
| Moment Matching (Salimans et al., 2024) | 1 | 3.00 |
| | 2 | 3.86 |
| **Diffusion Distillation** | | |
| DFNO (LPIPS) (Zheng et al., 2023b) | 1 | 7.83 |
| PID (LPIPS) (Tee et al., 2024) | 1 | 9.49 |
| TRACT (Berthelot et al., 2023) | 1 | 7.43 |
| | 2 | 4.97 |
| PD (Salimans & Ho, 2022) | 1 | 10.70 |
| (reimpl. from Heek et al. (2024)) | 2 | 4.70 |
| CD (LPIPS) (Song et al., 2023) | 1 | 6.20 |
| | 2 | 4.70 |
| MultiStep-CD (Heek et al., 2024) | 1 | 3.20 |
| | 2 | 1.90 |
| **sCD (ours)** | 1 | **2.44** |
| | 2 | **1.66** |
| **Consistency Training** | | |
| iCT (Song & Dhariwal, 2023) | 1 | 4.02 |
| | 2 | 3.20 |
| iCT-deep (Song & Dhariwal, 2023) | 1 | 3.25 |
| | 2 | 2.77 |
| ECT (Geng et al., 2024) | 1 | 2.49 |
| | 2 | 1.67 |
| **sCT (ours)** | 1 | **2.04** |
| | 2 | **1.48** |

## 5.1 TANGENT COMPUTATION IN LARGE-SCALE MODELS

The common setting for training large-scale diffusion models includes using half-precision (FP16) and Flash Attention (Dao et al., 2022; Dao, 2023). As training continuous-time CMs requires computing the tangent $\frac{\mathrm{d}\boldsymbol{f}_{\theta-}}{\mathrm{d}t}$ accurately, we need to improve numerical precision and also support memory-efficient attention computation, as detailed below.

**JVP Rearrangement.** Computing $\frac{\mathrm{d}\boldsymbol{f}_{\theta-}}{\mathrm{d}t}$ involves calculating $\frac{\mathrm{d}\boldsymbol{F}_{\theta-}}{\mathrm{d}t} = \nabla_{\boldsymbol{x}_t}\boldsymbol{F}_{\theta-} \cdot \frac{\mathrm{d}\boldsymbol{x}_t}{\mathrm{d}t} + \partial_t \boldsymbol{F}_{\theta-}$, which can be efficiently obtained via the Jacobian-vector product (JVP) for $\boldsymbol{F}_{\theta-}\left(\frac{\cdot}{\sigma_d}, \cdot\right)$ with the input

Table 2: Sample quality on class-conditional ImageNet $512 \times 512$. [†]Our reimplemented teacher diffusion model based on EDM2 (Karras et al., 2024) but with modifications in Sec. 4.1.

| METHOD | NFE (↓) | FID (↓) | #Params | METHOD | NFE (↓) | FID (↓) | #Params |
|---|---|---|---|---|---|---|---|
| **Diffusion models** | | | | [†]**Teacher Diffusion Model** | | | |
| ADM-G (Dhariwal & Nichol, 2021) | 250×2 | 7.72 | 559M | EDM2-S (Karras et al., 2024) | 63×2 | 2.29 | 280M |
| RIN (Jabri et al., 2022) | 1000 | 3.95 | 320M | EDM2-M (Karras et al., 2024) | 63×2 | 2.00 | 498M |
| U-ViT-H/4 (Bao et al., 2023) | 250×2 | 4.05 | 501M | EDM2-L (Karras et al., 2024) | 63×2 | 1.87 | 778M |
| DiT-XL/2 (Peebles & Xie, 2023) | 250×2 | 3.04 | 675M | EDM2-XL (Karras et al., 2024) | 63×2 | 1.80 | 1.1B |
| SimDiff (Hoogeboom et al., 2023) | 512×2 | 3.02 | 2B | EDM2-XXL (Karras et al., 2024) | 63×2 | 1.73 | 1.5B |
| VDM++ (Kingma & Gao, 2024) | 512×2 | 2.65 | 2B | **Consistency Training (sCT, ours)** | | | |
| DiffiT (Hatamizadeh et al., 2023) | 250×2 | 2.67 | 561M | | | | |
| DiMR-XL/3R (Liu et al., 2024) | 250×2 | 2.89 | 525M | sCT-S (ours) | 1 | 10.13 | 280M |
| DiffuSSM-XL (Yan et al., 2024) | 250×2 | 3.41 | 673M | | 2 | 9.86 | 280M |
| DiM-H (Teng et al., 2024) | 250×2 | 3.78 | 860M | sCT-M (ours) | 1 | 5.84 | 498M |
| U-DiT (Tian et al., 2024b) | 250 | 15.39 | 204M | | 2 | 5.53 | 498M |
| SiT-XL (Ma et al., 2024) | 250×2 | 2.62 | 675M | sCT-L (ours) | 1 | 5.15 | 778M |
| Large-DiT (Alpha-VLLM, 2024) | 250×2 | 2.52 | 3B | | 2 | 4.65 | 778M |
| MaskDiT (Zheng et al., 2023a) | 79×2 | 2.50 | 736M | sCT-XL (ours) | 1 | 4.33 | 1.1B |
| DiS-H/2 (Fei et al., 2024a) | 250×2 | 2.88 | 900M | | 2 | 3.73 | 1.1B |
| DRWKV-H/2 (Fei et al., 2024b) | 250×2 | 2.95 | 779M | sCT-XXL (ours) | 1 | 4.29 | 1.5B |
| EDM2-S (Karras et al., 2024) | 63×2 | 2.23 | 280M | | 2 | 3.76 | 1.5B |
| EDM2-M (Karras et al., 2024) | 63×2 | 2.01 | 498M | | | | |
| EDM2-L (Karras et al., 2024) | 63×2 | 1.88 | 778M | **Consistency Distillation (sCD, ours)** | | | |
| EDM2-XL (Karras et al., 2024) | 63×2 | 1.85 | 1.1B | sCD-S | 1 | 3.07 | 280M |
| EDM2-XXL (Karras et al., 2024) | 63×2 | **1.81** | 1.5B | | 2 | 2.50 | 280M |
| **GANs & Masked Models** | | | | sCD-M | 1 | 2.75 | 498M |
| | | | | | 2 | 2.26 | 498M |
| BigGAN (Brock, 2018) | 1 | 8.43 | 160M | sCD-L | 1 | 2.55 | 778M |
| StyleGAN-XL (Sauer et al., 2022) | 1×2 | 2.41 | 168M | | 2 | 2.04 | 778M |
| VQGAN (Esser et al., 2021) | 1024 | 26.52 | 227M | sCD-XL | 1 | 2.40 | 1.1B |
| MaskGIT (Chang et al., 2022) | 12 | 7.32 | 227M | | 2 | 1.93 | 1.1B |
| MAGVIT-v2 (Yu et al., 2023) | 64×2 | 1.91 | 307M | sCD-XXL | 1 | **2.28** | 1.5B |
| MAR (Li et al., 2024) | 64×2 | **1.73** | 481M | | 2 | **1.88** | 1.5B |
| VAR-$d$36-s (Tian et al., 2024a) | 10×2 | 2.63 | 2.3B | | | | |

vector $(\boldsymbol{x}_t, t)$ and the tangent vector $(\frac{\mathrm{d}\boldsymbol{x}_t}{\mathrm{d}t}, 1)$. However, we empirically find that the tangent may overflow in intermediate layers when $t$ is near 0 or $\frac{\pi}{2}$. To improve numerical precision, we propose to rearrange the computation of the tangent. Specifically, since the objective in Eq. (8) contains $\cos(t)\frac{\mathrm{d}\boldsymbol{f}_{\theta-}}{\mathrm{d}t}$ and $\frac{\mathrm{d}\boldsymbol{f}_{\theta-}}{\mathrm{d}t}$ is proportional to $\sin(t)\frac{\mathrm{d}\boldsymbol{F}_{\theta-}}{\mathrm{d}t}$, we can compute the JVP as:

$$\cos(t)\sin(t)\frac{\mathrm{d}\boldsymbol{F}_{\theta-}}{\mathrm{d}t} = \left(\nabla_{\frac{\boldsymbol{x}_t}{\sigma_d}}\boldsymbol{F}_{\theta-}\right) \cdot \left(\cos(t)\sin(t)\frac{\mathrm{d}\boldsymbol{x}_t}{\mathrm{d}t}\right) + \partial_t \boldsymbol{F}_{\theta-} \cdot (\cos(t)\sin(t)\sigma_d),$$

which is the JVP for $\boldsymbol{F}_{\theta-}(\cdot, \cdot)$ with the input $(\frac{\boldsymbol{x}_t}{\sigma_d}, t)$ and the tangent $(\cos(t)\sin(t)\frac{\mathrm{d}\boldsymbol{x}_t}{\mathrm{d}t}, \cos(t)\sin(t)\sigma_d)$. This rearrangement greatly alleviates the overflow issues in the intermediate layers, resulting in more stable training in FP16.

**JVP of Flash Attention.** Flash Attention (Dao et al., 2022; Dao, 2023) is widely used for attention computation in large-scale model training, providing both GPU memory savings and faster training. However, Flash Attention does not compute the Jacobian-vector product (JVP). To fill this gap, we propose a similar algorithm (detailed in Appendix F) that efficiently computes both softmax self-attention and its JVP in a single forward pass in the style of Flash Attention, significantly reducing GPU memory usage for JVP computation in attention layers.

## 5.2 EXPERIMENTS

To test our improvements, we employ both consistency training (referred to as **sCT**) and consistency distillation (referred to as **sCD**) to train and scale continuous-time CMs on CIFAR-10 (Krizhevsky, 2009), ImageNet 64×64 and ImageNet 512×512 (Deng et al., 2009). We benchmark the sample quality using FID (Heusel et al., 2017). We follow the settings of Score SDE (Song et al., 2021b) on CIFAR10 and EDM2 (Karras et al., 2024) on both ImageNet 64×64 and ImageNet 512×512, while changing the parameterization and architecture according to Section 4.1. We adopt the method proposed by Song et al. (2023) for two-step sampling of both sCT and sCD, using a fixed intermediate time step $t = 1.1$. For sCD models on ImageNet 512×512, since the teacher diffusion model relies on classifier-free guidance (CFG) (Ho & Salimans, 2021), we incorporate an additional input $s$ into the model $\boldsymbol{F}_\theta$ to represent the guidance scale (Meng et al., 2023). We train the model with sCD

by uniformly sampling $s \in [1, 2]$ and applying the corresponding CFG to the teacher model during distillation (more details are provided in Appendix G). For sCT models, we do not test CFG since it is incompatible with consistency training.

**Training compute of sCM.** We use the same batch size as the teacher diffusion model across all datasets. The effective compute per training iteration of sCD is approximately twice that of the teacher model. We observe that the quality of two-step samples from sCD converges rapidly, achieving results comparable to the teacher diffusion model using less than 20% of the teacher training compute. In practice, we can obtain high-quality samples after only 20k finetuning iterations with sCD.

**Benchmarks.** In Tables 1 and 2, we compare our results with previous methods by benchmarking the FIDs and the number of function evaluations (NFEs). First, sCM outperforms all previous few-step methods that do not rely on joint training with another network and is on par with, or even exceeds, the best results achieved with adversarial training. Notably, the 1-step FID of sCD-XXL on ImageNet 512×512 surpasses that of StyleGAN-XL (Sauer et al., 2022) and VAR (Tian et al., 2024a). Furthermore, the two-step FID of sCD-XXL outperforms all generative models except diffusion and is comparable with the best diffusion models that require 63 sequential steps. Second, the two-step sCM model significantly narrows the FID gap with the teacher diffusion model to within 10%, achieving FIDs of 2.06 on CIFAR-10 (compared to the teacher FID of 2.01), 1.48 on ImageNet 64×64 (teacher FID of 1.33), and 1.88 on ImageNet 512×512 (teacher FID of 1.73). Additionally, we observe that sCT is more effective at smaller scales but suffers from increased variance at larger scales, while sCD shows consistent performance across both small and large scales.

**Scaling study.** Based on our improved training techniques, we successfully scale continuous-time CMs without training instability. We train various sizes of sCMs using EDM2 configurations (S, M, L, XL, XXL) on ImageNet 64×64 and 512×512, and evaluate FID under optimal guidance scales, as shown in Fig. 6. First, as model FLOPs increase, both sCT and sCD show improved sample quality, showing that both methods benefit from scaling. Second, compared to sCD, sCT is more compute efficient at smaller resolutions but less efficient at larger resolutions. Third, sCD scales predictably for a given dataset, maintaining a consistent relative difference in FIDs across model sizes. This suggests that the FID of sCD decreases at the same rate as the teacher diffusion model, and therefore *sCD is as scalable as the teacher diffusion model*. As the FID of the teacher diffusion model decreases with scaling, the *absolute* difference in FID between sCD and the teacher model also diminishes. Finally, the relative difference in FIDs decreases with more sampling steps, and the sample quality of the two-step sCD becomes on par with that of the teacher diffusion model.

**Comparison with VSD.** Variational score distillation (VSD) (Wang et al., 2024; Yin et al., 2024b) and its multi-step generalization (Xie et al., 2024b; Salimans et al., 2024) represent another diffusion distillation technique that has demonstrated scalability on high-resolution images (Yin et al., 2024a). We apply one-step VSD from time $T$ to 0 to finetune a teacher diffusion model using the EDM2-M configuration and tune both the weighting functions and proposal distributions for fair comparisons. As shown in Figure 7, we compare sCD, VSD, a combination of sCD and VSD (by simply adding the two losses), and the teacher diffusion model by sweeping over the guidance scale. We observe that VSD has artifacts similar to those from applying large guidance scales in diffusion models: it increases fidelity (as evidenced by higher precision scores) while decreasing diversity (as shown by lower recall scores). This effect becomes more pronounced with increased guidance scales, ultimately causing severe mode collapse. In contrast, the precision and recall scores from two-step sCD are comparable with those of the teacher diffusion model, resulting in better FID scores than VSD.

## 6 CONCLUSION

Our improved formulations, architectures, and training objectives have simplified and stabilized the training of continuous-time consistency models, enabling smooth scaling up to 1.5 billion parameters on ImageNet 512×512. We ablated the impact of TrigFlow formulation, tangent normalization, and adaptive weighting, confirming their effectiveness. Combining these improvements, our method demonstrated predictable scalability across datasets and model sizes, outperforming other few-step sampling approaches at large scales. Notably, we narrowed the FID gap with the teacher model to within 10% using two-step generation, compared to state-of-the-art diffusion models that require significantly more sampling steps.

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

## DISCUSSIONS AND LIMITATIONS

**sCT is less effective than sCD in latent spaces.** As listed in Tables 1 and 2, sCT consistently outperforms sCD on CIFAR-10 and ImageNet 64×64 but is less effective than sCD across different model scales on ImageNet 512×512. We believe the higher training variance of CT is the main issue, particularly in the complex latent spaces defined by the pretrained encoder. We hypothesize that the current encoder/decoder may not be optimal for consistency models. Theoretically, since the ground truth mapping in consistency models aims to transform a Gaussian distribution into a multimodal data distribution with potentially disconnected supports, its tangent can become ill-conditioned at boundary points, resulting in worse optimization dynamics. If we could develop a better encoder/decoder to create a more well-conditioned ground truth mapping in the latent space, the training of consistency models would likely become significantly easier.

**Computation costs of sCM.** As Jacobian-vector product can be efficiently computed using *forward-mode automatic differentiation*, which requires the same memory and compute as a standard forward pass without saving intermediate activations. This is significantly cheaper than backpropagation, which relies on reverse-mode automatic differentiation. Consequently, our continuous-time consistency models require similar compute and memory to train when compared to their discrete-time counterparts, which perform two forward passes at each iteration.

**Limitations.** Despite large improvements in FID scores, our method can still produce images with noticeable artifacts. These artifacts are commonly observed when training generative models on the ImageNet dataset with class labels, whereas training on larger datasets with caption conditions may significantly alleviate this issue. Furthermore, our 2-step sCM still shows a small gap compared to state-of-the-art diffusion models, which we believe may be further reduced by incorporating our proposed techniques into multi-step consistency models (Heek et al., 2024). Additionally, since FID scores do not capture all semantic details, further validation is needed to determine whether our method can scale effectively to image or video generation tasks that require larger resolutions and fine details. Besides, ensuring the training stability of sCM requires several significant modifications of the network architecture, thus sCM may be not suitable for some architectures designed for diffusion models. Moreover, our best performing method sCD still highly relies on the performance of a pretrained diffusion models, which restricts the architecture family and potentially limits the few-step generation performance. Addressing these quality issues might require new sampling strategies or enhanced architectures to maintain high fidelity even with limited sampling steps.

## APPENDIX

We include additional derivations, experimental details, and results in the appendix. The detailed training algorithm for sCM, covering both sCT and sCD, is provided in Appendix A. We present a comprehensive discussion of the TrigFlow framework in Appendix B, including detailed derivations (Appendix B.1) and its connections with other parameterization (Appendix B.2). We introduce a new algorithm called *adaptive variational score distillation* in Appendix C, which eliminates the need for manually designed training weighting. Furthermore, we elaborate on a general framework for adaptive training weighting in Appendix D, applicable to diffusion models, consistency models, and variational score distillation. As our improvements discussed in Sec. 4 are also applicable for discrete-time consistency models, we provide detailed derivations and the training algorithm for discrete-time consistency models in Appendix E, incorporating all the improved techniques of sCM. We also provide a complete description of the Jacobian-vector product algorithm for Flash Attention in Appendix F. Finally, all experimental settings and evaluation results are listed in Appendix G, along with additional samples generated by our sCD-XXL model trained on ImageNet at 512×512 resolution in Appendix H.

## A   TRAINING ALGORITHM OF SCM

We provide the detailed algorithm of sCM in Algorithm 1, where we refer to consistency training of sCM as **sCT** and consistency distillation of sCM as **sCD**.

---

**Algorithm 1** Simplified and Stabilized Continuous-time Consistency Models (sCM).

---

**Input:** dataset $\mathcal{D}$ with std. $\sigma_d$, pretrained diffusion model $\boldsymbol{F}_{\text{pretrain}}$ with parameter $\theta_{\text{pretrain}}$, model $\boldsymbol{F}_\theta$, weighting $w_\phi$, learning rate $\eta$, proposal $(P_{\text{mean}}, P_{\text{std}})$, constant $c$, warmup iteration $H$.
**Init:** $\theta \leftarrow \theta_{\text{pretrain}}$, Iters $\leftarrow 0$.
**repeat**

    $\boldsymbol{x}_0 \sim \mathcal{D}, \boldsymbol{z} \sim \mathcal{N}(\boldsymbol{0}, \sigma_d^2 \boldsymbol{I}), \tau \sim \mathcal{N}(P_{\text{mean}}, P_{\text{std}}^2), t \leftarrow \arctan(\frac{e^\tau}{\sigma_d}), \boldsymbol{x}_t \leftarrow \cos(t)\boldsymbol{x}_0 + \sin(t)\boldsymbol{z}$

    $\frac{d\boldsymbol{x}_t}{dt} \leftarrow \cos(t)\boldsymbol{z} - \sin(t)\boldsymbol{x}_0$ if consistency training else $\frac{d\boldsymbol{x}_t}{dt} \leftarrow \sigma_d \boldsymbol{F}_{\text{pretrain}}(\frac{\boldsymbol{x}_t}{\sigma_d}, t)$

    $r \leftarrow \min(1, \text{Iters}/H)$                                                         ▷ Tangent warmup

    $\boldsymbol{g} \leftarrow -\cos^2(t)(\sigma_d \boldsymbol{F}_{\theta^-} - \frac{d\boldsymbol{x}_t}{dt}) - r \cdot \cos(t)\sin(t)(\boldsymbol{x}_t + \sigma_d \frac{d\boldsymbol{F}_{\theta^-}}{dt})$   ▷ JVP rearrangement

    $\boldsymbol{g} \leftarrow \boldsymbol{g}/(\|\boldsymbol{g}\| + c)$                                                    ▷ Tangent normalization

    $\mathcal{L}(\theta, \phi) \leftarrow \frac{e^{w_\phi(t)}}{D}\|\boldsymbol{F}_\theta(\frac{\boldsymbol{x}_t}{\sigma_d}, t) - \boldsymbol{F}_{\theta^-}(\frac{\boldsymbol{x}_t}{\sigma_d}, t) - \boldsymbol{g}\|_2^2 - w_\phi(t)$   ▷ Adaptive weighting

    $(\theta, \phi) \leftarrow (\theta, \phi) - \eta\nabla_{\theta,\phi}\mathcal{L}(\theta, \phi)$

    Iters $\leftarrow$ Iters $+ 1$
**until** convergence

---

# B  TRIGFLOW: A SIMPLE FRAMEWORK UNIFYING EDM, FLOW MATCHING AND VELOCITY PREDICTION

## B.1  DERIVATIONS

Denote the standard deviation of the data distribution $p_d$ as $\sigma_d$. We consider a general forward diffusion process at time $t \in [0, T]$ with $\boldsymbol{x}_t = \alpha_t \boldsymbol{x}_0 + \sigma_t \boldsymbol{z}$ for the data sample $\boldsymbol{x}_0 \sim p_d$ and the noise sample $\boldsymbol{z} \sim \mathcal{N}(\boldsymbol{0}, \sigma_d^2 \boldsymbol{I})$ (note that the variance of $\boldsymbol{z}$ is the same as that of the data $\boldsymbol{x}_0$)[1], where $\alpha_t > 0, \sigma_t > 0$ are *noise schedules* such that $\alpha_t/\sigma_t$ is monotonically decreasing w.r.t. $t$, with $\alpha_0 = 1, \sigma_0 = 0$. The general training loss for diffusion model can always be rewritten as

$$\mathcal{L}_{\text{Diff}}(\theta) = \mathbb{E}_{\boldsymbol{x}_0, \boldsymbol{z}, t}\left[w(t)\|\boldsymbol{D}_\theta(\boldsymbol{x}_t, t) - \boldsymbol{x}_0\|_2^2\right], \tag{9}$$

where different diffusion model formulation contains four different parts:

1. Parameterization of $\boldsymbol{D}_\theta$, such as score function (Song & Ermon, 2019; Song et al., 2021b), noise prediction model (Song & Ermon, 2019; Song et al., 2021b; Ho et al., 2020), data prediction model (Ho et al., 2020; Kingma et al., 2021; Salimans & Ho, 2022), velocity prediction model (Salimans & Ho, 2022), EDM (Karras et al., 2022) and flow matching (Lipman et al., 2022; Liu et al., 2022; Albergo et al., 2023).

2. Noise schedule for $\alpha_t$ and $\sigma_t$, such as variance preserving process (Ho et al., 2020; Song et al., 2021b), variance exploding process (Song et al., 2021b; Karras et al., 2022), cosine schedule (Nichol & Dhariwal, 2021), and conditional optimal transport path (Lipman et al., 2022).

3. Weighting function for $w(t)$, such as uniform weighting (Ho et al., 2020; Nichol & Dhariwal, 2021; Karras et al., 2022), weighting by functions of signal-to-noise-ratio (SNR) (Salimans & Ho, 2022), monotonic weighting (Kingma & Gao, 2024) and adaptive weighting (Karras et al., 2024).

4. Proposal distribution for $t$, such as uniform distribution within $[0, T]$ (Ho et al., 2020; Song et al., 2021b), log-normal distribution (Karras et al., 2022), SNR sampler (Esser et al., 2024), and adaptive importance sampler (Song et al., 2021a; Kingma et al., 2021).

Below we show that, under the *unit variance principle* proposed in EDM (Karras et al., 2022), we can obtain a general but simple framework for all the above four parts, which can equivalently reproduce all previous diffusion models.

---

[1]For any diffusion process with $\boldsymbol{x}_t = \alpha_t'\boldsymbol{x}_0 + \sigma_t'\boldsymbol{\epsilon}$ where $\boldsymbol{\epsilon} \sim \mathcal{N}(\boldsymbol{0}, \boldsymbol{I})$, we can always equivalently convert it to $\boldsymbol{x}_t = \alpha_t'\boldsymbol{x}_0 + \frac{\sigma_t'}{\sigma_d} \cdot (\sigma_d \boldsymbol{\epsilon})$ and let $\boldsymbol{z} := \sigma_d \boldsymbol{\epsilon}, \alpha_t := \alpha_t', \sigma_t := \frac{\sigma_t'}{\sigma_d}$. So the assumption for $\boldsymbol{z} \sim \mathcal{N}(\boldsymbol{0}, \sigma_d^2 \boldsymbol{I})$ does not result in any loss of generality.

**Step 1: General EDM parameterization.** We consider the parameterization for $\boldsymbol{D}_\theta$ as the same principle in EDM (Karras et al., 2022) by

$$\boldsymbol{D}_\theta(\boldsymbol{x}_t, t) = c_{\text{skip}}(t)\boldsymbol{x}_t + c_{\text{out}}(t)\boldsymbol{F}_\theta(c_{\text{in}}(t)\boldsymbol{x}_t, c_{\text{noise}}(t)), \tag{10}$$

and thus the training objective becomes

$$\mathcal{L}_{\text{Diff}} = \mathbb{E}_{\boldsymbol{x}_0, \boldsymbol{z}, t} \left[ w(t)c_{\text{out}}^2(t) \left\| \boldsymbol{F}_\theta(c_{\text{in}}(t)\boldsymbol{x}_t, c_{\text{noise}}(t)) - \frac{(1 - c_{\text{skip}}(t)\alpha_t)\boldsymbol{x}_0 - c_{\text{skip}}(t)\sigma_t \boldsymbol{z}}{c_{\text{out}}(t)} \right\|_2^2 \right]. \tag{11}$$

To ensure the input data of $\boldsymbol{F}_\theta$ has unit variance, we should ensure $\text{Var}[c_{\text{in}}(t)\boldsymbol{x}_t] = 1$ by letting

$$c_{\text{in}}(t) = \frac{1}{\sigma_d \sqrt{\alpha_t^2 + \sigma_t^2}}. \tag{12}$$

To ensure the training target of $\boldsymbol{F}_\theta$ has unit variance, we have

$$c_{\text{out}}^2(t) = \sigma_d^2(1 - c_{\text{skip}}(t)\alpha_t)^2 + \sigma_d^2 c_{\text{skip}}^2(t)\sigma_t^2. \tag{13}$$

To reduce the error amplification from $\boldsymbol{F}_\theta$ to $\boldsymbol{D}_\theta$, we should ensure $c_{\text{out}}(t)$ to be as small as possible, which means we should take $c_{\text{skip}}(t)$ by letting $\frac{\partial c_{\text{out}}}{\partial c_{\text{skip}}} = 0$, which results in

$$c_{\text{skip}}(t) = \frac{\alpha_t}{\alpha_t^2 + \sigma_t^2}, \quad c_{\text{out}}(t) = \pm \frac{\sigma_d \sigma_t}{\sqrt{\alpha_t^2 + \sigma_t^2}}. \tag{14}$$

Though equivalent, we choose $c_{\text{out}}(t) = -\frac{\sigma_d \sigma_t}{\sqrt{\alpha_t^2 + \sigma_t^2}}$ which can simplify some derivations below.

In summary, the parameterization and objective for the general diffusion noise schedule are

$$\boldsymbol{D}_\theta(\boldsymbol{x}_t, t) = \frac{\alpha_t}{\alpha_t^2 + \sigma_t^2}\boldsymbol{x}_t - \frac{\sigma_t}{\sqrt{\alpha_t^2 + \sigma_t^2}}\sigma_d \boldsymbol{F}_\theta\left(\frac{\boldsymbol{x}_t}{\sigma_d \sqrt{\alpha_t^2 + \sigma_t^2}}, c_{\text{noise}}(t)\right), \tag{15}$$

$$\mathcal{L}_{\text{Diff}} = \mathbb{E}_{\boldsymbol{x}_0, \boldsymbol{z}, t} \left[ w(t)\frac{\sigma_t^2}{\alpha_t^2 + \sigma_t^2} \left\| \sigma_d \boldsymbol{F}_\theta\left(\frac{\boldsymbol{x}_t}{\sigma_d \sqrt{\alpha_t^2 + \sigma_t^2}}, c_{\text{noise}}(t)\right) - \frac{\alpha_t \boldsymbol{z} - \sigma_t \boldsymbol{x}_0}{\sqrt{\alpha_t^2 + \sigma_t^2}} \right\|_2^2 \right]. \tag{16}$$

**Step 2: All noise schedules can be equivalently transformed.** One nice property of the *unit variance principle* is that the $\alpha_t, \sigma_t$ in both the parameterization and the objective are *homogenous*, which means we can always assume $\alpha_t^2 + \sigma_t^2 = 1$ without loss of generality. To see this, we can apply a simple change-of-variable of $\hat{\alpha}_t = \frac{\alpha_t}{\sqrt{\alpha_t^2 + \sigma_t^2}}$, $\hat{\sigma}_t = \frac{\sigma_t}{\sqrt{\alpha_t^2 + \sigma_t^2}}$ and $\hat{\boldsymbol{x}}_t = \frac{\boldsymbol{x}_t}{\sqrt{\alpha_t^2 + \sigma_t^2}} = \hat{\alpha}_t \boldsymbol{x}_0 + \hat{\sigma}_t \boldsymbol{z}$, thus we have

$$\boldsymbol{D}_\theta(\boldsymbol{x}_t, t) = \hat{\alpha}_t \hat{\boldsymbol{x}}_t - \hat{\sigma}_t \sigma_d \boldsymbol{F}_\theta\left(\frac{\hat{\boldsymbol{x}}_t}{\sigma_d}, c_{\text{noise}}(t)\right), \tag{17}$$

$$\mathcal{L}_{\text{Diff}} = \mathbb{E}_{\boldsymbol{x}_0, \boldsymbol{z}, t} \left[ w(t)\hat{\sigma}_t^2 \left\| \sigma_d \boldsymbol{F}_\theta\left(\frac{\hat{\boldsymbol{x}}_t}{\sigma_d}, c_{\text{noise}}(t)\right) - (\hat{\alpha}_t \boldsymbol{z} - \hat{\sigma}_t \boldsymbol{x}_0) \right\|_2^2 \right]. \tag{18}$$

As for the sampling procedure, according to DPM-Solver++ (Lu et al., 2022b), the exact solution of diffusion ODE from time $s$ to time $t$ satisfies

$$\boldsymbol{x}_t = \frac{\sigma_t}{\sigma_s}\boldsymbol{x}_s + \sigma_t \int_{\lambda_s}^{\lambda_t} e^\lambda \boldsymbol{D}_\theta(\boldsymbol{x}_\lambda, \lambda)\mathrm{d}\lambda, \tag{19}$$

where $\lambda_t = \log \frac{\alpha_t}{\sigma_t}$, so the sampling procedure is also *homogenous* for $\alpha_t, \sigma_t$. To see this, we can use the fact that $\frac{\boldsymbol{x}_t}{\sigma_t} = \frac{\hat{\boldsymbol{x}}_t}{\hat{\sigma}_t}$ and $\lambda_t = \log \frac{\hat{\alpha}_t}{\hat{\sigma}_t} := \hat{\lambda}_t$, thus the above equation is equivalent to

$$\hat{\boldsymbol{x}}_t = \frac{\hat{\sigma}_t}{\hat{\sigma}_s}\hat{\boldsymbol{x}}_s + \hat{\sigma}_t \int_{\hat{\lambda}_s}^{\hat{\lambda}_t} e^{\hat{\lambda}} \hat{\boldsymbol{D}}_\theta(\hat{\boldsymbol{x}}_{\hat{\lambda}}, \hat{\lambda})\mathrm{d}\hat{\lambda}, \tag{20}$$

which is exactly the sampling procedure of the diffusion process $\hat{\boldsymbol{x}}_t$, which means **noise schedules of diffusion models won't affect the performance of sampling**. In other words, for any diffusion

process $(\alpha_t, \sigma_t, \boldsymbol{x}_t)$ at time $t$, we can always divide them by $\sqrt{\alpha_t^2 + \sigma_t^2}$ to obtain the diffusion process $(\hat{\alpha}_t, \hat{\sigma}_t, \hat{\boldsymbol{x}}_t)$ with $\hat{\alpha}_t^2 + \hat{\sigma}_t^2 = 1$ and all the parameterization, training objective and sampling procedure can be equivalently transformed. The only difference is the corresponding training weighting $w(t)\sigma_d^2\hat{\sigma}_t^2$ in Eq. (18), which we will discuss in the next step.

A straightforward corollary is that the "optimal transport path" (Lipman et al., 2022) in flow matching with $\alpha_t = 1 - t, \sigma_t = t$ can be equivalently converted to other noise schedules. The reason of its better empirical performance is essentially due to the different weighting during training and the lack of advanced diffusion sampler such as DPM-Solver series (Lu et al., 2022a;b) during sampling, not the "straight path" (Lipman et al., 2022) itself. By converting the diffusion process to the space satisfying $\sqrt{\hat{\alpha}_t^2 + \hat{\sigma}_t^2} = 1$, the path connection $\boldsymbol{x}_0$ and $\boldsymbol{z}$ has consistent variance which matches the unit-variance design principles of EDM.

**Step 3: Unified framework by TrigFlow.** As we showed in the previous step, we can always assume $\hat{\alpha}_t^2 + \hat{\sigma}_t^2 = 1$. An equivalent change-of-variable of such constraint is to define

$$\hat{t} := \arctan\left(\frac{\hat{\sigma}_t}{\hat{\alpha}_t}\right) = \arctan\left(\frac{\sigma_t}{\alpha_t}\right), \tag{21}$$

so $\hat{t} \in [0, \frac{\pi}{2}]$ is a monotonically increasing function of $t \in [0, T]$, thus there exists a one-one mapping between $t$ and $\hat{t}$ to convert the proposal distribution $p(t)$ to the distribution of $\hat{t}$, denoted as $p\left(\hat{t}\right)$. As $\hat{\alpha}_t = \cos\left(\hat{t}\right), \hat{\sigma}_t = \sin\left(\hat{t}\right)$, the training objective in Eq. (18) is equivalent to

$$\mathcal{L}_{\text{Diff}} = \mathbb{E}_{\boldsymbol{x}_0, \boldsymbol{z}}\left[\int_0^{\frac{\pi}{2}} \underbrace{p\left(\hat{t}\right) w\left(\hat{t}\right) \sin^2\left(\hat{t}\right)}_{\text{training weighting}} \left\|\underbrace{\sigma_d \boldsymbol{F}_\theta\left(\frac{\hat{\boldsymbol{x}}_{\hat{t}}}{\sigma_d}, c_{\text{noise}}\left(\hat{t}\right)\right) - \left(\cos\left(\hat{t}\right) \boldsymbol{z} - \sin\left(\hat{t}\right) \boldsymbol{x}_0\right)}_{\text{independent from } \alpha_t \text{ and } \sigma_t}\right\|_2^2 d\hat{t}\right]. \tag{22}$$

Therefore, we can always put the influence of different noise schedules into the training weighting of the integral for $\hat{t}$ from $0$ to $\frac{\pi}{2}$, while the $\ell_2$ norm loss at each $\hat{t}$ is independent from the choices of $\alpha_t$ and $\sigma_t$. As we equivalently convert the noise schedules by trigonometric functions, we name such framework for diffusion models as *TrigFlow*.

For simplicity and with a slight abuse of notation, we omit the $\hat{t}$ and denote the whole training weighting as a single $w(t)$, we summarize the diffusion process, parameterization, training objective and samplers of TrigFlow as follows.

**Diffusion Process.** $\boldsymbol{x}_0 \sim p_d(\boldsymbol{x}_0), \boldsymbol{z} \sim \mathcal{N}(\boldsymbol{0}, \sigma_d^2\boldsymbol{I}), \boldsymbol{x}_t = \cos(t)\boldsymbol{x}_0 + \sin(t)\boldsymbol{z}$ for $t \in [0, \frac{\pi}{2}]$.

**Parameterization.**

$$\boldsymbol{D}_\theta(\boldsymbol{x}_t, t) = \cos(t)\boldsymbol{x}_t - \sin(t)\sigma_d\boldsymbol{F}_\theta\left(\frac{\boldsymbol{x}_t}{\sigma_d}, c_{\text{noise}}(t)\right), \tag{23}$$

where $c_{\text{noise}}(t)$ is the conditioning input of the noise levels for $\boldsymbol{F}_\theta$, which can be arbitrary one-one mapping of $t$. Moreover, the parameterized diffusion ODE is defined by

$$\frac{d\boldsymbol{x}_t}{dt} = \sigma_d\boldsymbol{F}_\theta\left(\frac{\boldsymbol{x}_t}{\sigma_d}, c_{\text{noise}}(t)\right). \tag{24}$$

**Training Objective.**

$$\mathcal{L}_{\text{Diff}}(\theta) = \mathbb{E}_{\boldsymbol{x}_0, \boldsymbol{z}}\left[\int_0^{\frac{\pi}{2}} w(t) \left\|\sigma_d\boldsymbol{F}_\theta\left(\frac{\boldsymbol{x}_t}{\sigma_d}, c_{\text{noise}}\left(t\right)\right) - \left(\cos(t)\boldsymbol{z} - \sin(t)\boldsymbol{x}_0\right)\right\|_2^2 dt\right], \tag{25}$$

where $w(t)$ is the training weighting, which we will discuss in details in Appendix D.

As for the sampling procedure, although we can directly solve the diffusion ODE in Eq. (24) by Euler's or Heun's solvers as in flow matching (Lipman et al., 2022), the parameterization for $\sigma_d\boldsymbol{F}_\theta$

may be not the optimal parameterization for reducing the discreteization errors. As proved in DPM-Solver-v3 (Zheng et al., 2023c), the optimal parameterization should cancel all the linearity of the ODE, and the data prediction model $D_\theta$ is an effective approximation of such parameterization. Thus, we can also apply DDIM, DPM-Solver and DPM-Solver++ for TrigFlow by rewriting the coefficients into the TrigFlow notation, as listed below.

**1st-order Sampler by DDIM.** Starting from $x_s$ at time $s$, the solution $x_t$ at time $t$ is

$$x_t = \cos(s-t)x_t - \sin(s-t)\sigma_d F_\theta\left(\frac{x_s}{\sigma_d}, c_{\text{noise}}(s)\right) \tag{26}$$

One good property of TrigFlow is that the 1st-order sampler can naturally support zero-SNR sampling (Lin et al., 2024) by letting $s = \frac{\pi}{2}$ without any numerical issues.

**2nd-order Sampler by DPM-Solver.** Starting from $x_s$ at time $s$, by reusing a previous solution $x_{s'}$ at time $s'$, the solution $x_t$ at time $t$ is

$$x_t = \cos(s-t)x_s - \sin(s-t)\sigma_d F_\theta\left(\frac{x_s}{\sigma_d}, c_{\text{noise}}(s)\right) - \frac{\sin(s-t)}{2r_s\cos(s)}\left(\epsilon_\theta(x_{s'}, s') - \epsilon_\theta(x_s, s)\right), \tag{27}$$

where $\epsilon_\theta(x_t, t) = \sin(t)x_t + \cos(t)\sigma_d F_\theta\left(\frac{x_t}{\sigma_d}, c_{\text{noise}}(t)\right)$ is the noise prediction model, and $r_s = \frac{\log\tan(s)-\log\tan(s')}{\log\tan(s)-\log\tan(t)}$.

**2nd-order Sampler by DPM-Solver++.** Starting from $x_s$ at time $s$, by reusing a previous solution $x_{s'}$ at time $s'$, the solution $x_t$ at time $t$ is

$$x_t = \cos(s-t)x_s - \sin(s-t)\sigma_d F_\theta\left(\frac{x_s}{\sigma_d}, c_{\text{noise}}(s)\right) + \frac{\sin(s-t)}{2r_s\sin(s)}\left(D_\theta(x_{s'}, s') - D(x_s, s)\right), \tag{28}$$

where $r_s = \frac{\log\tan(s)-\log\tan(s')}{\log\tan(s)-\log\tan(t)}$.

## B.2 Relationship with other parameterization

As previous diffusion models define the forward process with $x_{t'} = \alpha_{t'}x_0 + \sigma_{t'}\epsilon = \alpha_{t'}x_0 + \frac{\sigma_{t'}}{\sigma_d}(\sigma_d\epsilon)$ for $\epsilon \sim \mathcal{N}(0, I)$, we can obtain the relationship between $t'$ and TrigFlow time steps $t \in [0, \frac{\pi}{2}]$ by

$$t = \arctan\left(\frac{\sigma_{t'}}{\sigma_d\alpha_{t'}}\right), \quad x_t = \frac{\sigma_d}{\sqrt{\alpha_{t'}^2\sigma_d^2 + \sigma_{t'}^2}}x_{t'}. \tag{29}$$

Thus, we can always translate the notation from previous noise schedules to TrigFlow notations. Moreover, below we show that TrigFlow unifies different current frameworks for training diffusion models, including EDM, flow matching and velocity prediction.

**EDM.** As our derivations closely follow the *unit variance principle* proposed in EDM (Karras et al., 2022), our parameterization can be equivalently converted to EDM notations. Specifically, the transformation between TrigFlow $(x_t, t)$ and EDM $(x_\sigma, \sigma)$ is

$$t = \arctan\left(\frac{\sigma}{\sigma_d}\right), \quad x_t = \cos(t)x_\sigma. \tag{30}$$

The reason why TrigFlow notation is much simpler than EDM is just because we define the end point of the diffusion process as $z \sim \mathcal{N}(0, \sigma_d^2 I)$ with the same variance as the data distribution. Thus, the *unit variance principle* can ensure that all the intermediate $x_t$ does not need to multiply other coefficients as in EDM.

**Flow Matching.** Flow matching (Lipman et al., 2022; Liu et al., 2022; Albergo et al., 2023; Kornilov et al., 2024) defines a stochastic path between two samples $x_0$ from data distribution and $z$ from a tractable distribution which is usually some Gaussian distribution. For a general path $x_t = \alpha_t x_0 + \sigma_t z$ with $\alpha_0 = 1, \alpha_T = 0, \sigma_0 = 0, \sigma_T = 1$, the conditional probability path is

$$v_t = \frac{d\alpha_t}{dt}x_0 + \frac{d\sigma_t}{dt}z, \tag{31}$$

and it learns a parameterized model $\boldsymbol{v}_\theta(\boldsymbol{x}_t, t)$ by minimizing

$$\mathbb{E}_{\boldsymbol{x}_0, \boldsymbol{z}, t} \left[ w(t) \left\| \boldsymbol{v}_\theta(\boldsymbol{x}_t, t) - \boldsymbol{v}_t \right\|_2^2 \right], \tag{32}$$

and the final probability flow ODE is defined by

$$\frac{\mathrm{d}\boldsymbol{x}_t}{\mathrm{d}t} = \boldsymbol{v}_\theta(\boldsymbol{x}_t, t). \tag{33}$$

As TrigFlow uses $\alpha_t = \cos(t)$ and $\sigma_t = \sin(t)$, it is easy to see that the training objective and the diffusion ODE of TrigFlow are also the same as flow matching with $\boldsymbol{v}_\theta(\boldsymbol{x}_t, t) = \sigma_d \boldsymbol{F}_\theta(\frac{\boldsymbol{x}_t}{\sigma_d}, c_{\text{noise}}(t))$. To the best of our knowledge, TrigFlow is the first framework that unifies EDM and flow matching for training diffusion models.

**Velocity Prediction.** The velocity prediction parameterization (Salimans & Ho, 2022) trains a parameterization network with the target $\alpha_t \boldsymbol{z} - \sigma_t \boldsymbol{x}_0$. As TrigFlow uses $\alpha_t = \cos(t), \sigma_t = \sin(t)$, it is easy to see that the training target in TrigFlow is also the velocity.

**Discussions on SNR.** Another good property of TrigFlow is that it can define a data-variance-invariant SNR. Specifically, previous diffusion models define the SNR at time $t$ as $\text{SNR}(t) = \frac{\alpha_t^2}{\sigma_t^2}$ for $\boldsymbol{x}_t = \alpha_t \boldsymbol{x}_0 + \sigma_t \boldsymbol{\epsilon}$ with $\boldsymbol{\epsilon} \sim \mathcal{N}(\boldsymbol{0}, \boldsymbol{I})$. However, such definition ignores the influence of the variance of $\boldsymbol{x}_0$: if we rescale the data $\boldsymbol{x}_0$ by a constant, then such SNR doesn't get rescaled correspondingly, which is not reasonable in practice. Instead, in TrigFlow we can define the SNR by

$$\hat{\text{SNR}}(t) = \frac{\alpha_t^2 \sigma_d^2}{\sigma_t^2} = \frac{1}{\tan^2(t)}, \tag{34}$$

which is data-variance-invariant and also simple.

## C ADAPTIVE VARIATIONAL SCORE DISTILLATION IN TRIGFLOW FRAMEWORK

In this section, we propose the detailed derivation for variational score distillation (VSD) in TrigFlow framework and an improved objective with adaptive weighting.

### C.1 DERIVATIONS

Assume we have samples $\boldsymbol{x}_0 \in \mathbb{R}^D$ from data distribution $p_d$ with standard deviation $\sigma_d$, and define a corresponding forward diffusion process $\{p_t\}_{t=0}^T$ starting at $p_0 = p_d$ and ending at $p_T \approx \mathcal{N}(\boldsymbol{0}, \hat{\sigma}^2 \boldsymbol{I})$, with $p_{t0}(\boldsymbol{x}_t | \boldsymbol{x}_0) = \mathcal{N}(\boldsymbol{x}_t | \alpha_t \boldsymbol{x}_0, \sigma_t^2 \boldsymbol{I})$. Variational score distillation (VSD) (Wang et al., 2024; Yin et al., 2024b;a) trains a generator $\boldsymbol{g}_\theta : \mathbb{R}^D \to \mathbb{R}^D$ aiming to map noise samples $\boldsymbol{z} \sim \mathcal{N}(\boldsymbol{0}, \hat{\sigma}^2 \boldsymbol{I})$ to the data distribution, by minimizing

$$\min_\theta \mathbb{E}_t \left[ w(t) D_{\text{KL}} \left( q_t^\theta \parallel p_t \right) \right] = \mathbb{E}_{t, \boldsymbol{z}, \boldsymbol{\epsilon}} \left[ w(t) \left( \log q_t^\theta(\alpha_t \boldsymbol{g}_\theta(\boldsymbol{z}) + \sigma_t \boldsymbol{\epsilon}) - \log p_t(\alpha_t \boldsymbol{g}_\theta(\boldsymbol{z}) + \sigma_t \boldsymbol{\epsilon}) \right) \right],$$

where $\boldsymbol{\epsilon} \sim \mathcal{N}(\boldsymbol{0}, \boldsymbol{I})$, $q_t^\theta$ is the diffused distribution at time $t$ with the same forward diffusion process as $p_t$ while starting at $q_0^\theta$ as the distribution of $\boldsymbol{g}_\theta(\boldsymbol{z})$, $w(t)$ is an ad-hoc training weighting (Poole et al., 2022; Wang et al., 2024; Yin et al., 2024b), and $t$ follows a proposal distribution such as uniform distribution. It is proved that the optimum of $q_t^\theta$ satisfies $q_0 = p_d$ (Wang et al., 2024) and thus the distribution of the generator matches the data distribution.

Moreover, by denoting $x_t^\theta := \alpha_t g_\theta(z) + \sigma_t \epsilon$ and taking the gradient w.r.t. $\theta$, we have

$$\nabla_\theta \mathbb{E}_t \left[ w(t) D_{\mathrm{KL}} \left( q_t^\theta \parallel p_t \right) \right]$$

$$= \mathbb{E}_{t,z,\epsilon} \left[ w(t) \nabla_\theta \left( \log q_t^\theta(x_t^\theta) - \log p_t(x_t^\theta) \right) \right]$$

$$= \mathbb{E}_{t,z,\epsilon} \left[ w(t) \left( \partial_\theta \log q_t^\theta(x_t) + \left( \nabla_{x_t} \log q_t^\theta(x_t) - \nabla_{x_t} \log p_t(x_t) \right) \frac{\partial x_t^\theta}{\partial \theta} \right) \right]$$

$$= \underbrace{\mathbb{E}_{t,x_t} \left[ w(t) \partial_\theta \log q_t^\theta(x_t) \right]}_{=0} + \mathbb{E}_{t,z,\epsilon} \left[ w(t) \left( \nabla_{x_t} \log q_t^\theta(x_t) - \nabla_{x_t} \log p_t(x_t) \right) \frac{\alpha_t \partial g_\theta(z)}{\partial \theta} \right]$$

$$= \mathbb{E}_{t,z,\epsilon} \left[ \alpha_t w(t) \left( \nabla_{x_t} \log q_t^\theta(x_t) - \nabla_{x_t} \log p_t(x_t) \right) \frac{\partial g_\theta(z)}{\partial \theta} \right].$$

Therefore, we need to approximate the score functions $\nabla_{x_t} \log q_t^\theta(x_t)$ for the generator and $\nabla_{x_t} \log p_t(x_t)$ for the data distribution. VSD trains a diffusion model for samples from $g_\theta(z)$ to approximate $\nabla_{x_t} \log q_t^\theta(x_t)$ and uses a pretrained diffusion model to approximate $\nabla_{x_t} \log p_t(x_t)$.

In this work, we train the diffusion model in TrigFlow framework, with $\alpha_t = \cos(t)$, $\sigma_t = \sigma_d \sin(t)$, $\hat{\sigma} = \sigma_d$, $T = \frac{\pi}{2}$. Specifically, assume we have a pretrained diffusion model $F_{\mathrm{pretrain}}$ parameterized by TrigFlow, and we train another diffusion model $F_\phi$ to approximate the diffused generator distribution, by

$$\min_\phi \mathbb{E}_{z,z',t} \left[ w(t) \left\| \sigma_d F_\phi \left( \frac{x_t}{\sigma_d}, t \right) - v_t \right\|_2^2 \right],$$

where $x_t = \cos(t) x_0 + \sin(t) z$, $v_t = \cos(t) z - \sin(t) x_0$, $z \sim \mathcal{N}(0, \sigma_d^2 I)$, $x_0 = g_\theta(z')$ with $z' \sim \mathcal{N}(0, \sigma_d^2 I)$. Moreover, the relationship between the ground truth diffusion model $F_{\mathrm{Diff}}(x_t, t)$ and the score function $\nabla_{x_t} \log p_t(x_t)$ is

$$\sigma_d F_{\mathrm{Diff}}(x_t, t) = \mathbb{E}[v_t | x_t] = \frac{1}{\tan(t)} x_t - \frac{1}{\sin(t)} \mathbb{E}_{x_0 | x_t} [x_0],$$

$$\nabla_{x_t} \log p_t(x_t) = \mathbb{E}_{x_0 | x_t} \left[ -\frac{x_t - \cos(t) x_0}{\sigma_d^2 \sin^2(t)} \right] = -\frac{\cos(t) \sigma_d F_{\mathrm{Diff}} + \sin(t) x_t}{\sigma_d^2 \sin(t)}.$$

Thus, we train the generator $g_\theta$ by the following gradient w.r.t. $\theta$:

$$\mathbb{E}_{t,z,z'} \left[ \frac{\cos^2(t)}{\sigma_d \sin(t)} w(t) \left( F_{\mathrm{pretrain}} \left( \frac{x_t}{\sigma_d}, t \right) - F_\phi \left( \frac{x_t}{\sigma_d}, t \right) \right) \frac{\partial g_\theta(z')}{\partial \theta} \right],$$

which is equivalent to the gradient of the following objective:

$$\mathbb{E}_{t,z,z'} \left[ \frac{\cos^2(t)}{\sigma_d \sin(t)} w(t) \left\| g_\theta(z') - g_{\theta^-}(z') + F_{\mathrm{pretrain}} \left( \frac{x_t}{\sigma_d}, t \right) - F_\phi \left( \frac{x_t}{\sigma_d}, t \right) \right\|_2^2 \right],$$

where $g_{\theta^-}(z')$ is the same as $g_\theta(z')$ but stops the gradient for $\theta$. Note that the weighting functions used in previous works (Wang et al., 2024; Yin et al., 2024b) is proportional to $\frac{\sin^2(t)}{\cos(t)}$, thus the prior weighting is proportional to $\sin(t) \cos(t)$, which has a U-shape similar to the log-normal distribution used in Karras et al. (2022). Thus, we can instead use a log-normal proposal distribution and apply the adaptive weighting by training another weighting network $w_\psi(t)$. We refer to Appendix D for detailed discussions about the learnable adaptive weighting. Thus we can obtain the training objective, as listed below.

## C.2 TRAINING OBJECTIVE

**Training Objective of Adaptive Variational Score Distillation (aVSD).**

$$\min_\phi \mathcal{L}_{\mathrm{Diff}}(\phi) := \mathbb{E}_{z,z',t} \left[ w(t) \left\| \sigma_d F_\phi \left( \frac{x_t}{\sigma_d}, t \right) - v_t \right\|_2^2 \right],$$

$$\min_{\theta,\psi} \mathcal{L}_{\mathrm{VSD}}(\theta, \psi) := \mathbb{E}_{t,z,z'} \left[ \frac{e^{w_\psi(t)}}{D} \left\| g_\theta(z') - g_{\theta^-}(z') + F_{\mathrm{pretrain}} \left( \frac{x_t}{\sigma_d}, t \right) - F_\phi \left( \frac{x_t}{\sigma_d}, t \right) \right\|_2^2 - w_\psi(t) \right].$$

And we also choose a proportional distribution of $t$ for estimating $\mathcal{L}_{\text{VSD}}(\theta, \psi)$ by $\log(\tan(t)\sigma_d) \sim \mathcal{N}(P_{\text{mean}}, P_{\text{std}}^2)$ and tune these two hyperparameters (note that they may be different from the proposal distribution for training $\mathcal{L}_{\text{Diff}}(\phi)$, as detailed in Appendix G.

In addition, for consistency models $\boldsymbol{f}_\theta(\boldsymbol{x}_t, t)$, we choose $\boldsymbol{z}' \sim \mathcal{N}(\boldsymbol{0}, \sigma_d^2 \boldsymbol{I})$ and $\boldsymbol{g}_\theta(\boldsymbol{z}') := \boldsymbol{f}_\theta(\boldsymbol{z}', \frac{\pi}{2}) = -\sigma_d \boldsymbol{F}_\theta(\frac{\boldsymbol{z}'}{\sigma_d}, \frac{\pi}{2})$, and thus the corresponding objective is

$$\min_{\theta, \psi} \mathbb{E}_{t, \boldsymbol{z}, \boldsymbol{z}'} \left[ \frac{e^{w_\psi(t)}}{D} \left\| \sigma_d \boldsymbol{F}_\theta \left( \frac{\boldsymbol{z}'}{\sigma_d}, \frac{\pi}{2} \right) - \sigma_d \boldsymbol{F}_{\theta^-} \left( \frac{\boldsymbol{z}'}{\sigma_d}, \frac{\pi}{2} \right) - \boldsymbol{F}_{\text{pretrain}} \left( \frac{\boldsymbol{x}_t}{\sigma_d}, t \right) + \boldsymbol{F}_\phi \left( \frac{\boldsymbol{x}_t}{\sigma_d}, t \right) \right\|_2^2 - w_\psi(t) \right] .$$

## D    ADAPTIVE WEIGHTING FOR DIFFUSION MODELS, CONSISTENCY MODELS AND VARIATIONAL SCORE DISTILLATION

We first list the objectives of diffusion models, consistency models and variational score distillation (VSD). For diffusion models, as shown in Eq. (25), the gradient of the objective is

$$\nabla_\theta \mathcal{L}_{\text{Diff}}(\theta) = \nabla_\theta \mathbb{E}_{t, \boldsymbol{x}_0, \boldsymbol{z}} w(t) \left[ \| \sigma_d \boldsymbol{F}_\theta - \boldsymbol{v}_t \|_2^2 \right] = \nabla_\theta \mathbb{E}_{t, \boldsymbol{x}_0, \boldsymbol{z}} \left[ w(t) \sigma_d \boldsymbol{F}_\theta^\top (\sigma_d \boldsymbol{F}_{\theta^-} - \boldsymbol{v}_t) \right],$$

where $\boldsymbol{F}_{\theta^-}$ is the same as $\boldsymbol{F}_\theta$ but stops the gradient w.r.t. $\theta$. For VSD, the gradient of the objective is

$$\nabla_\theta \mathcal{L}_{\text{Diff}}(\theta) = \nabla_\theta \mathbb{E}_{t, \boldsymbol{z}, \boldsymbol{z}'} \left[ w(t) g_\theta(\boldsymbol{z}')^\top (\boldsymbol{F}_{\text{pretrain}} - \boldsymbol{F}_\phi) \right].$$

And for continuous-time CMs parameterized by TrigFlow, the objective is

$$\nabla_\theta \mathcal{L}_{\text{CM}}(\theta) = \nabla_\theta \mathbb{E}_{t, \boldsymbol{x}_0, \boldsymbol{z}} \left[ -w(t) \sin(t) \boldsymbol{f}_\theta^\top \frac{\mathrm{d}\boldsymbol{f}_{\theta^-}}{\mathrm{d}t} \right],$$

where $\boldsymbol{f}_{\theta^-}$ is the same as $\boldsymbol{f}_\theta$ but stops the gradient w.r.t. $\theta$. Interestingly, all these objectives can be rewritten into a form of inner product between a neural network and a target function which has the same dimension (denoted as $D$) as the output of the neural network. Specifically, assume the neural network is $\boldsymbol{F}_\theta$ parameterized by $\theta$, we study the following objective:

$$\min_\theta \mathbb{E}_t \left[ \boldsymbol{F}_\theta^\top \boldsymbol{y} \right],$$

where we do not compute the gradients w.r.t. $\theta$ for $\boldsymbol{y}$. In such case, the gradient will be equivalent to

$$\nabla_\theta \mathbb{E}_t \left[ \| \boldsymbol{F}_\theta - \boldsymbol{F}_{\theta^-} + \boldsymbol{y} \|_2^2 \right],$$

where $\boldsymbol{F}_{\theta^-}$ is the same as $\boldsymbol{F}_\theta$ but stops the gradient w.r.t. $\theta$. In such case, we can balance the gradient variance w.r.t. $t$ by training an adaptive weighting network $w_\phi(t)$ to estimate the loss norm, i.e., minimizing

$$\min_\phi \mathbb{E}_t \left[ \frac{e^{w_\phi(t)}}{D} \| \boldsymbol{F}_\theta - \boldsymbol{F}_{\theta^-} + \boldsymbol{y} \|_2^2 - w_\phi(t) \right].$$

This is the *adaptive weighting* proposed by EDM2 (Karras et al., 2024), which balances the loss variance across different time steps, inspired by the uncertainty estimation of Sener & Koltun (2018). By taking the partial derivative w.r.t. $w$ in the above equation, it is easy to verify that the optimal $w^*(t)$ satisfies

$$\frac{e^{w^*(t)}}{D} \mathbb{E} \left[ \| \boldsymbol{F}_\theta - \boldsymbol{F}_{\theta^-} + \boldsymbol{y} \|_2^2 \right] \equiv 1.$$

Therefore, the adaptive weighting reduces the loss variance across different time steps. In such case, all we need to do is to choose

1. A prior weighting $\lambda(t)$ for $\boldsymbol{y}$, which may be helpful for further reducing the variance of $\boldsymbol{y}$. Then the objective becomes

$$\min_{\theta, \phi} \mathbb{E}_t \left[ \frac{e^{w_\phi(t)}}{D} \| \boldsymbol{F}_\theta - \boldsymbol{F}_{\theta^-} + \lambda(t) \boldsymbol{y} \|_2^2 - w_\phi(t) \right].$$

   e.g., for diffusion models and VSD, since the target is either $\boldsymbol{y} = \boldsymbol{F} - \boldsymbol{v}_t$ or $\boldsymbol{y} = \boldsymbol{F}_{\text{pretrain}} - \boldsymbol{F}_\phi$ which are stable across different time steps, we can simply choose $\lambda(t) = 1$; while for consistency models, the target $\boldsymbol{y} = \sin(t) \frac{\mathrm{d}\boldsymbol{f}}{\mathrm{d}t}$ may have huge variance, we choose $\lambda(t) = \frac{1}{\sigma_d \tan(t)}$ to reduce the variance of $\lambda(t)\boldsymbol{y}$, which empirically is critical for better performance.

2. A proposal distribution for sampling the training $t$, which determines *which part of $t$ we should focus on more*. For diffusion models, we generally need to focus on the intermediate time steps since both the clean data and pure noise cannot provide precise training signals. Thus, the common choice is to choose a normal distribution over the log-SNR of time steps, which is proposed by Karras et al. (2022) and also known as *log-normal distribution*.

In this way, we do not need to manually choose the weighting functions, significantly reducing the tuning complexity of training diffusion models, CMs and VSD.

## E   DISCRETE-TIME CONSISTENCY MODELS WITH IMPROVED TRAINING OBJECTIVES

Note that the improvements proposed in Sec. 4 can also be applied to discrete-time consistency models (CMs). In this section, we discuss the improved version of discrete-time CMs for consistency distillation.

### E.1   PARAMETERIZATION AND TRAINING OBJECTIVE

**Parameterization.**  We also parameterize the CM by TrigFlow:

$$\boldsymbol{f}_\theta(\boldsymbol{x}_t, t) = \cos(t)\boldsymbol{x}_t - \sigma_d \sin(t)\boldsymbol{F}_\theta\left(\frac{\boldsymbol{x}_t}{\sigma_d}, t\right).$$

And we denote the pretrained teacher diffusion model as $\frac{\mathrm{d}\boldsymbol{x}_t}{\mathrm{d}t} = \boldsymbol{F}_{\text{pretrain}}(\frac{\boldsymbol{x}_t}{\sigma_d}, t)$.

**Reference sample by DDIM.**  Assume we sample $\boldsymbol{x}_0 \sim p_d$, $\boldsymbol{z} \sim \mathcal{N}(\boldsymbol{0}, \sigma_d^2\boldsymbol{I})$, and $\boldsymbol{x}_t = \cos(t)\boldsymbol{x}_0 + \sin(t)\boldsymbol{z}$, we need a reference sample $\boldsymbol{x}_{t'}$ at time $t' < t$ to guide the training of the CM, which can be obtained by one-step DDIM from $t$ to $t'$:

$$\boldsymbol{x}_{t'} = \cos(t - t')\boldsymbol{x}_t - \sigma_d \sin(t - t')\boldsymbol{F}_{\text{pretrain}}\left(\frac{\boldsymbol{x}_t}{\sigma_d}, t\right).$$

Thus, the output of the consistency model at time $t'$ is

$$\boldsymbol{f}_{\theta^-}(\boldsymbol{x}_{t'}, t') = \cos(t')\cos(t-t')\boldsymbol{x}_t - \sigma_d \cos(t')\sin(t-t')\boldsymbol{F}_{\text{pretrain}}\left(\frac{\boldsymbol{x}_t}{\sigma_d}, t\right) - \sigma_d \sin(t')\boldsymbol{F}_{\theta^-}\left(\frac{\boldsymbol{x}_{t'}}{\sigma_d}, t'\right). \tag{35}$$

**Original objective of discrete-time CMs.**  The consistency model at time $t$ can be rewritten into

$$\begin{aligned}
\boldsymbol{f}_{\theta^-}(\boldsymbol{x}_t, t) &= \cos(t)\boldsymbol{x}_t - \sigma_d \sin(t)\boldsymbol{F}_{\theta^-}\left(\frac{\boldsymbol{x}_t}{\sigma_d}, t\right) \\
&= (\cos(t')\cos(t-t') - \sin(t')\sin(t-t'))\boldsymbol{x}_t \\
&\quad - \sigma_d(\sin(t-t')\cos(t') + \cos(t-t')\sin(t'))\boldsymbol{F}_{\theta^-}\left(\frac{\boldsymbol{x}_t}{\sigma_d}, t\right)
\end{aligned} \tag{36}$$

Therefore, by computing the difference between Eq. (35) and Eq. (36), we define

$$\begin{aligned}
\Delta_{\theta^-}(\boldsymbol{x}_t, t, t') &:= \frac{\boldsymbol{f}_{\theta^-}(\boldsymbol{x}_t, t) - \boldsymbol{f}_{\theta^-}(\boldsymbol{x}_{t'}, t')}{\sin(t - t')} \\
&= -\cos(t')\Big(\sigma_d\boldsymbol{F}_{\theta^-}\left(\frac{\boldsymbol{x}_t}{\sigma_d}, t\right) - \underbrace{\sigma_d\boldsymbol{F}_{\text{pretrain}}\left(\frac{\boldsymbol{x}_t}{\sigma_d}, t\right)}_{\frac{\mathrm{d}\boldsymbol{x}_t}{\mathrm{d}t}}\Big) \\
&\quad - \sin(t')\Big(\boldsymbol{x}_t + \underbrace{\frac{\sigma_d\cos(t-t')\boldsymbol{F}_{\theta^-}\left(\frac{\boldsymbol{x}_t}{\sigma_d}, t\right) - \sigma_d\boldsymbol{F}_{\theta^-}\left(\frac{\boldsymbol{x}_{t'}}{\sigma_d}, t'\right)}{\sin(t-t')}}_{\approx \sigma_d \frac{\mathrm{d}\boldsymbol{F}_{\theta^-}}{\mathrm{d}t}}\Big).
\end{aligned} \tag{37}$$

Comparing to Eq. (6), it is easy to see that $\lim_{t' \to t} \Delta_{\theta^-}(\boldsymbol{x}_t, t, t') = \frac{d\boldsymbol{f}_{\theta^-}}{dt}(\boldsymbol{x}_t, t)$. Moreover, when using $d(\boldsymbol{x}, \boldsymbol{y}) = \|\boldsymbol{x} - \boldsymbol{y}\|_2^2$, and $\Delta t = t - t'$, the training objective of discrete-time CMs in Eq. (1) becomes

$$\mathbb{E}_{\boldsymbol{x}_t, t}\left[ w(t)\|\boldsymbol{f}_\theta(\boldsymbol{x}_t, t) - \boldsymbol{f}_{\theta^-}(\boldsymbol{x}_{t-\Delta t}, t - \Delta t)\|_2^2 \right],$$

which has the gradient of

$$\begin{aligned}
&\mathbb{E}_{\boldsymbol{x}_t, t}\left[ w(t)\nabla_\theta \boldsymbol{f}_\theta^\top(\boldsymbol{x}_t, t)\left(\boldsymbol{f}_{\theta^-}(\boldsymbol{x}_t, t) - \boldsymbol{f}_{\theta^-}(\boldsymbol{x}_{t-\Delta t}, t - \Delta t)\right) \right] \\
&= \nabla_\theta \mathbb{E}_{\boldsymbol{x}_t, t}\left[ -w(t)\sin(t - t')\boldsymbol{f}_\theta^\top(\boldsymbol{x}_t, t)\Delta_{\theta^-}(\boldsymbol{x}_t, t, t') \right] \\
&= \nabla_\theta \mathbb{E}_{\boldsymbol{x}_t, t}\left[ w(t)\sin(t - t')\sin(t)\boldsymbol{F}_\theta^\top(\boldsymbol{x}_t, t)\Delta_{\theta^-}(\boldsymbol{x}_t, t, t') \right]
\end{aligned} \tag{38}$$

**Adaptive weighting for discrete-time CMs.** Inspired by the continuous-time consistency models, we can also apply the adaptive weighting technique into discrete-time training objectives in Eq. (38). Specifically, since $\Delta_{\theta^-}(\boldsymbol{x}_t, t, t')$ is a first-order approximation of $\frac{d\boldsymbol{f}_{\theta^-}}{dt}(\boldsymbol{x}_t, t)$, we can directly replace the tangent in Eq. (8) with $\Delta_{\theta^-}(\boldsymbol{x}_t, t, t')$, and obtain the improved objective of discrete-time CMs by:

$$\mathcal{L}_{\text{sCM}}(\theta, \phi) := \mathbb{E}_{\boldsymbol{x}_t, t}\left[ \frac{e^{w_\phi(t)}}{D} \left\| \boldsymbol{F}_\theta\left(\frac{\boldsymbol{x}_t}{\sigma_d}, t\right) - \boldsymbol{F}_{\theta^-}\left(\frac{\boldsymbol{x}_t}{\sigma_d}, t\right) - \cos(t)\Delta_{\theta^-}(\boldsymbol{x}_t, t, t') \right\|_2^2 - w_\phi(t) \right], \tag{39}$$

where $w_\phi(t)$ is the adaptive weighting network.

**Tangent normalization for discrete-time CMs.** We apply the simliar tangent normalization method as continuous-time CMs by defining

$$\boldsymbol{g}_{\theta^-}(\boldsymbol{x}_t, t, t') := \frac{\cos(t)\Delta_{\theta^-}(\boldsymbol{x}_t, t, t')}{\|\cos(t)\Delta_{\theta^-}(\boldsymbol{x}_t, t, t')\| + c},$$

where $c > 0$ is a hyperparameter, and then the objective in Eq. (39) becomes

$$\mathcal{L}_{\text{sCM}}(\theta, \phi) := \mathbb{E}_{\boldsymbol{x}_t, t}\left[ \frac{e^{w_\phi(t)}}{D} \left\| \boldsymbol{F}_\theta\left(\frac{\boldsymbol{x}_t}{\sigma_d}, t\right) - \boldsymbol{F}_{\theta^-}\left(\frac{\boldsymbol{x}_t}{\sigma_d}, t\right) - \boldsymbol{g}_{\theta^-}(\boldsymbol{x}_t, t, t') \right\|_2^2 - w_\phi(t) \right],$$

**Tangent warmup for discrete-time CMs.** We replace the $\Delta_{\theta^-}(\boldsymbol{x}_t, t, t')$ with the warmup version:

$$\begin{aligned}
\Delta_{\theta^-}(\boldsymbol{x}_t, t, t', r) = &-\cos(t')\left( \sigma_d \boldsymbol{F}_{\theta^-}\left(\frac{\boldsymbol{x}_t}{\sigma_d}, t\right) - \sigma_d \boldsymbol{F}_{\text{pretrain}}\left(\frac{\boldsymbol{x}_t}{\sigma_d}, t\right) \right) \\
&- r \cdot \sin(t')\left( \boldsymbol{x}_t + \frac{\sigma_d \cos(t - t')\boldsymbol{F}_{\theta^-}\left(\frac{\boldsymbol{x}_t}{\sigma_d}, t\right) - \sigma_d \boldsymbol{F}_{\theta^-}\left(\frac{\boldsymbol{x}_{t'}}{\sigma_d}, t'\right)}{\sin(t - t')} \right),
\end{aligned}$$

where $r$ linearly increases from 0 to 1 over the first 10k training iterations.

We provide the detailed algorithm of discrete-time sCM (**dsCM**) in Algorithm 2, where we refer to consistency distillation of discrete-time sCM as **dsCD**.

---

**Algorithm 2** Simplified and Stabilized Discrete-time Consistency Distillation (dsCD).

---

**Input:** dataset $\mathcal{D}$ with std. $\sigma_d$, pretrained diffusion model $\boldsymbol{F}_{\text{pretrain}}$ with parameter $\theta_{\text{pretrain}}$, model $\boldsymbol{F}_\theta$, weighting $w_\phi$, learning rate $\eta$, proposal $(P_{\text{mean}}, P_{\text{std}})$, constant $c$, warmup iteration $H$.
**Init:** $\theta \leftarrow \theta_{\text{pretrain}}$, Iters $\leftarrow 0$.
**repeat**
    $\boldsymbol{x}_0 \sim \mathcal{D}, \boldsymbol{z} \sim \mathcal{N}(\boldsymbol{0}, \sigma_d^2 \boldsymbol{I}), \tau \sim \mathcal{N}(P_{\text{mean}}, P_{\text{std}}^2), t \leftarrow \arctan(\frac{e^\tau}{\sigma_d}), \boldsymbol{x}_t \leftarrow \cos(t)\boldsymbol{x}_0 + \sin(t)\boldsymbol{z}$
    $\boldsymbol{x}_{t'} \leftarrow \cos(t - t')\boldsymbol{x}_t - \sigma_d \sin(t - t')\boldsymbol{F}_{\text{pretrain}}\left(\frac{\boldsymbol{x}_t}{\sigma_d}, t\right)$
    $r \leftarrow \min(1, \text{Iters}/H)$                                           ▷ Tangent warmup
    $\boldsymbol{g} \leftarrow \cos(t)\Delta_{\theta^-}(\boldsymbol{x}_t, t, t', r)$                          ▷ JVP rearrangement
    $\boldsymbol{g} \leftarrow \boldsymbol{g}/(\|\boldsymbol{g}\| + c)$                                 ▷ Tangent normalization
    $\mathcal{L}(\theta, \phi) \leftarrow \frac{e^{w_\phi(t)}}{D}\|\boldsymbol{F}_\theta(\frac{\boldsymbol{x}_t}{\sigma_d}, t) - \boldsymbol{F}_{\theta^-}(\frac{\boldsymbol{x}_t}{\sigma_d}, t) - \boldsymbol{g}\|_2^2 - w_\phi(t)$     ▷ Adaptive weighting
    $(\theta, \phi) \leftarrow (\theta, \phi) - \eta\nabla_{\theta, \phi}\mathcal{L}(\theta, \phi)$
    Iters $\leftarrow$ Iters $+ 1$
**until** convergence

---

### E.2   EXPERIMENTS OF DISCRETE-TIME sCM

We use the algorithm in Algorithm 2 to train discrete-time sCM, where we split $[0, \frac{\pi}{2}]$ into $N$ intervals by EDM sampling spacing. Specifically, we first obtain the EDM time step by $\sigma_i = (\sigma_{\min}^{1/\rho} + \frac{i}{M}(\sigma_{\max}^{1/\rho} - \sigma_{\min}^{1/\rho}))^\rho$ with $\rho = 7, \sigma_{\min} = 0.002$ and $\sigma_{\max} = 80$, and then obtain $t_i = \arctan(\sigma_i/\sigma_d)$ and set $t_0 = 0$. During training, we sample $t$ with a discrete categorical distribution that splits the log-normal proposal distribution as used in continuous-time sCM, similar to Song & Dhariwal (2023).

As demonstrated in Figure 5(c), increasing the number of discretization steps $N$ in discrete-time CMs improves sample quality by reducing discretization errors, but obviously degrades once $N$ becomes too large (after $N > 1024$) to suffer from numerical precision issues. By contrast, continuous-time CMs significantly outperform discrete-time CMs across all $N$'s which provides strong justification for choosing continuous-time CMs over discrete-time counterparts.

### E.3   COMPARISON WITH ECT

We compare the 1-step sampling FID scores at different training iterations between ECT (Geng et al., 2024) and sCT on CIFAR-10. As shown in Table 3, our proposed sCT significantly outperforms ECT during the training, demonstrating the effectiveness of the compute efficiency and faster convergence of sCT.

For fair comparison, we use the same network architecture with ECT on CIFAR-10, which is the DDPM++ network proposed by Ho et al. (2020) and does not have AdaGN layer, and use the same dropout rate of 0.20 as ECT, and use the same batch size (128) as ECT (which is different from our default setting of 512 in our reported results in Table 1). We choose $P_{\text{mean}} = -1.0$ and $P_{\text{std}} = 1.8$ for sCT, and use TrigFlow parameterization (with $c_{\text{noise}} = t$). All the other hyperparameters are the same as the experiments in Table 1.

Table 3: Sample quality measured by FID score ($\downarrow$) of ECT (Geng et al., 2024) and sCT at different training iterations on CIFAR-10.

| Training Iterations | 100k | 200k | 400k |
|---|---|---|---|
| ECT | 4.54 | 3.86 | 3.60 |
| sCT (ours) | **3.97** | **3.51** | **3.09** |

## F   JACOBIAN-VECTOR PRODUCT OF FLASH ATTENTION

The attention operator (Vaswani, 2017) needs to compute $\boldsymbol{y} = \text{softmax}(\boldsymbol{x})\boldsymbol{V}$ where $\boldsymbol{x} \in \mathbb{R}^{1 \times L}, \boldsymbol{V} \in \mathbb{R}^{L \times D}, \boldsymbol{y} \in \mathbb{R}^{1 \times D}$. Flash Attention (Dao et al., 2022; Dao, 2023) computes the output by maintaining three variables $m(\boldsymbol{x}) \in \mathbb{R}, \ell(\boldsymbol{x}) \in \mathbb{R}$, and $\boldsymbol{f}(\boldsymbol{x})$ with the same dimension as $\boldsymbol{x}$. The computation is done recursively: for each block, we have

$$m(\boldsymbol{x}) = \max(e^{\boldsymbol{x}}), \quad \ell(\boldsymbol{x}) = \sum_i e^{\boldsymbol{x}^{(i)} - m(\boldsymbol{x})}, \quad \boldsymbol{f}(\boldsymbol{x}) = e^{\boldsymbol{x} - m(\boldsymbol{x})}\boldsymbol{V},$$

and for combining two blocks $\boldsymbol{x} = [\boldsymbol{x}^{(a)}, \boldsymbol{x}^{(b)}]$, we merge their corresponding $m, \ell, \boldsymbol{f}$ by

$$m(\boldsymbol{x}) = \max\left(\boldsymbol{x}^{(a)}, \boldsymbol{x}^{(b)}\right), \quad \ell(\boldsymbol{x}) = e^{m(\boldsymbol{x}^{(a)}) - m(\boldsymbol{x})}\ell(\boldsymbol{x}^{(a)}) + e^{m(\boldsymbol{x}^{(b)}) - m(\boldsymbol{x})}\ell(\boldsymbol{x}^{(b)}),$$

$$\boldsymbol{f}(\boldsymbol{x}) = \left[e^{m(\boldsymbol{x}^{(a)}) - m(\boldsymbol{x})}\boldsymbol{f}(\boldsymbol{x}^{(a)}), e^{m(\boldsymbol{x}^{(b)}) - m(\boldsymbol{x})}\boldsymbol{f}(\boldsymbol{x}^{(b)})\right], \quad \boldsymbol{y} = \frac{\boldsymbol{f}(\boldsymbol{x})}{\ell(\boldsymbol{x})}.$$

However, to the best of knowledge, there does not exist an algorithm for computing the Jacobian-Vector product of the attention operator in the Flash Attention style for faster computation and memory saving. We propose a recursive algorithm for the JVP computation of Flash Attention below.

Denote $\boldsymbol{p} := \text{softmax}(\boldsymbol{x})$. Denote the tangent vector for $\boldsymbol{x} \in \mathbb{R}^{1 \times L}, \boldsymbol{p} \in \mathbb{R}^{1 \times L}, \boldsymbol{V} \in \mathbb{R}^{L \times D}, \boldsymbol{y} \in \mathbb{R}^{1 \times D}$ as $\boldsymbol{t_x} \in \mathbb{R}^{1 \times L}, \boldsymbol{t_p} \in \mathbb{R}^{1 \times L}, \boldsymbol{t_V} \in \mathbb{R}^{L \times D}, \boldsymbol{t_y} \in \mathbb{R}^{1 \times D}$, correspondingly. The JVP for attention is computing $(\boldsymbol{x}, \boldsymbol{t_x}), (\boldsymbol{V}, \boldsymbol{t_V}) \to (\boldsymbol{y}, \boldsymbol{t_y})$, which is

$$\boldsymbol{t_y} = \boldsymbol{t_p}\boldsymbol{V} + \underbrace{\boldsymbol{p}\boldsymbol{t_V}}_{\text{softmax}(\boldsymbol{x})\boldsymbol{t_V}}, \quad \text{where} \quad \boldsymbol{t_p}\boldsymbol{V} = \underbrace{(\boldsymbol{p} \odot \boldsymbol{t_x})}_{1 \times L}\boldsymbol{V} - \underbrace{(\boldsymbol{p}\boldsymbol{t_x}^\top)}_{1 \times 1} \cdot \underbrace{(\boldsymbol{p}\boldsymbol{V})}_{\boldsymbol{y}}.$$

Notably, the computation for both $\boldsymbol{p}\boldsymbol{t}_V$ and $\boldsymbol{p}V$ can be done by the standard Flash Attention with the value matrices $\boldsymbol{V}$ and $\boldsymbol{t}_V$. Thus, to compute $\boldsymbol{t}_y$, we only need to maintain a vector $\boldsymbol{g}(\boldsymbol{x}) := (\boldsymbol{p} \odot \boldsymbol{t}_x)\boldsymbol{V}$ and a scalar $\mu(\boldsymbol{x}) := \boldsymbol{p}\boldsymbol{t}_x^\top$ during the Flash Attention computation loop. Moreover, since we do not know $\boldsymbol{p}$ during the loop, we can reuse the intermediate $m, \ell, \boldsymbol{f}$ in Flash Attention. Specifically, for each block,

$$\boldsymbol{g}(\boldsymbol{x}) = \left(e^{\boldsymbol{x}-m(\boldsymbol{x})} \odot \boldsymbol{t}_x\right)\boldsymbol{V}, \quad \mu(\boldsymbol{x}) = \sum_i e^{\boldsymbol{x}^{(i)}-m(\boldsymbol{x})}\boldsymbol{t}_x^{(i)},$$

and for combining two blocks $\boldsymbol{x} = [\boldsymbol{x}^{(a)}, \boldsymbol{x}^{(b)}]$, we merge their corresponding $\boldsymbol{g}$ and $\mu$ by

$$\boldsymbol{g}(\boldsymbol{x}) = \left[e^{m(\boldsymbol{x}^{(a)})-m(\boldsymbol{x})}\boldsymbol{g}(\boldsymbol{x}^{(a)}), e^{m(\boldsymbol{x}^{(b)})-m(\boldsymbol{x})}\boldsymbol{g}(\boldsymbol{x}^{(b)})\right],$$

$$\mu(\boldsymbol{x}) = e^{m(\boldsymbol{x}^{(a)})-m(\boldsymbol{x})}\mu(\boldsymbol{x}^{(a)}) + e^{m(\boldsymbol{x}^{(b)})-m(\boldsymbol{x})}\mu(\boldsymbol{x}^{(b)}),$$

and after obtaining $m, \ell, \boldsymbol{f}, \boldsymbol{g}, \mu$ for the row vector $\boldsymbol{x}$, the final result of $\boldsymbol{t}_p V$ is

$$\boldsymbol{t}_p V = \frac{\boldsymbol{g}(\boldsymbol{x})}{\ell(\boldsymbol{x})} - \frac{\mu(\boldsymbol{x})}{\ell(\boldsymbol{x})} \cdot \boldsymbol{y}.$$

Therefore, we can use a single loop to obtain both the output $\boldsymbol{y}$ and the JVP output $\boldsymbol{t}_y$, which accesses the memory for the attention matrices only once and avoids saving the intermediate activations, thus saving the GPU memory.

## G    EXPERIMENT SETTINGS AND RESULTS

### G.1    TRIGFLOW FOR DIFFUSION MODELS

We train the teacher diffusion models on CIFAR-10, ImageNet 64×64 and ImageNet 512×512 with the proposed improvements of parameterization and architecture, including TrigFlow parameterization, positional time embedding and adaptive double normalization layer. We list the detailed settings below.

**CIFAR-10.** Our architecture is based on the Score SDE (Song et al., 2021b) architecture (DDPM++). We use the same settings of EDM (Karras et al., 2022): dropout rate is 0.13, batch size is 512, number of training iterations is 400k, learning rate is 0.001, Adam $\epsilon = 10^{-8}$, $\beta_1 = 0.9$, $\beta_2 = 0.999$. We use 2nd-order single-step DPM-Solver (Lu et al., 2022a) (DPM-Solver-2S) with Heun's intermediate time step with 18 steps (NFE=35), which is exactly equivalent to EDM Heun's sampler. We obtain FID of 2.15 for the teacher model.

**ImageNet 64×64.** We preprocess the ImageNet dataset following Dhariwal & Nichol (2021) by

1. Resize the shorter width / height to $64 \times 64$ resolution with bicubic interpolation.
2. Center crop the image.
3. Disable data augmentation such as horizontal flipping.

Except for the TrigFlow parameterization, positional time embedding and adaptive double normalization layer, we follow exactly the same setting in EDM2 config G (Karras et al., 2024) to train models with sizes of S, M, L, and XL, while the only difference is that we use Adam $\epsilon = 10^{-11}$.

**ImageNet 512×512.** We preprocess the ImageNet dataset following Dhariwal & Nichol (2021) and Karras et al. (2024) by

1. Resize the shorter width / height to $512 \times 512$ resolution with bicubic interpolation.
2. Center crop the image.
3. Disable data augmentation such as horizontal flipping.
4. Encode the images into latents by stable diffusion VAE[2] (Rombach et al., 2022; Janner et al., 2022), and rescale the latents by channel mean $\mu_c = [1.56, -0.695, 0.483, 0.729]$ and channel std $\sigma_c = [5.27, 5.91, 4.21, 4.31]$. We keep the $\sigma_d = 0.5$ as in EDM2 (Karras et al., 2024), so for each latent we substract $\mu_c$ and multiply it by $\sigma_d/\sigma_c$.

---

[2]https://huggingface.co/stabilityai/sd-vae-ft-mse

When sampling from the model, we redo the scaling of the generated latents and then run the VAE decoder. Notably, our channel mean and channel std are different from those in EDM2 (Karras et al., 2024). It is because when training the VAE, the images are normalized to $[-1, 1]$ before passing to the encoder. However, the channel mean and std used in EDM2 assumes the input images are in $[0, 1]$ range, which mismatches the training phase of the VAE. We empirically find that it is hard to distinguish the reconstructed samples by human eyes of these two different normalization, while it has non-ignorable influence for training diffusion models evaluated by FID. After fixing this mismatch, our diffusion model slightly outperforms the results of EDM2 at larger scales (XL and XXL). More results are provided in Table 6.

Except for the TrigFlow parameterization, positional time embedding and adaptive double normalization layer, we follow exactly the same setting in EDM2 config G (Karras et al., 2024) to train models with sizes of S, M, L, XL and XXL, while the only difference is that we use Adam $\epsilon = 10^{-11}$. We enable label dropout with rate $0.1$ to support classifier-free guidance. We use 2nd-order single-step DPM-Solver (Lu et al., 2022a) (DPM-Solver-2S) with Heun's intermediate time step with 32 steps (NFE=63), which is exactly equivalent to EDM Heun's sampler. We find that the optimal guidance scale for classifier-free guidance and the optimal EMA rate are also the same as EDM2 for all model sizes.

## G.2 Continuous-time Consistency Models

In all experiments, we use $c = 0.1$ for tangent normalization, and use $H = 10000$ for tangent warmup. We always use the same batch size as the teacher diffusion training, which is different from Song & Dhariwal (2023). During sampling, we start at $t_{\max} = \arctan\left(\frac{\sigma_{\max}}{\sigma_d}\right)$ with $\sigma_{\max} = 80$ such that it matches the starting time of EDM (Karras et al., 2022) and EDM2 (Karras et al., 2024). For 2-step sampling, we use the algorithm in Song et al. (2023) with an intermediate $t = 1.1$ for all the experiments. We always initialize the CM from the EMA parameters of the teacher diffusion model. For sCD, we always use the $F_{\text{pretrain}}$ of the teacher diffusion model with its EMA parameters during distillation.

We empirically find that the proposal distribution should have small $P_{\text{mean}}$, i.e. close to the clean data, to ensure the training stability and improve the final performance. Intuitively, this is because the training signal of CMs only come from the clean data, so we need to reduce the training error for $t$ near to 0 to further reduce the accumulation errors.

**CIFAR-10.** For both sCT and sCD, we initialize from the teacher diffusion model trained with the settings in Appendix G.1, and use RAdam optimizer (Liu et al., 2019) with learning rate of $0.0001$, $\beta_1 = 0.9$, $\beta_2 = 0.99$, $\epsilon = 10^{-8}$, and without learning rate schedulers. proposal distribution of $P_{\text{mean}} = -1.0, P_{\text{std}} = 1.4$. For the attention layers, we use the implementation in (Karras et al., 2022) which naturally supports JVP by PyTorch (Paszke et al., 2019) auto-grad. We use EMA half-life of $0.5$ Mimg (Karras et al., 2022). We use dropout rate of $0.20$ for sCT and disable dropout for sCD.

**ImageNet 64×64.** We only enable dropout at the resolutions equal to or less than 16, following Simple Diffusion (Hoogeboom et al., 2023) and iCT (Song & Dhariwal, 2023). We multiply the learning rate of the teacher diffusion model by $0.01$ for both sCT and sCD. We train the model with half precision (FP16), and use the flash attention jvp proposed in Appendix F for computing the tangents of flash attention layers. Other training settings are the same as the teacher diffusion models. More details of training and sampling are provided in Table 4 and Table 8. During sampling, we always use EMA length $\sigma_{\text{rel}} = 0.05$ for sampling from CMs.

**ImageNet 512×512.** We only enable dropout at the resolutions equal to or less than 16, following Simple Diffusion (Hoogeboom et al., 2023) and iCT (Song & Dhariwal, 2023). We multiply the learning rate of the teacher diffusion model by $0.01$ for both sCT and sCD. We train the model with half precision (FP16), and use the flash attention jvp proposed in Appendix F for computing the tangents of flash attention layers. Other training settings are the same as the teacher diffusion models. More details of training and sampling are provided in Table 5 and Table 6. During sampling, we always use EMA length $\sigma_{\text{rel}} = 0.05$ for sampling from CMs.

We add an additional input in $F_\theta(\frac{x_t}{\sigma_d}, t, s)$ where $s$ represents the CFG guidance scale of the teacher model, where $s$ is embedded by positioinal embedding layer and an additional linear layer, and the embedding is added to the embedding of $t$, similar to the label conditioning. During training, we

Table 4: Training settings of all models and training algorithms on ImageNet 64×64 dataset.

| | Model Size | | | |
| | S | M | L | XL |
|---|---|---|---|---|
| **Model details** | | | | |
| Batch size | 2048 | 2048 | 2048 | 2048 |
| Channel multiplier | 192 | 256 | 320 | 384 |
| Time embedding layer | positional | positional | positional | positional |
| noise conditioning $c_{\text{noise}}(t)$ | $t$ | $t$ | $t$ | $t$ |
| adaptive double normalization | ✓ | ✓ | ✓ | ✓ |
| Learning rate decay ($t_{\text{ref}}$) | 35000 | 35000 | 35000 | 35000 |
| Adam $\beta_1$ | 0.9 | 0.9 | 0.9 | 0.9 |
| Adam $\beta_2$ | 0.99 | 0.99 | 0.99 | 0.99 |
| Adam $\epsilon$ | 1.0e-11 | 1.0e-11 | 1.0e-11 | 1.0e-11 |
| Model capacity (Mparams) | 280.2 | 497.8 | 777.6 | 1119.4 |
| **Training details of diffusion models (TrigFlow)** | | | | |
| Training iterations | 1048k | 1486k | 761k | 540k |
| Learning rate max ($\alpha_{\text{ref}}$) | 1.0e-2 | 9.0e-3 | 8.0e-3 | 7.0e-3 |
| Dropout probability | 0% | 10% | 10% | 10% |
| Proposal $P_{\text{mean}}$ | -0.8 | -0.8 | -0.8 | -0.8 |
| Proposal $P_{\text{std}}$. | 1.6 | 1.6 | 1.6 | 1.6 |
| **Shared details of consistency models** | | | | |
| Learning rate max ($\alpha_{\text{ref}}$) | 1.0e-4 | 9.0e-5 | 8.0e-5 | 7.0e-5 |
| Proposal $P_{\text{mean}}$ | -1.0 | -1.0 | -1.0 | -1.0 |
| Proposal $P_{\text{std}}$. | 1.6 | 1.6 | 1.6 | 1.6 |
| Tangent normalization constant ($c$) | 0.1 | 0.1 | 0.1 | 0.1 |
| Tangent warm up iterations | 10k | 10k | 10k | 10k |
| EMA length ($\sigma_{\text{rel}}$) of pretrained diffusion | 0.075 | 0.06 | 0.04 | 0.04 |
| **Training details of sCT** | | | | |
| Training iterations | 400k | 400k | 400k | 400k |
| Dropout probability for resolution $\leq 16$ | 45% | 45% | 45% | 45% |
| **Training details of sCD** | | | | |
| Training iterations | 400k | 400k | 400k | 400k |
| Dropout probability for resolution $\leq 16$ | 0% | 0% | 0% | 0% |

uniformly sample $s \in [1, 2]$ and apply CFG with guidance scale $s$ to the teacher diffusion model to get $\boldsymbol{F}_{\text{pretrain}}$.

**VSD experiments.** We do not use EMA for $\boldsymbol{F}_\phi$ in VSD, instead we always use the original model for $\boldsymbol{F}_\phi$ for stabilizing the training. The learning rate of $\boldsymbol{F}_\phi$ is the same as the learning rate of CMs. More details and results are provided in Tables 5 to 7.

Table 5: Training settings of all models and training algorithms on ImageNet 512×512 dataset.

| | Model Size | | | | |
|---|---|---|---|---|---|
| | S | M | L | XL | XXL |
| **Model details** | | | | | |
| Batch size | 2048 | 2048 | 2048 | 2048 | 2048 |
| Channel multiplier | 192 | 256 | 320 | 384 | 448 |
| Time embedding layer | positional | positional | positional | positional | positional |
| noise conditioning $c_{\text{noise}}(t)$ | $t$ | $t$ | $t$ | $t$ | $t$ |
| adaptive double normalization | ✓ | ✓ | ✓ | ✓ | ✓ |
| Learning rate decay ($t_{\text{ref}}$) | 70000 | 70000 | 70000 | 70000 | 70000 |
| Adam $\beta_1$ | 0.9 | 0.9 | 0.9 | 0.9 | 0.9 |
| Adam $\beta_2$ | 0.99 | 0.99 | 0.99 | 0.99 | 0.99 |
| Adam $\epsilon$ | 1.0e-11 | 1.0e-11 | 1.0e-11 | 1.0e-11 | 1.0e-11 |
| Model capacity (Mparams) | 280.2 | 497.8 | 777.6 | 1119.4 | 1523.4 |
| **Training details of diffusion models (TrigFlow)** | | | | | |
| Training iterations | 1048k | 1048k | 696k | 598k | 376k |
| Learning rate max ($\alpha_{\text{ref}}$) | 1.0e-2 | 9.0e-3 | 8.0e-3 | 7.0e-3 | 6.5e-3 |
| Dropout probability | 0% | 10% | 10% | 10% | 10% |
| Proposal $P_{\text{mean}}$ | -0.4 | -0.4 | -0.4 | -0.4 | -0.4 |
| Proposal $P_{\text{std}}$. | 1.0 | 1.0 | 1.0 | 1.0 | 1.0 |
| **Shared details of consistency models** | | | | | |
| Learning rate max ($\alpha_{\text{ref}}$) | 1.0e-4 | 9.0e-5 | 8.0e-5 | 7.0e-5 | 6.5e-5 |
| Proposal $P_{\text{mean}}$ | -0.8 | -0.8 | -0.8 | -0.8 | -0.8 |
| Proposal $P_{\text{std}}$. | 1.6 | 1.6 | 1.6 | 1.6 | 1.6 |
| Tangent normalization constant ($c$) | 0.1 | 0.1 | 0.1 | 0.1 | 0.1 |
| Tangent warm up iterations | 10k | 10k | 10k | 10k | 10k |
| EMA length ($\sigma_{\text{rel}}$) of pretrained diffusion | 0.025 | 0.03 | 0.015 | 0.02 | 0.015 |
| **Training details of sCT** | | | | | |
| Training iterations | 100k | 100k | 100k | 100k | 100k |
| Dropout probability for resolution $\leq 16$ | 25% | 35% | 35% | 35% | 35% |
| **Training details of sCD** | | | | | |
| Training iterations | 200k | 200k | 200k | 200k | 200k |
| Dropout probability for resolution $\leq 16$ | 0% | 10% | 10% | 10% | 10% |
| Maximum of CFG scale | 2.0 | 2.0 | 2.0 | 2.0 | 2.0 |
| **Training details of sCD with adaptive VSD** | | | | | |
| Training iterations | 20k | 20k | 20k | 20k | 20k |
| Learning rate max ($\alpha_{\text{ref}}$) for $\boldsymbol{F}_\phi$ | 1.0e-4 | 9.0e-5 | 8.0e-5 | 7.0e-5 | 6.5e-5 |
| Dropout probability for $\boldsymbol{F}_\phi$ | 0% | 10% | 10% | 10% | 10% |
| Proposal $P_{\text{mean}}$ for $\mathcal{L}_{\text{Diff}}(\phi)$ | -0.8 | -0.8 | -0.8 | -0.8 | -0.8 |
| Proposal $P_{\text{std}}$. for $\mathcal{L}_{\text{Diff}}(\phi)$ | 1.6 | 1.6 | 1.6 | 1.6 | 1.6 |
| Number of updating of $\phi$ per updating of $\theta$ | 1 | 1 | 1 | 1 | 1 |
| One-step sampling starting time $t_{\text{max}}$ | $\arctan(\frac{80}{\sigma_d})$ | $\arctan(\frac{80}{\sigma_d})$ | $\arctan(\frac{80}{\sigma_d})$ | $\arctan(\frac{80}{\sigma_d})$ | $\arctan(\frac{80}{\sigma_d})$ |
| Proposal $P_{\text{mean}}$ for $\mathcal{L}_{\text{VSD}}(\theta)$ | 0.4 | 0.4 | 0.4 | 0.4 | 0.4 |
| Proposal $P_{\text{std}}$. for $\mathcal{L}_{\text{VSD}}(\theta)$ | 2.0 | 2.0 | 2.0 | 2.0 | 2.0 |
| Loss weighting $\lambda_{\text{VSD}}$ for $\mathcal{L}_{\text{VSD}}$ | 1.0 | 1.0 | 1.0 | 1.0 | 1.0 |

Table 6: Evaluation of sample quality of different models on ImageNet 512×512 dataset. Results of EDM2 (Karras et al., 2024) are with EDM parameterization and the original AdaGN layer. [†]The $\text{FD}_{\text{DINOv2}}$ in EDM2 are obtained by tuned EMA rate, which is different from our EMA rates that are tuned for FID scores.

| | Model Size | | | | |
| --- | --- | --- | --- | --- | --- |
| | S | M | L | XL | XXL |
| **Sampling by diffusion models (NFE = 126)** | | | | | |
| EMA length ($\sigma_{\text{rel}}$) | 0.025 | 0.030 | 0.015 | 0.020 | 0.015 |
| Guidance scale for FID | 1.4 | 1.2 | 1.2 | 1.2 | 1.2 |
| [†]Guidance scale for $\text{FD}_{\text{DINOv2}}$ | 2.0 | 1.8 | 1.8 | 1.8 | 1.8 |
| FID (TrigFlow) | 2.29 | 2.00 | 1.87 | 1.80 | 1.73 |
| FID (EDM2) | 2.23 | 2.01 | 1.88 | 1.85 | 1.81 |
| $\text{FD}_{\text{DINOv2}}$(TrigFlow) | 52.08 | 43.33 | 39.23 | 36.73 | 35.93 |
| [†]$\text{FD}_{\text{DINOv2}}$(EDM2) with $\sigma_{\text{rel}}$ for $\text{FD}_{\text{DINOv2}}$ | 52.32 | 41.98 | 38.20 | 35.67 | 33.09 |
| **Sampling by consistency models trained with sCT** | | | | | |
| Intermediate time $t_{\text{mid}}$ in 2-step sampling | 1.1 | 1.1 | 1.1 | 1.1 | 1.1 |
| 1-step FID | 10.13 | 5.84 | 5.15 | 4.33 | 4.29 |
| 2-step FID | 9.86 | 5.53 | 4.65 | 3.73 | 3.76 |
| 1-step $\text{FD}_{\text{DINOv2}}$ | 278.35 | 192.13 | 169.98 | 147.06 | 146.31 |
| 2-step $\text{FD}_{\text{DINOv2}}$ | 244.41 | 160.66 | 135.80 | 114.65 | 112.69 |
| **Sampling by consistency models trained with sCD** | | | | | |
| Intermediate time $t_{\text{mid}}$ in 2-step sampling | 1.1 | 1.1 | 1.1 | 1.1 | 1.1 |
| Guidance scale for FID, 1-step sampling | 1.5 | 1.3 | 1.3 | 1.3 | 1.3 |
| Guidance scale for FID, 2-step sampling | 1.4 | 1.2 | 1.2 | 1.2 | 1.2 |
| Guidance scale for $\text{FD}_{\text{DINOv2}}$, 1-step sampling | 2.0 | 2.0 | 2.0 | 2.0 | 2.0 |
| Guidance scale for $\text{FD}_{\text{DINOv2}}$, 2-step sampling | 2.0 | 2.0 | 1.9 | 1.9 | 1.9 |
| 1-step FID | 3.07 | 2.75 | 2.55 | 2.40 | 2.28 |
| 2-step FID | 2.50 | 2.26 | 2.04 | 1.93 | 1.88 |
| 1-step $\text{FD}_{\text{DINOv2}}$ | 104.22 | 83.78 | 76.10 | 70.30 | 67.80 |
| 2-step $\text{FD}_{\text{DINOv2}}$ | 71.15 | 55.70 | 50.63 | 46.66 | 44.97 |
| **Sampling by consistency models trained with multistep sCD** | | | | | |
| Guidance scale for FID | 1.4 | 1.2 | 1.2 | 1.15 | 1.15 |
| Guidance scale for $\text{FD}_{\text{DINOv2}}$ | 2.0 | 2.0 | 2.0 | 1.9 | 1.9 |
| FID, $M = 2$ | 2.79 | 2.51 | 2.32 | 2.29 | 2.16 |
| FID, $M = 4$ | 2.78 | 2.46 | 2.28 | 2.22 | 2.10 |
| FID, $M = 8$ | 2.49 | 2.24 | 2.04 | 2.02 | 1.90 |
| FID, $M = 16$ | 2.34 | 2.18 | 1.99 | 1.90 | 1.82 |
| $\text{FD}_{\text{DINOv2}}$, $M = 2$ | 76.29 | 60.47 | 54.91 | 51.91 | 50.70 |
| $\text{FD}_{\text{DINOv2}}$, $M = 4$ | 72.01 | 56.38 | 50.99 | 47.61 | 46.78 |
| $\text{FD}_{\text{DINOv2}}$, $M = 8$ | 60.13 | 49.46 | 44.87 | 41.26 | 40.56 |
| $\text{FD}_{\text{DINOv2}}$, $M = 16$ | 55.89 | 46.94 | 42.55 | 39.30 | 38.55 |
| **Sampling by consistency models trained with sCD + adaptive VSD** | | | | | |
| Intermediate time $t_{\text{mid}}$ in 2-step sampling | 1.1 | 1.1 | 1.1 | 1.1 | 1.1 |
| Guidance scale for FID, 1-step sampling | 1.2 | 1.0 | 1.0 | 1.0 | 1.0 |
| Guidance scale for FID, 2-step sampling | 1.2 | 1.0 | 1.0 | 1.0 | 1.0 |
| Guidance scale for $\text{FD}_{\text{DINOv2}}$, 1-step sampling | 1.7 | 1.5 | 1.6 | 1.5 | 1.5 |
| Guidance scale for $\text{FD}_{\text{DINOv2}}$, 2-step sampling | 1.7 | 1.5 | 1.6 | 1.5 | 1.5 |
| 1-step FID | 3.37 | 2.67 | 2.26 | 2.39 | 2.16 |
| 2-step FID | 2.70 | 2.29 | 1.99 | 2.01 | 1.89 |
| 1-step $\text{FD}_{\text{DINOv2}}$ | 72.12 | 54.81 | 50.46 | 48.11 | 45.54 |
| 2-step $\text{FD}_{\text{DINOv2}}$ | 69.00 | 53.53 | 48.54 | 46.61 | 43.93 |

Table 7: Ablation of adaptive VSD and sCD on ImageNet 512×512 dataset with model size M.

|  | Method | | |
| --- | --- | --- | --- |
|  | VSD | sCD | sCD + VSD |
| EMA length ($\sigma_{\text{rel}}$) | 0.05 | 0.05 | 0.05 |
| Guidance scale for FID, 1-step sampling | 1.1 | 1.3 | 1.0 |
| Guidance scale for FID, 2-step sampling | \ | 1.2 | 1.0 |
| Guidance scale for FD$_{\text{DINOv2}}$, 1-step sampling | 1.4 | 2.0 | 1.5 |
| Guidance scale for FD$_{\text{DINOv2}}$, 2-step sampling | \ | 2.0 | 1.5 |
| 1-step FID | 3.02 | 2.75 | **2.67** |
| 2-step FID | \ | **2.26** | 2.29 |
| 1-step FD$_{\text{DINOv2}}$ | 57.19 | 83.78 | **54.81** |
| 2-step FD$_{\text{DINOv2}}$ | \ | 55.70 | **53.53** |

Table 8: Evaluation of sample quality of different models on ImageNet 64×64 dataset.

|  | Model Size | | | |
| --- | --- | --- | --- | --- |
|  | S | M | L | XL |
| **Sampling by diffusion models (NFE=63)** | | | | |
| EMA length ($\sigma_{\text{rel}}$) | 0.075 | 0.06 | 0.04 | 0.04 |
| FID (TrigFlow) | 1.70 | 1.55 | 1.44 | 1.38 |
| **Sampling by consistency models trained with sCT** | | | | |
| Intermediate time $t_{\text{mid}}$ in 2-step sampling | 1.1 | 1.1 | 1.1 | 1.1 |
| 1-step FID | 3.23 | 2.25 | 2.08 | 2.04 |
| 2-step FID | 2.93 | 1.81 | 1.57 | 1.48 |
| **Sampling by consistency models trained with sCD** | | | | |
| Intermediate time $t_{\text{mid}}$ in 2-step sampling | 1.1 | 1.1 | 1.1 | 1.1 |
| 1-step FID | 2.97 | 2.79 | 2.43 | 2.44 |
| 2-step FID | 2.07 | 1.89 | 1.70 | 1.66 |

# H ADDITIONAL SAMPLES

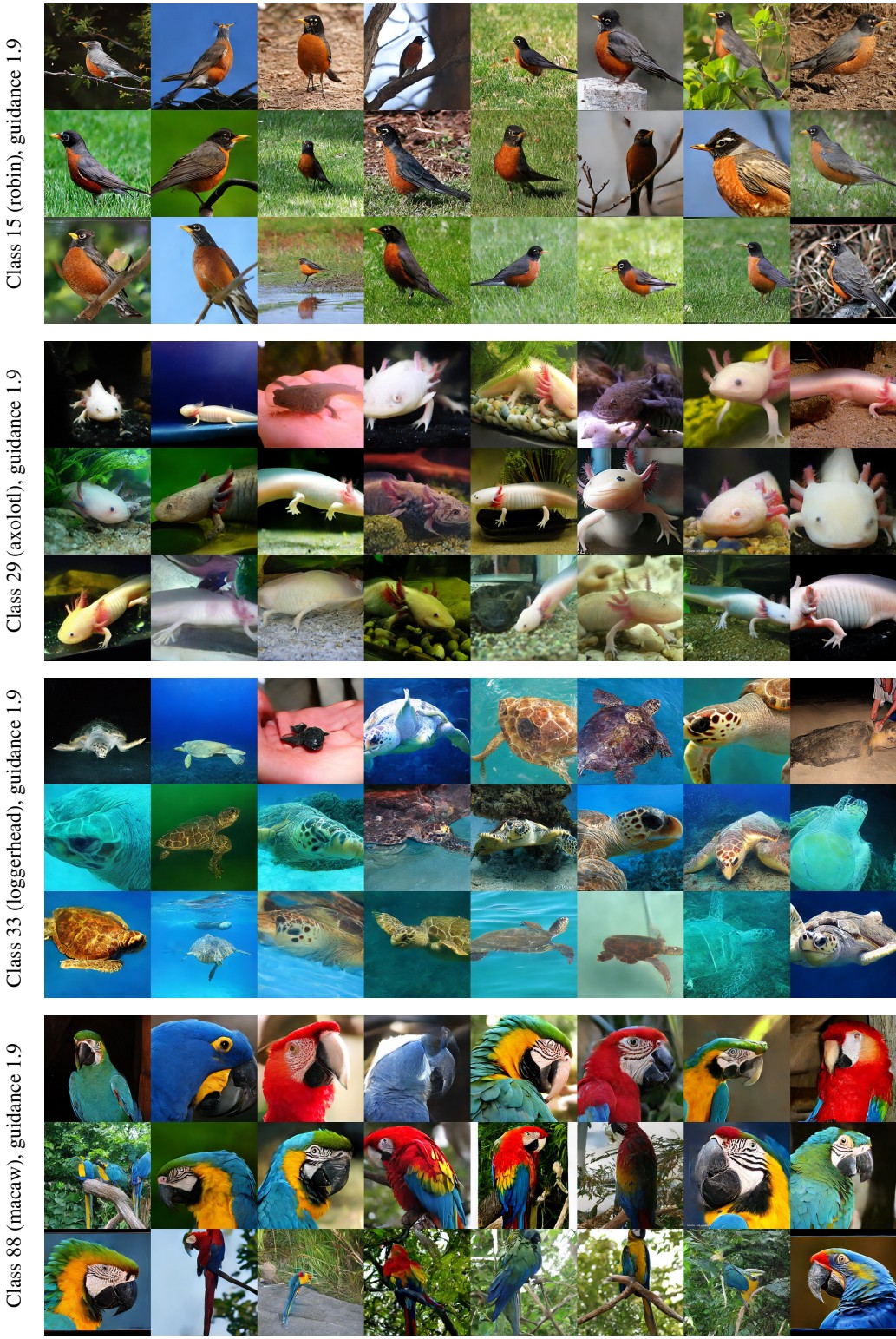

Figure 8: Uncurated 1-step samples generated by our sCD-XXL trained on ImageNet 512×512.

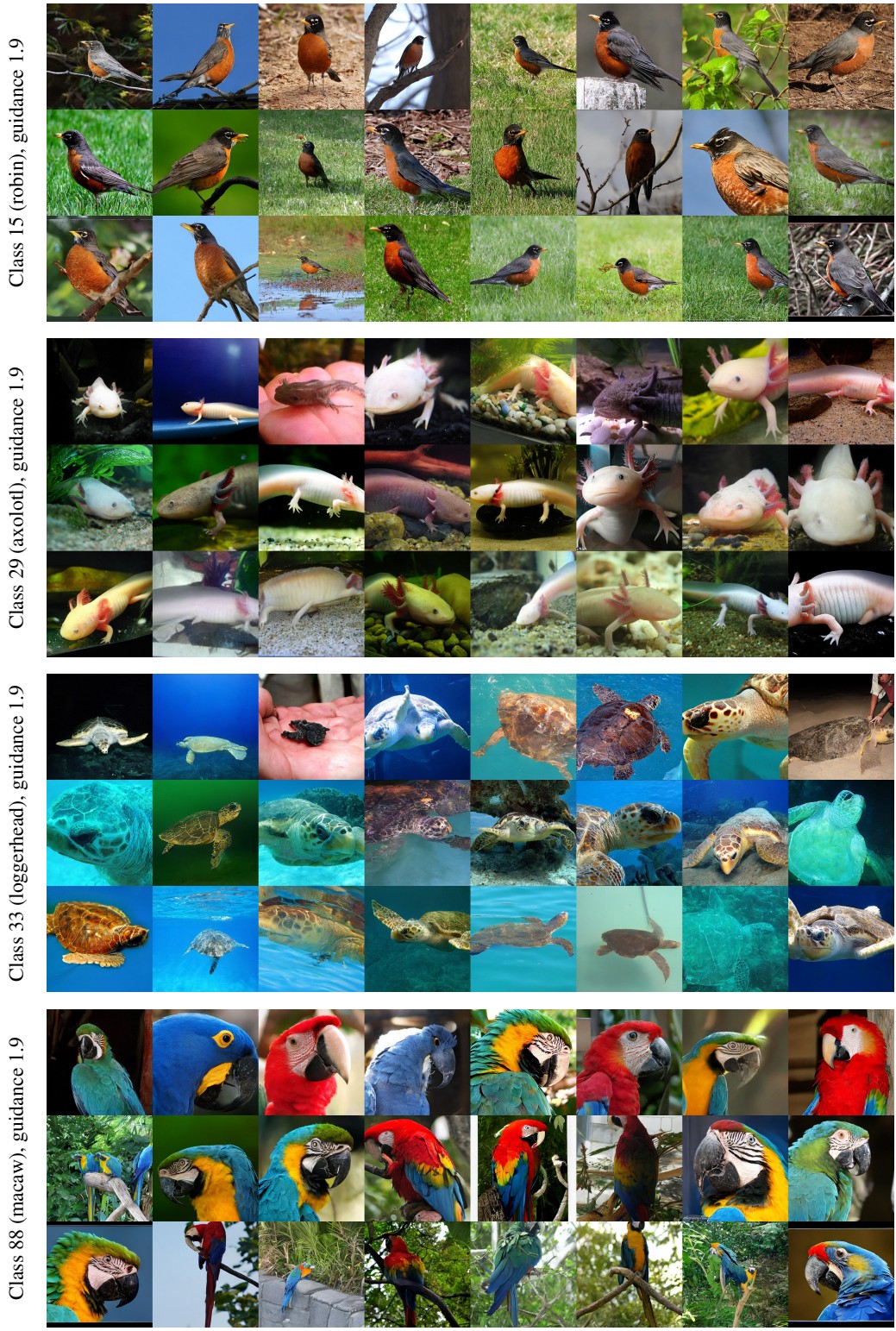

Figure 9: Uncurated 2-step samples generated by our sCD-XXL trained on ImageNet 512×512.

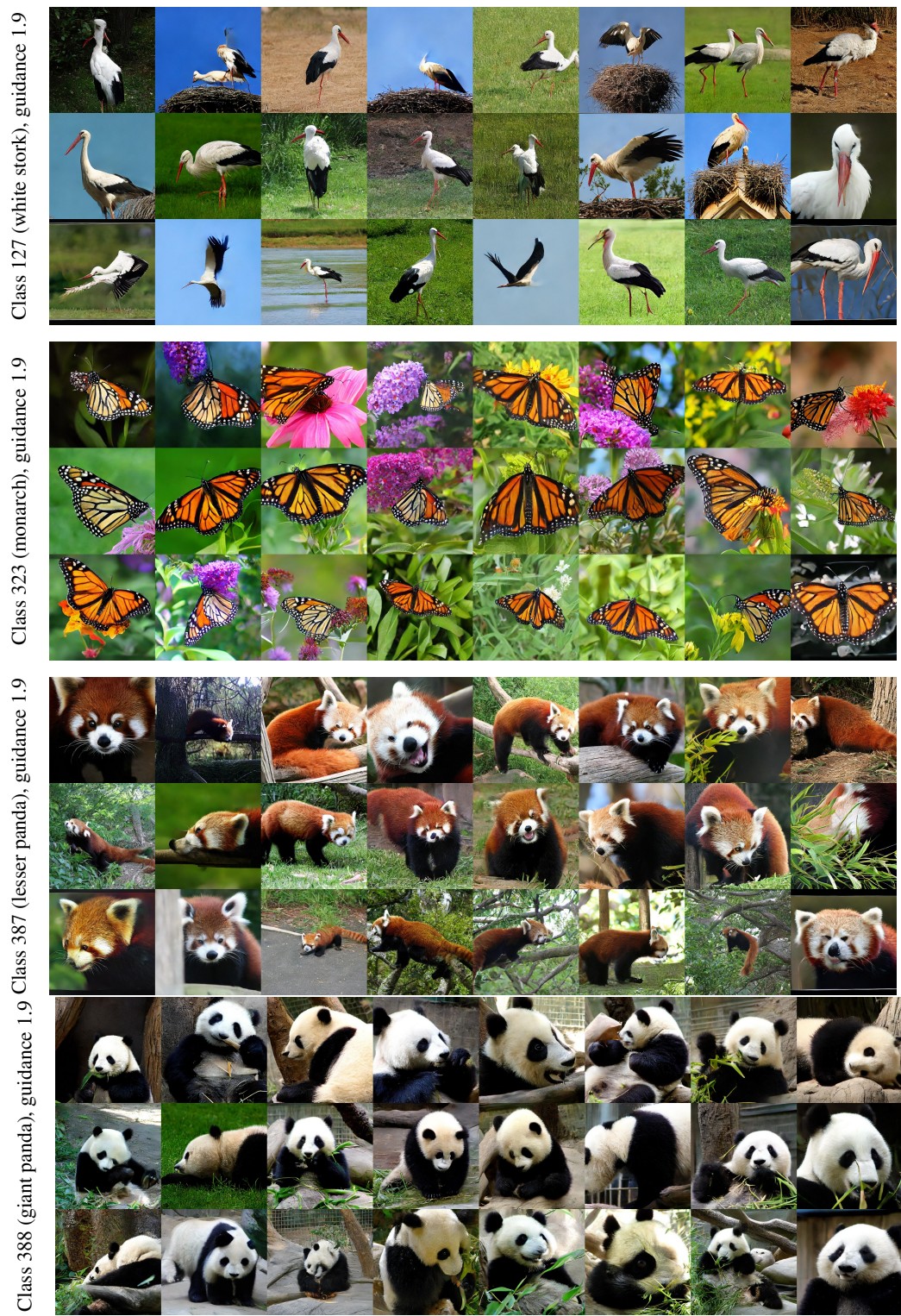

Figure 10: Uncurated 1-step samples generated by our sCD-XXL trained on ImageNet 512×512.

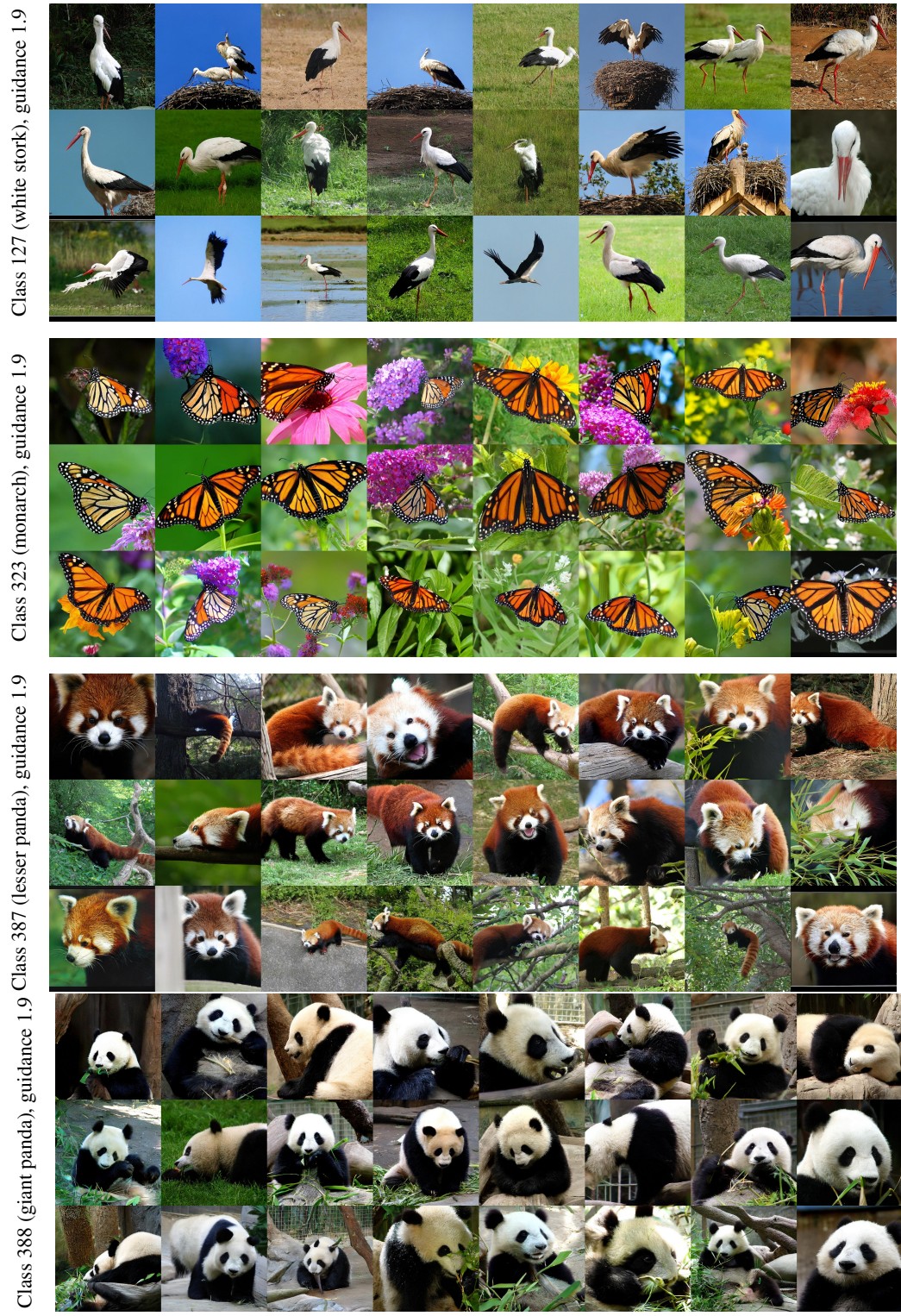

Figure 11: Uncurated 2-step samples generated by our sCD-XXL trained on ImageNet 512×512.

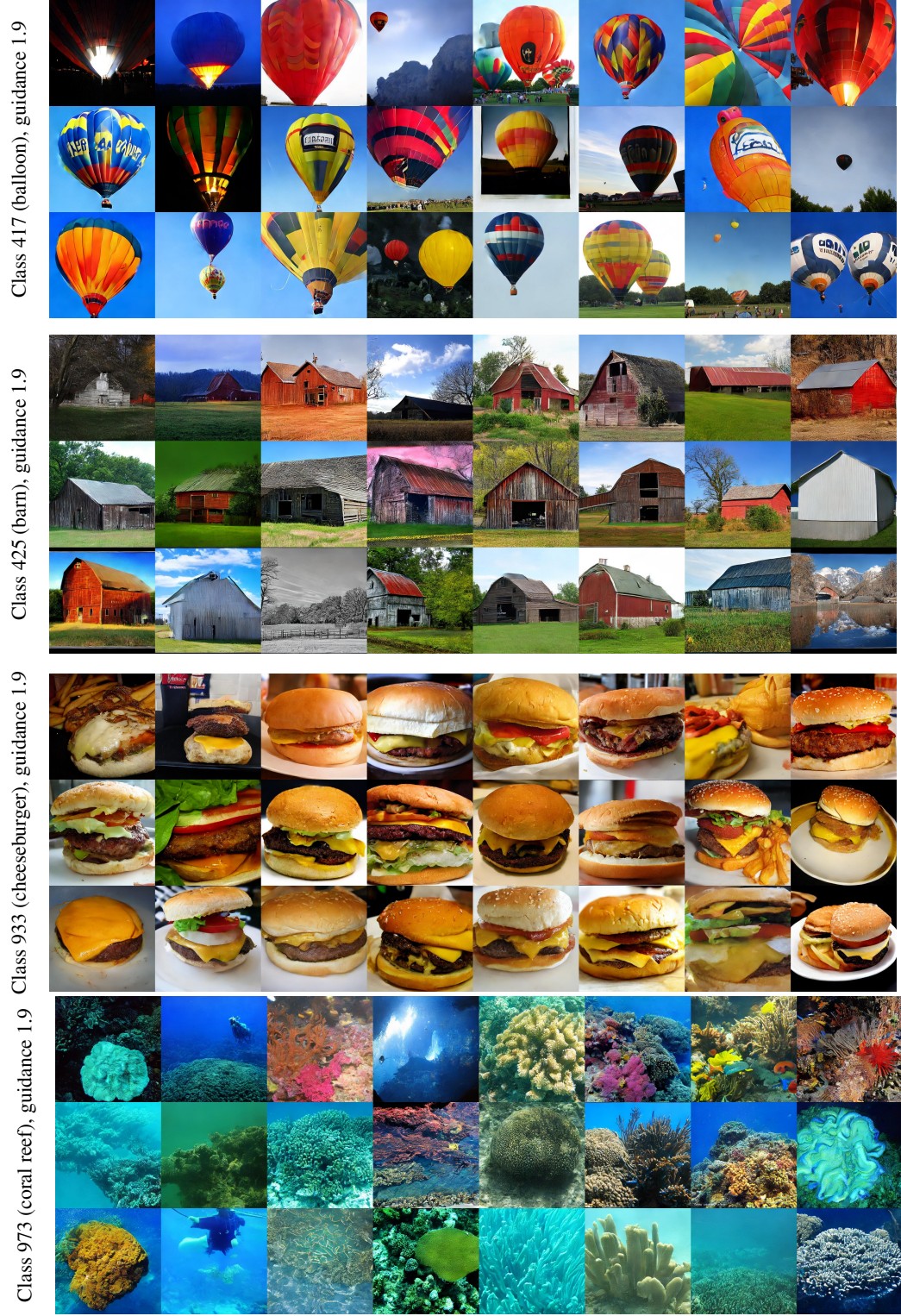

Figure 12: Uncurated 1-step samples generated by our sCD-XXL trained on ImageNet 512×512.

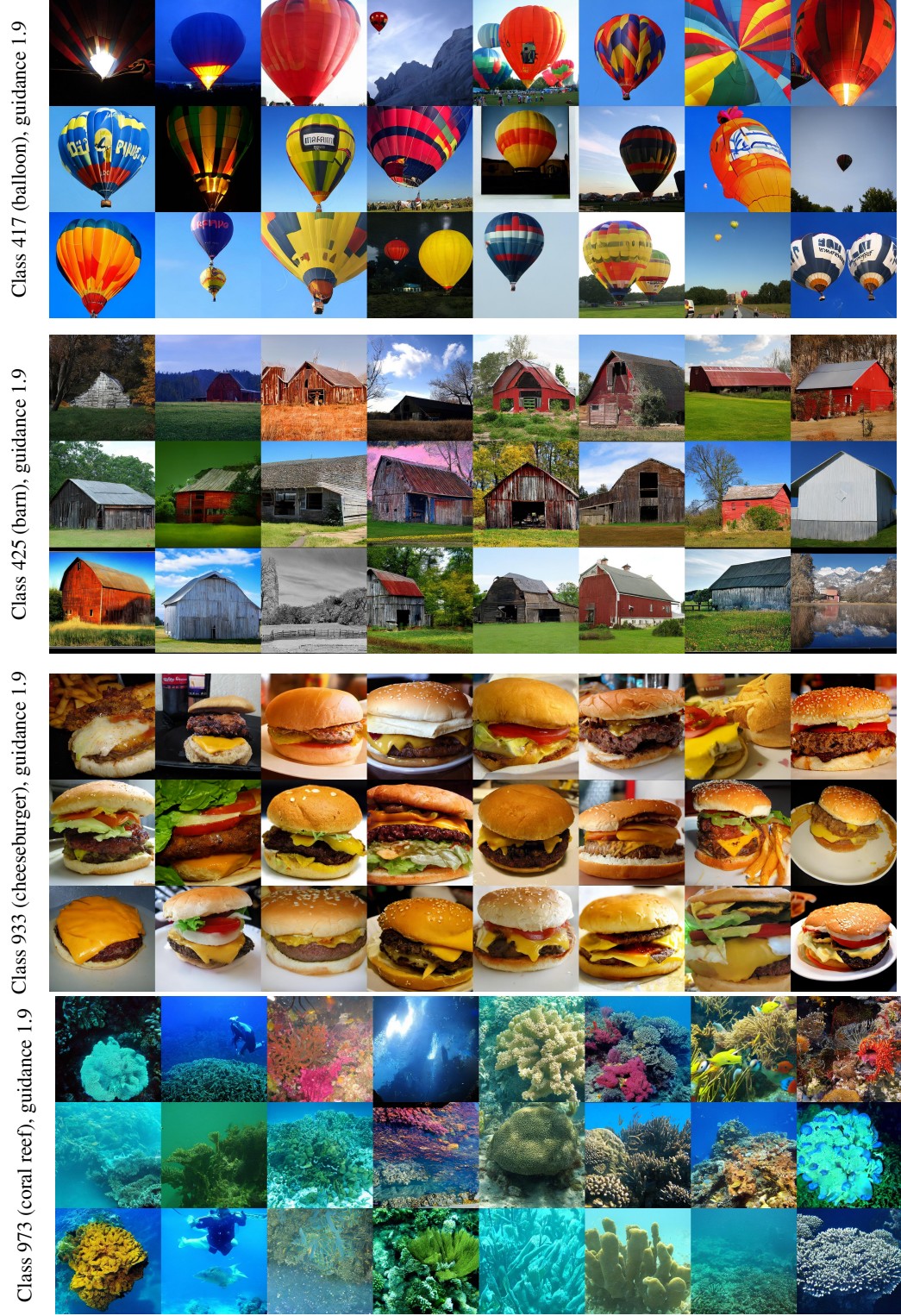

Figure 13: Uncurated 2-step samples generated by our sCD-XXL trained on ImageNet 512×512.

