# Simplifying, Stabilizing & Scaling Continuous-Time Consistency Models

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

(\boldsymbol{x}_t) + \left( \nabla_{\boldsymbol{x}_t} \log q_t^\theta(\boldsymbol{x}_t) - \nabla_{\boldsymbol{x}_t} \log p_t(\boldsymbol{x}_t) \right) \frac{\partial \boldsymbol{x}_t^\theta}{\partial \theta} \right) \right]$$

$$= \underbrace{\mathbb{E}_{t,\boldsymbol{x}_t} \left[ w(t) \partial_\theta \log q_t^\theta(\boldsymbol{x}_t) \right]}_{=0} + \mathbb{E}_{t,\boldsymbol{z},\boldsymbol{\epsilon}} \left[ w(t) \left( \nabla_{\boldsymbol{x}_t} \log q_t^\theta(\boldsymbol{x}_t) - \nabla_{\boldsymbol{x}_t} \log p_t(\boldsymbol{x}_t) \right) \frac{\alpha_t \partial \boldsymbol{g}_\theta(\boldsymbol{z})}{\partial \theta} \right]$$

$$= \mathbb{E}_{t,\boldsymbol{z},\boldsymbol{\epsilon}} \left[ \alpha_t w(t) \left( \nabla_{\boldsymbol{x}_t} \log q_t^\theta(\boldsymbol{x}_t) - \nabla_{\boldsymbol{x}_t} \log p_t(\boldsymbol{x}_t) \right) \frac{\partial \boldsymbol{g}_\theta(\boldsymbol{z})}{\partial \theta} \right].$$

Therefore, we need to approximate the score functions $\nabla_{\boldsymbol{x}_t} \log q_t^\theta(\boldsymbol{x}_t)$ for the generator and $\nabla_{\boldsymbol{x}_t} \log p_t(\boldsymbol{x}_t)$ for the data distribution. VSD trains a diffusion model for samples from $\boldsymbol{g}_\theta(\boldsymbol{z})$ to approximate $\nabla_{\boldsymbol{x}_t} \log q_t^\theta(\boldsymbol{x}_t)$ and uses a pretrained diffusion model to approximate $\nabla_{\boldsymbol{x}_t} \log p_t(\boldsymbol{x}_t)$.

In this work, we train the diffusion model in TrigFlow framework, with $\alpha_t = \cos(t)$, $\sigma_t = \sigma_d \sin(t)$, $\hat{\sigma} = \sigma_d$, $T = \frac{\pi}{2}$. Specifically, assume we have a pretrained diffusion model $\boldsymbol{F}_{\mathrm{pretrain}}$ parameterized by TrigFlow, and we train another diffusion model $\boldsymbol{F}_\phi$ to approximate the diffused generator distribution, by

$$\min_\phi \mathbb{E}_{\boldsymbol{z},\boldsymbol{z}',t} \left[ w(t) \left\| \sigma_d \boldsymbol{F}_\phi \left( \frac{\boldsymbol{x}_t}{\sigma_d}, t \right) - \boldsymbol{v}_t \right\|_2^2 \right],$$

where $\boldsymbol{x}_t = \cos(t) \boldsymbol{x}_0 + \sin(t) \boldsymbol{z}$, $\boldsymbol{v}_t = \cos(t) \boldsymbol{z} - \sin(t) \boldsymbol{x}_0$, $\boldsymbol{z} \sim \mathcal{N}(\boldsymbol{0}, \sigma_d^2 \boldsymbol{I})$, $\boldsymbol{x}_0 = \boldsymbol{g}_\theta(\boldsymbol{z}')$ with $\boldsymbol{z}' \sim \mathcal{N}(\boldsymbol{0}, \sigma_d^2 \boldsymbol{I})$. Moreover, the relationship between the ground truth diffusion model $\boldsymbol{F}_{\mathrm{Diff}}(\boldsymbol{x}_t, t)$ and the score function $\nabla_{\boldsymbol{x}_t} \log p_t(\boldsymbol{x}_t)$ is

$$\sigma_d \boldsymbol{F}_{\mathrm{Diff}}(\boldsymbol{x}_t, t) = \mathbb{E}[\boldsymbol{v}_t | \boldsymbol{x}_t] = \frac{1}{\tan(t)} \boldsymbol{x}_t - \frac{1}{\sin(t)} \mathbb{E}_{\boldsymbol{x}_0 | \boldsymbol{x}_t}[\boldsymbol{x}_0],$$

$$\nabla_{\boldsymbol{x}_t} \log p_t(\boldsymbol{x}_t) = \mathbb{E}_{\boldsymbol{x}_0 | \boldsymbol{x}_t} \left[ -\frac{\boldsymbol{x}_t - \cos(t) \boldsymbol{

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

| Guidance scale for FID | 1.4 | 1.2 | 1.2 | 1.2 | 1.2 |
| $^\dagger$Guidance scale for $FD_{DINOv2}$ | 2.0 | 1.8 | 1.8 | 1.8 | 1.8 |
| FID (TrigFlow) | 2.29 | 2.00 | 1.87 | 1.80 | 1.73 |
| FID (EDM2) | 2.23 | 2.01 | 1.88 | 1.85 | 1.81 |
| $FD_{DINOv2}$(TrigFlow) | 52.08 | 43.33 | 39.23 | 36.73 | 35.93 |
| $^\dagger FD_{DINOv2}$(EDM2) with $\sigma_{rel}$ for $FD_{DINOv2}$ | 52.32 | 41.98 | 38.20 | 35.67 | 33.09 |
| **Sampling by consistency models trained with sCT** | | | | | |
| Intermediate time $t_{mid}$ in 2-step sampling | 1.1 | 1.1 | 1.1 | 1.1 | 1.1 |
| 1-step FID | 10.13 | 5.84 | 5.15 | 4.33 | 4.29 |
| 2-step FID | 9.86 | 5.53 | 4.65 | 3.73 | 3.76 |
| 1-step $FD_{DINOv2}$ | 278.35 | 192.13 | 169.98 | 147.06 | 146.31 |
| 2-step $FD_{DINOv2}$ | 244.41 | 160.66 | 135.80 | 114.65 | 112.69 |
| **Sampling by consistency models trained with sCD** | | | | | |
| Intermediate time $t_{mid}$ in 2-step sampling | 1.1 | 1.1 | 1.1 | 1.1 | 1.1 |
| Guidance scale for FID, 1-step sampling | 1.5 | 1.3 | 1.3 | 1.3 | 1.3 |
| Guidance scale for FID, 2-step sampling | 1.4 | 1.2 | 1.2 | 1.2 | 1.2 |
| Guidance scale for $FD_{DINOv2}$, 1-step sampling | 2.0 | 2.0 | 2.0 | 2.0 | 2.0 |
| Guidance scale for $FD_{DINOv2}$, 2-step sampling | 2.0 | 2.0 | 1.9 | 1.9 | 1.9 |
| 1-step FID | 3.07 | 2.75 | 2.55 | 2.40 | 2.28 |
| 2-step FID | 2.50 | 2.26 | 2.04 | 1.93 | 1.88 |
| 1-step $FD_{DINOv2}$ | 104.22 | 83.78 | 76.10 | 70.30 | 67.80 |
| 2-step $FD_{DINOv2}$ | 71.15 | 55.70 | 50.63 | 46.66 | 44.97 |
| **Sampling by consistency models trained with multistep sCD** | | | | | |
| Guidance scale for FID | 1.4 | 1.2 | 1.2 | 1.15 | 1.15 |
| Guidance scale for $FD_{DINOv2}$ | 2.0 | 2.0 | 2.0 | 1.9 | 1.9 |
| FID, $M = 2$ | 2.79 | 2.51 | 2.32 | 2.29 | 2.16 |
| FID, $M = 4$ | 2.78 | 2.46 | 2.28 | 2.22 | 2.10 |
| FID, $M = 8$ | 2.49 | 2.24 | 2.04 | 2.02 | 1.90 |
| FID, $M = 16$ | 2.34 | 2.18 | 1.99 | 1.90 | 1.82 |
| $FD_{DINOv2}$, $M = 2$ | 76.29 | 60.47 | 54.91 | 51.91 | 50.70 |
| $FD_{DINOv2}$, $M = 4$ | 72.01 | 56.38 | 50.99 | 47.61 | 46.78 |
| $FD_{DINOv2}$, $M = 8$ | 60.13 | 49.46 | 44.87 | 41.26 | 40.56 |
| $FD_{DINOv2}$, $M = 16$ | 55.89 | 46.94 | 42.55 | 39.30 | 38.55 |
| **Sampling by consistency models trained with sCD + adaptive VSD** | | | | | |
| Intermediate time $t_{mid}$ in 2-step sampling | 1.1 | 1.1 | 1.1 | 1.1 | 1.1 |
| Guidance scale for FID, 1-step sampling | 1.2 | 1.0 | 1.0 | 1.0 | 1.0 |
| Guidance scale for FID, 2-step sampling | 1.2 | 1.0 | 1.0 | 1.0 | 1.0 |
| Guidance scale for $FD_{DINOv2}$, 1-step sampling | 1.7 | 1.5 | 1.6 | 1.5 | 1.5 |
| Guidance scale for $FD_{DINOv2}$, 2-step sampling | 1.7 | 1.5 | 1.6 | 1.5 | 1.5 |
| 1-step FID | 3.37 | 2.67 | 2.26 | 2.39 | 2.16 |
| 2-step FID | 2.70 | 2.29 | 1.99 | 2.01 | 1.89 |
| 1-step $FD_{DINOv2}$ | 72.12 | 54.81 | 50.46 | 48.11 | 45.54 |
| 2-step $FD_{DINOv2}$ | 69.00 | 53.53 | 48.54 | 46.61 | 43.93 |

Table 7: Ablation of adaptive VSD and sCD on ImageNet 512×512 dataset with model size M.

| | Method | | |
|---|---|---|---|
| | VSD | sCD | sCD + VSD |
| EMA length ($\sigma_{\text{rel}}$) | 0.05 | 0.05 | 0.05 |
| Guidance scale for FID, 1-step sampling | 1.1 | 1.3 | 1.0 |
| Guidance scale for FID, 2-step sampling | \ | 1.2 | 1.0 |
| Guidance scale for FD$_{\text{DINOv2}}$, 1-step sampling | 1.4 | 2.0 | 1.5 |
| Guidance scale for FD$_{\text{DINOv2}}$, 2-step sampling | \ | 2.0 | 1.5 |
| 1-step FID | 3.02 | 2.75 | **2.67** |
| 2-step FID | \ | **2.26** | 2.29 |
| 1-step FD$_{\text{DINOv2}}$ | 57.19 | 83.78 | **54.81** |
| 2-step FD$_{\text{DINOv2}}$ | \ | 55.70 | **53.53** |

Table 8: Evaluation of sample quality of different models on ImageNet 64×64 dataset.

| | Model Size | | | |
|---|---|---|---|---|
| | S | M | L | XL |
| **Sampling by diffusion models (NFE=63)** | | | | |
| EMA length ($\sigma_{\text{rel}}$) | 0.075 | 0.06 | 0.04 | 0.04 |
| FID (TrigFlow) | 1.70 | 1.55 | 1.44 | 1.38 |
| **Sampling by consistency models trained with sCT** | | | | |
| Intermediate time $t_{\text{mid}}$ in 2-step sampling | 1.1 | 1.1 | 1.1 | 1.1 |
| 1-step FID | 3.23 | 2.25 | 2.08 | 2.04 |
| 2-step FID | 2.93 | 1.81 | 1.57 | 1.48 |
| **Sampling by consistency models trained with sCD** | | | | |
| Intermediate time $t_{\text{mid}}$ in 2-step sampling | 1.1 | 1.1 | 1.1 | 1.1 |
| 1-step FID | 2.97 | 2.79 | 2.43 | 2.44 |
| 2-step FID | 2.07 | 1.89 | 1.70 | 1.66 |

## H  ADDITIONAL SAMPLES

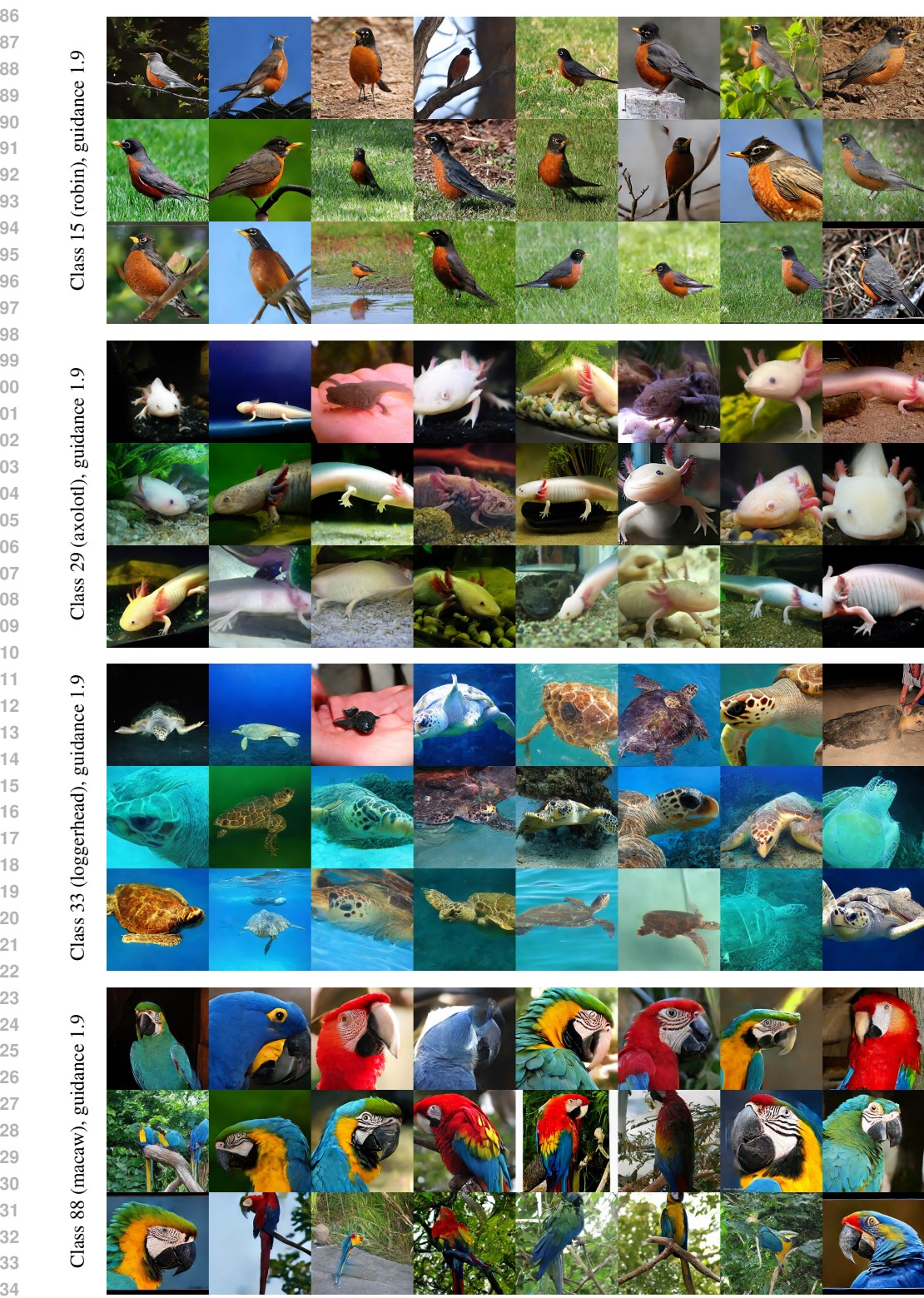

Figure 8: Uncurated 1-step samples generated by our sCD-XXL trained on ImageNet 512×512.

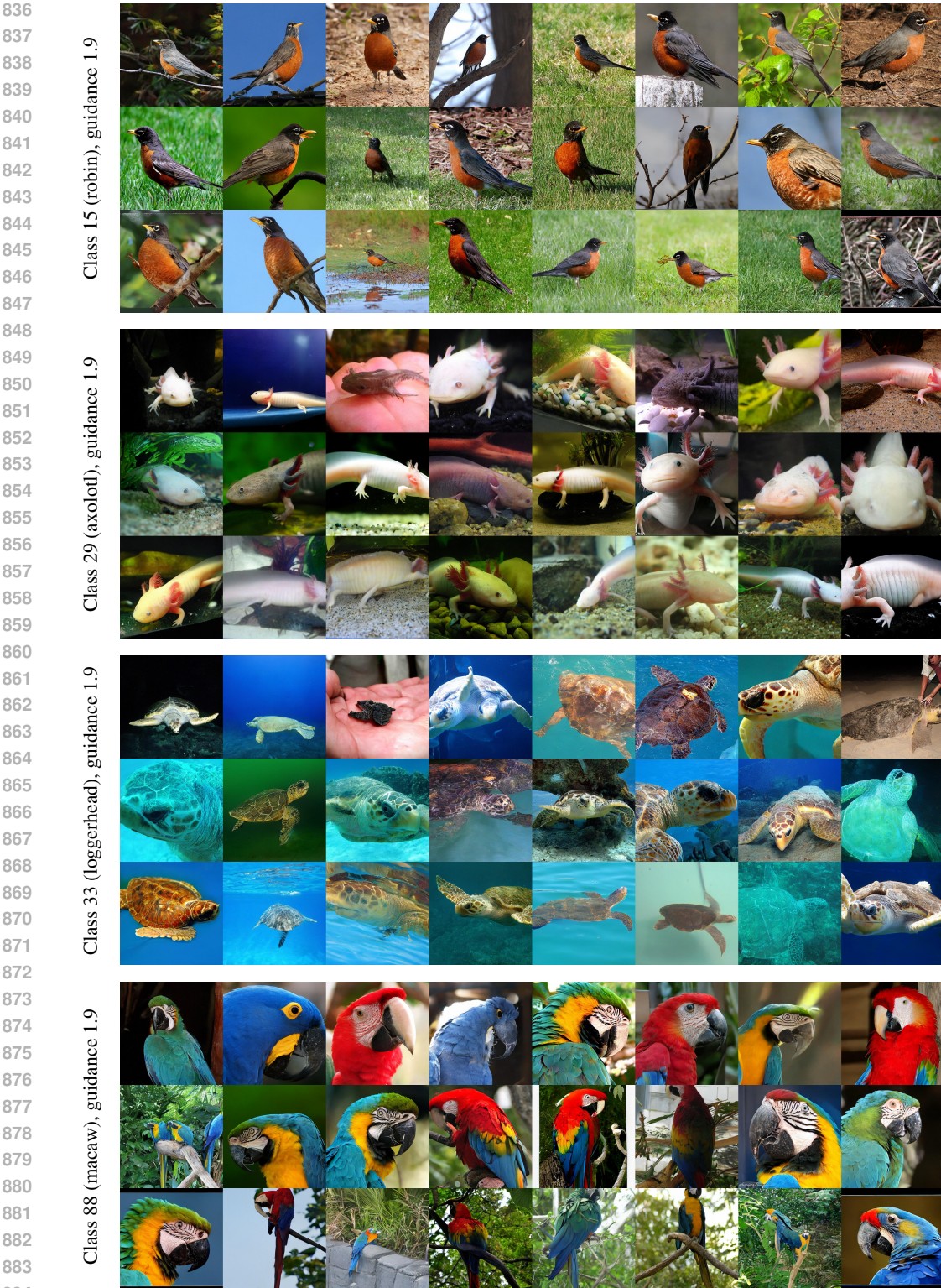

Figure 9: Uncurated 2-step samples generated by our sCD-XXL trained on ImageNet 512×512.

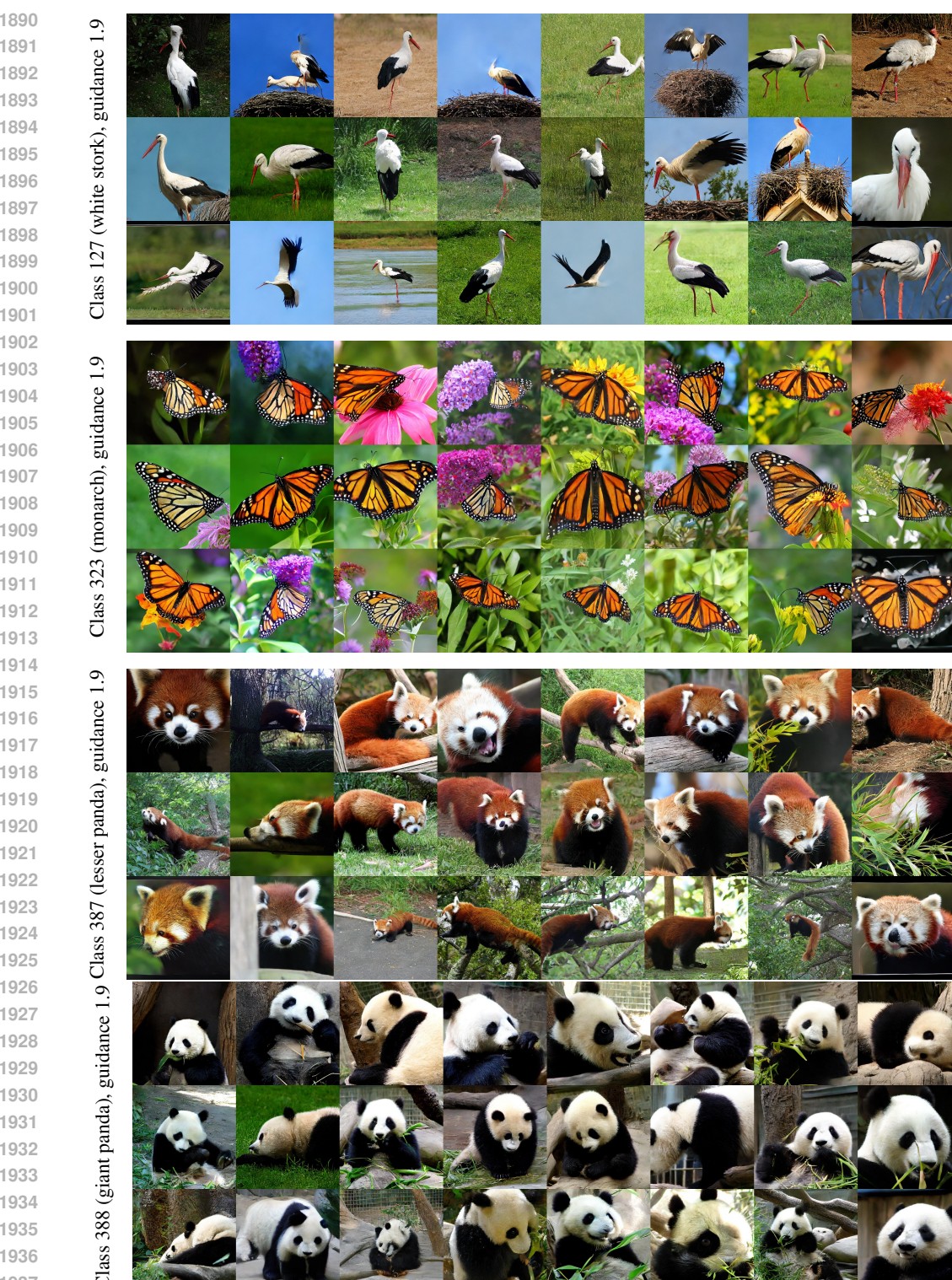

Figure 10: Uncurated 1-step samples generated by our sCD-XXL trained on ImageNet 512×512.

1944
1945
1946
1947
1948
1949
1950
1951
1952
1953
1954
1955
1956
1957
1958
1959
1960
1961
1962
1963
1964
1965
1966
1967
1968
1969
1970
1971
1972
1973
1974
1975
1976
1977
1978
1979
1980
1981
1982
1983
1984
1985
1986
1987
1988
1989
1990
1991
1992
1993
1994
1995
1996
1997

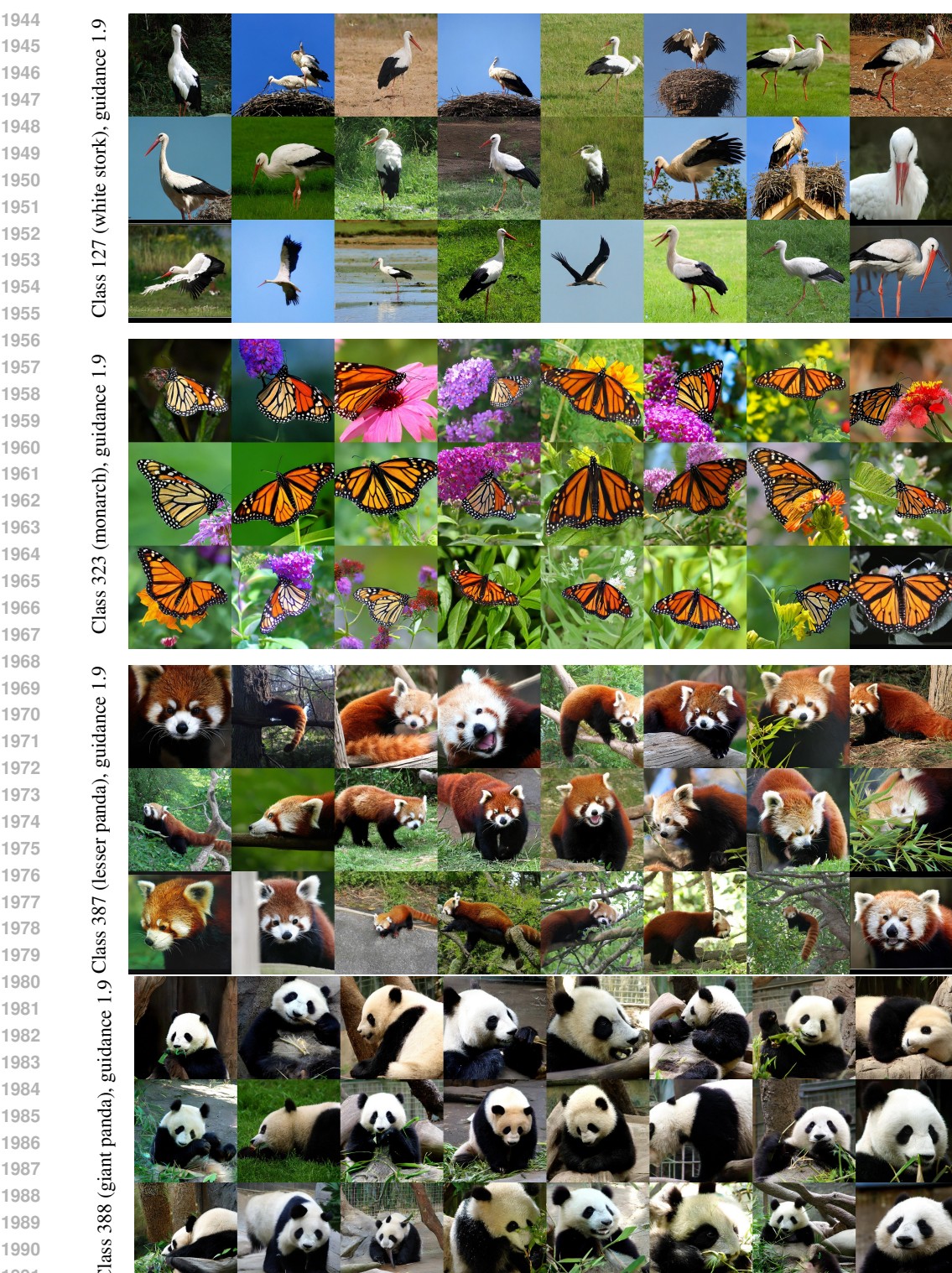

Figure 11: Uncurated 2-step samples generated by our sCD-XXL trained on ImageNet 512×512.

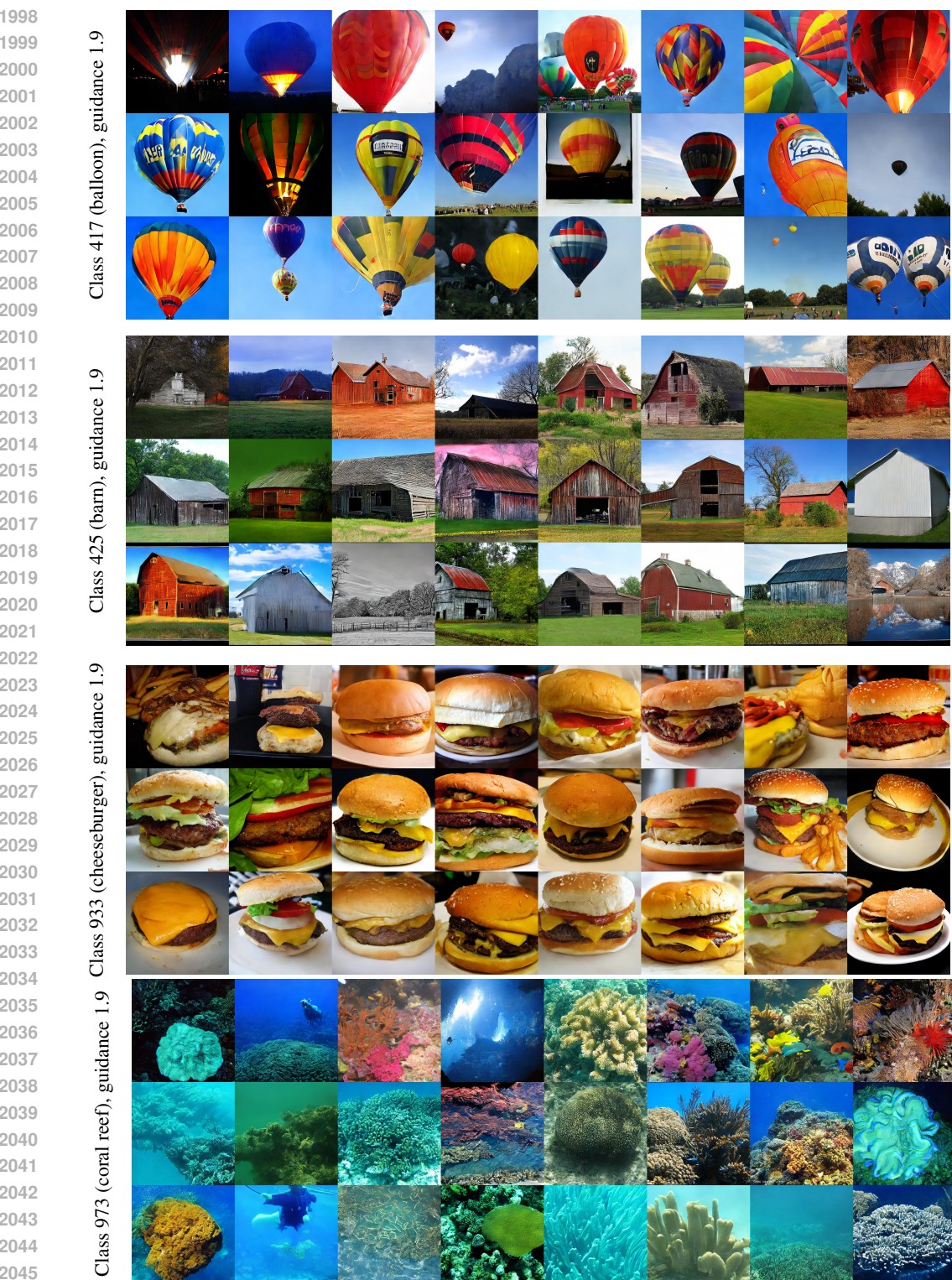

Figure 12: Uncurated 1-step samples generated by our sCD-XXL trained on ImageNet 512×512.

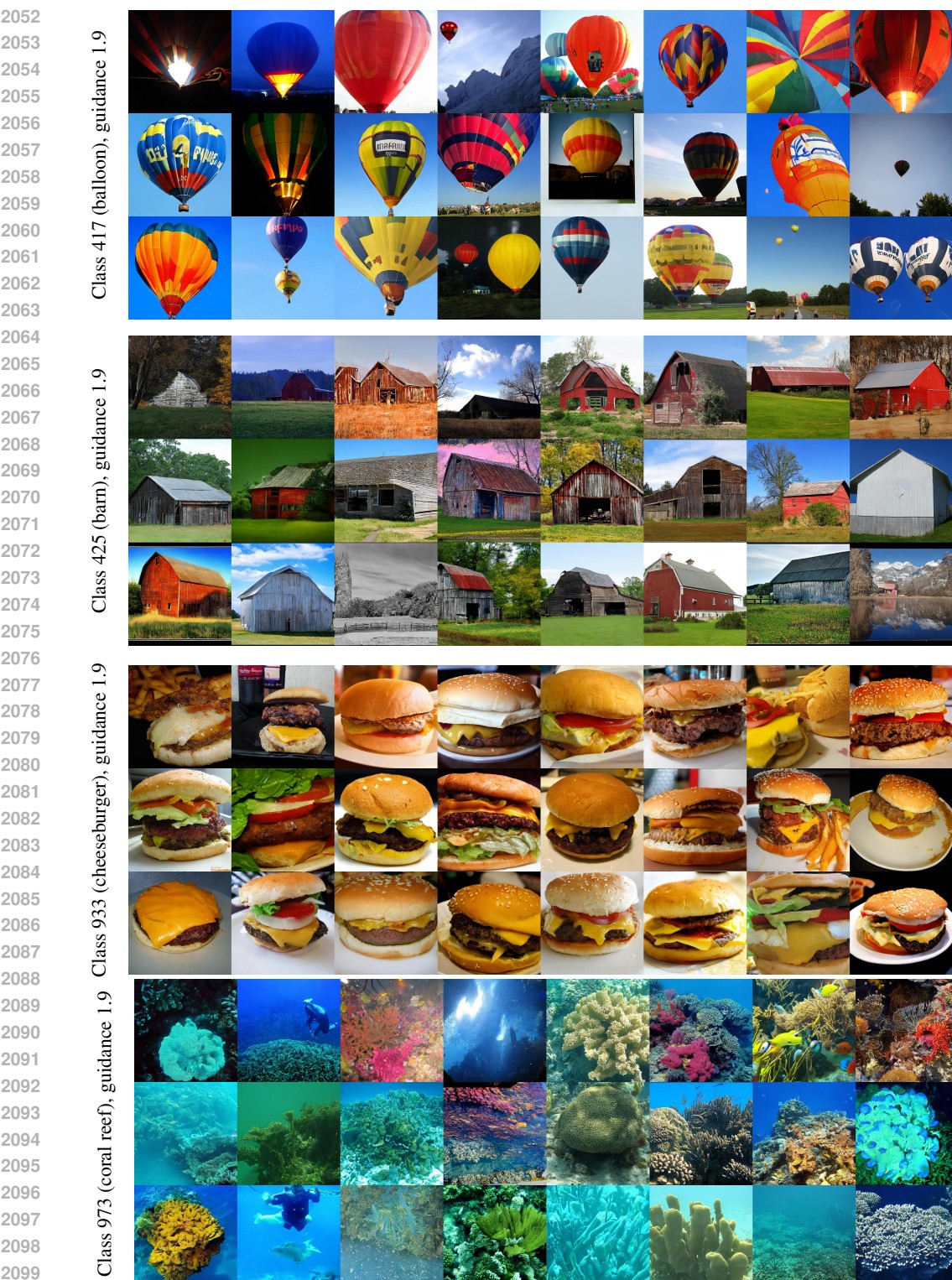

Figure 13: Uncurated 2-step samples generated by our sCD-XXL trained on ImageNet 512×512.