# OpenReview forum: "Simplifying, Stabilizing and Scaling Continuous-time Consistency Models"
_ICLR.cc/2025/Conference — ICLR 2025 Oral_

### Official Review · Reviewer_pgsr · 2024-10-31

**Soundness:** 3
**Presentation:** 3
**Contribution:** 3
**Rating:** 8
**Confidence:** 4

**Summary:**

This work proposed a set of improved training techniques to stabilize the training of continuous-time consistency models, including new consistency function formulations, new network architectures and new training objectives. With these new training techniques, the proposed method called sCMs outperformed all previous consistency models in terms of one-step and two-step FIDs.

**Strengths:**

- This paper is very well-written and easy to read.
- This work proposed a new diffusion formulation, called TrigFlow, that unifies EDM and Flowing Matching, and also simplifies the analysis of continuous-time consistency models.
- It provided a thorough analysis of the training stability of continuous-time consistency models, from the perspective of network architecture, training objective and diffusion process parameterization.
- Experiments on CIFAR-10, ImageNet-64 and ImageNet-512 demonstrate the effectiveness of the proposed method and the scalability of continuous-time consistency models.

**Weaknesses:**

- Although I really like the improvements of continuous-time consistency models, which could fundamentally eliminate the discretization error in discrete-time consistency models, it comes with more time and memory costs related to JVP computation in the loss function. To this end, this work introduces JVP of Flash Attention to reduce the costs, which is great. Still, there may be a considerable gap between the continuous-time and discrete-time consistency models. I wonder if the paper can provide a more detailed comparison between sCMs and the previous discrete-time consistency models - ECMs, in terms of the training convergence and memory cost.
- There is no explanation for the phenomenon that sCT performs better than sCD on CIFAR-10 and ImageNet-64, but sCTs performs worse on ImageNet-512. Any intuition of why sCT suffers from increased variance at larger scales?
- There are no ablation study results on “Adaptive Double Normalization” except for claiming it “removes its instability in CM training”.
- In Figure 5b, it looks like “w/o adaptive weighting” achieves better two-step FIDs than “w/ adaptive weighting” and very similar one-step FIDs to “w/ adaptive weighting”. Why do we need adaptive weighting?
- In Figure 5c, do discrete-time CMs have a constant number of time steps $N$ or a timestep schedule up to the maximum number of steps $N$? If it is the former one, it seems to be a bit unfair to discrete-time CMs because the scheduling of time steps is very important to them. Does it make more sense to compare with the best-performing discrete-time CMs?
- In Figure 7, does the paper apply TTUR proposed by DMD2 (Yin et al. 2024a)? From the DMD2 paper, TTUR improves the performance of VSD. Thus, a comparison with VSD + TTUR is more convincing.
- In Figure 7, sCDs condition the consistency network on the guidance scale $s$. I wonder if VSD also condition the generator on the guidance scale, for a consistent evaluation setting?
- A minor issue: In line 266, should it be $c_{\text{noise}}(t) = \frac{1}{4} \log(\sigma_d \tan t) $?

**Questions:**

See the weaknesses in the above.

---

> ### Author Response · Authors · 2024-11-22
> **Official response**
>
> Thank you for the detailed review and thoughtful feedback. Below we address specific questions.
>
> ***Q1: Does JVP cost more time and memory?***
>
> ***A1:*** Not really. JVP (Jacobian-vector product) can be efficiently computed using [forward-mode automatic differentiation](https://en.wikipedia.org/wiki/Automatic_differentiation), which requires the same memory and compute as a standard forward pass without saving intermediate activations. This is significantly cheaper than backpropagation, which relies on reverse-mode automatic differentiation. Consequently, our continuous-time consistency models require similar compute and memory to train when compared to their discrete-time counterparts, which perform two forward passes.
>
> ***Q2: Detailed comparison with ECT.***
>
> ***A2:*** We have updated the paper to include detailed comparison with ECT in terms of training convergence and compute. See also our [summary of updates, part C](https://openreview.net/forum?id=LyJi5ugyJx&noteId=LDtCNvtiA5).
>
> ***Q3: Why is sCT worse than sCD on ImageNet-512?***
>
> ***A3***: We believe the higher training variance of CT is the main issue, particularly in the complex latent spaces defined by the pretrained encoder. We agree with the reviewer 2i3a’s hypothesis that the current encoder/decoder may not be optimal for consistency models. Theoretically, since the ground truth mapping in consistency models aims to transform a Gaussian distribution into a multimodal data distribution with potentially disconnected supports, its tangent can become ill-conditioned at boundary points, resulting in worse optimization dynamics. If we could develop a better encoder/decoder to create a more well-conditioned ground truth mapping in the latent space, the training of consistency models would likely become significantly easier. This is a fascinating research direction, and we plan to explore it in future work.
>
> ***Q4: Ablation study for Adaptive Double Normalization.***
>
> ***A4:*** Adaptive Double Normalization is crucial for the model to converge. Without it, the default AdaGN layers lead to divergence, forcing us to terminate the runs prematurely. As a result, we were unable to conduct ablation studies for this feature in the same manner as for others, as the model simply fails to train without it. To clarify this point, we have updated Lines 304-305 in the paper accordingly.
>
> ***Q5: Adaptive weighting achieves worse results in Figure 5(b).***
>
> ***A5:*** Thank you for pointing this out! The figure initially had incorrect legends—we mistakenly labeled the runs with adaptive weighting as "w/o adaptive weighting," leading to the confusion. We have since updated the paper to correct this. The revised Figure 5 now clearly shows that adaptive weighting improves upon fixed weighting functions for 2-step sampling and achieves comparable results for 1-step sampling.
>
> ***Q6: In Figure 5c, do discrete-time CMs have a constant number of time steps $N$ or a timestep schedule up to the maximum number of steps? If it is the former, it seems to be a bit unfair to discrete-time CMs.***
>
> ***A6:*** We used a constant number of time steps for discrete-time CMs for two reasons:
>
> 1. Results in Figure 5c focus on **consistency distillation**. Previous papers [1][2] employed timestep schedules only for consistency training, not for distillation.
>
> 2. Figure 5c is designed to analyze how performance changes as $N$ varies. To ensure a cleaner comparison, we chose to fix $N$ throughout the course of training, avoiding confounding effects introduced by varying $N$ during training.
>
> If the reviewer is curious about the performance of sCT versus discrete-time CT that uses timestep schedules, we have provided a comparison with the best-performing discrete-time CM (ECT) in our [summary of updates, part C](). Notably, our continuous-time CT consistently outperforms the best discrete-time counterparts.
>
>
> ***References:***
>
> [1] Song, Yang, Prafulla Dhariwal, Mark Chen, and Ilya Sutskever. "Consistency models." arXiv preprint arXiv:2303.01469 (2023).
>
> [2] Song, Yang, and Prafulla Dhariwal. "Improved techniques for training consistency models." arXiv preprint arXiv:2310.14189 (2023).

---

> > ### Author Response · Authors · 2024-11-22
> > **cont.**
> >
> > ***Q7: In Figure 7, is TTUR from DMD2 applied?***
> >
> > ***A7:*** No, we did not apply TTUR because we empirically found that incorporating TTUR exacerbates mode collapse issues in VSD. Specifically, while TTUR increases precision, it reduces recall, leading to worse final FIDs. We have tuned the multiplier for the training iterations of the variational network and determined that the optimal value is 1. We hypothesize that the training dynamics for ImageNet512 differ slightly from those of text-to-image diffusion models used in DMD2, which may explain this discrepancy.
> >
> > ***Q8: In Figure 7, is the generator of VSD also conditioned on the guidance scale?***
> >
> > ***A8:*** Yes, the generator of VSD is conditioned on the guidance scale. It shares the same architecture as sCD. The results in Figure 7 were obtained using the best guidance scales identified through a sweep at test time.
> >
> > ***Q9: Typo in line 266.***
> >
> > ***A9:*** Thank you for pointing this out. We’ve fixed it in our revision.

---

> > > ### Comment · Reviewer_pgsr · 2024-11-25
> > > **Thank you for your response!**
> > >
> > > The response has addressed my major concerns so I increase my rating from 6 to 8.

---

### Official Review · Reviewer_QC4D · 2024-11-03

**Soundness:** 3
**Presentation:** 3
**Contribution:** 4
**Rating:** 10
**Confidence:** 4

**Summary:**

This paper tackles the problem of  instability in continuous consistency models and presents many contributions.

The first novelty is the simplification of model parametrization in the EDM/CM frameworks. It is shown that rescaling the mean ($\alpha_t$ ) and noise ($\sigma_t$) schedule of conditional probability flows with the norm $||(\alpha_t, \sigma_t)||$, produces a second conditional flow, such that the EDM model parametrization and loss in the first are identical to those in the second. Furthermore, the ODE sampling procedure remains unchanged. However, the connection between the new flow schedule parameters and the  EDM scaling parameters is greatly simplified after normalization, facilitating further theoretical analysis, while maintaining the benefits of the EDM model definition (unit variance of input, target and minimizing the scaling of the output). The formulation with $\alpha_t=cos(t)$ and  $\sigma_t=sin(t)$ is named *TrigFlow*.

Then the training objective is studied and probed for causes of instability. Several factors are detected such as: an inappropriate $c_{noise}(t)$ which is fixed by being set to $t$; inappropriate Fourier scales in the time positional embeddings which are then reduced; the usage  'AdaGN layer', which is then replaced with 'Adaptive Double Normalization'; highly varying target norm alleviated by target normalization; inappropriate weighting mitigated by adaptive weighting and unstable terms solved by slowly introducing such terms into the loss with respect to the number of parameter updates.

In addition, methods for scaling such models to large sizes and datasets are proposed, namely JVP Rearrangement and JVP of Flash Attention.

Finally, the proposed models are compared against the state of the art methods, and results show that continuous consistency models are excellent generators, that only require 1 or 2 sampling steps to generate new quality data from learned distributions with continuous support.

**Strengths:**

The contributions of this paper are novel, clear, significant and positioned correctly.

Trigflow simplifies the theoretical analysis of flows by showing that  the mean ($\alpha_t$ ) and noise ($\sigma_t$) schedule in the definition of conditional flows can be normalized while preserving the model/loss formulation and the integrator-generated paths, while simplifying the connection between the scaling parameters of the model and those of the conditional flow. **(novelty, significance)**

Several components of continuous consistency models are studied and probed for potential causes of instability. They include: an inappropriate $c_{noise}(t)$ which is fixed by being set to $t$; inappropriate Fourier scales in the time positional embeddings which are then reduced; the usage  'AdaGN layer', which is then replaced with 'Adaptive Double Normalization'; highly varying target norm alleviated by target normalization; inappropriate weighting mitigated by adaptive weighting and unstable terms solved by slowly introducing such terms into the loss with respect to the number of parameter updates.  **(novelty, significance)**

Strategies for scaling such models to large sizes and datasets are proposed, namely JVP Rearrangement and JVP of Flash Attention. **(quality, significance)**

The proposed method shows competitiveness with the state of the art models, while only using 1 or 2 steps for generation, and outperforms all other tested methods that use 1 to 2 generation steps **(significance, quality)**

**Weaknesses:**

**While the contributions of the paper are numerous, the paper could be strengthened even further by:**

1) Comparing the proposed model with more recent flow models such as [1] and [2] and adding the results for rectified flows with 2 generation steps.

2) Placing the number of parameters and the number of parameter updates (or training time on identical hardware) for each model in Table 1 is very important as using an equal umber of parameters/compute is essential for ensuring a fair comparison.

**Some smaller issues and suggestions:**

a) The paper would be improved if an intuitive explanation is given for the loss in Equation 2. Also it could save readers some time if it is mentioned that the loss is derived in Song et al 2023, *Remark 10*.

b) The paper would be enriched by adding some generated images with one step.

c) Shouldn't $c_{skip}$ and $c_{out}$ be $\sigma_t$ and  $ \alpha_t \sigma_d$, that is for $\sigma_t=t$ and  $\alpha_t=1$: $c_{skip}=t$  and $c_{out}=\sigma_d$ in line 201/202?

d) In Equation (20) Appendix, shouldn't it be $\hat{D}$ for $D_{\theta}(x_t)=\hat{D}_{\theta}(\hat{x}_t(x_t))$?

e) The paragraph from line 924 to 938 (appendix) needs additional elaboration regarding the implications of having $||(\alpha_t, \sigma_t)||=1$ with respect to the invariance of the geometric set connecting $x_0$ and $z$.



**Typos:**

i) In line 126, a 2 is squared instead of the norm.

ii) in line 122, $z_t$ does not depend on time.


**References**


[1]  Tong et al. 2024. Improving and Generalizing Flow-Based Generative Models with Minibatch Optimal Transport

[2]  Kornilov et al 2024. Optimal Flow Matching: Learning Straight Trajectories in Just One Step

**Questions:**

How do the proposed models compare with [1} and [2]? Even tests on a small dataset using small models of equal size would be informative.. I completely understand however  if such comparisons cannot be made due to the short length of the discussion period.

Shouldn't $c_{skip}$ and $c_{out}$ be $\sigma_t$ and  $ \alpha_t \sigma_d$, that is for $\sigma_t=t$ and  $\alpha_t=1$: $c_{skip}=t$  and $c_{out}=\sigma_d$ in line 201/202?

Based on the experiments performed, are there any indications that the continuous consistency models will still face instability issues for even larger scales, in particular as compared to diffusion/flow models? This question mostly relates to this part in the paper: "Additionally,
we observe that consistency training is more effective at smaller scales but suffers from increased
variance at larger scales, while consistency distillation shows consistent performance across both
small and large scales."

[1]  Tong et al. 2024. Improving and Generalizing Flow-Based Generative Models with Minibatch Optimal Transport

[2]  Kornilov et al 2024. Optimal Flow Matching: Learning Straight Trajectories in Just One Step

---

> ### Author Response · Authors · 2024-11-22
> **Official response**
>
> Thank you for the detailed review and thoughtful feedback. Please find our responses to your questions below.
>
> ***Q1: Adding the results for more recent flow models and the 2-step rectified flow.***
>
> ***A1:*** Thank you for pointing out the reference. We have added the results from [1] to Table 1. However, since [2] does not include experimental results on CIFAR10 or ImageNet, it is difficult to make a direct comparison. Nevertheless, we have cited [2] in the updated version for completeness. In addition, the "2-rectified-flow" listed in Table 1 already refers to the 2-step distillation of rectified flow.
>
> ***Q2: An intuitive explanation for the loss in Eq(2), and mention a detailed citation (Remark 10) of it.***
>
>
> ***A2:*** Thanks for the suggestion. We’ve added the detailed citation in Line 165-166. The key to understanding Eq(2) lies in intuitively grasping the tangent $\frac{df}{dt}$, which represents the direction of instantaneous change that drives $f(t)$ toward $f(0)$. Optimizing $f$ along this direction, as described in Eq(2), ensures consistency with the boundary condition at $t=0$.
>
>
> ***Q3: Adding some generated images with one step.***
>
> ***A3:*** Thanks for the suggestion. We’ve added uncurated samples for both 1-step and 2-step sampling in Appendix H, generated by our sCD-XXL trained on ImageNet 512. All corresponding 1-step and 2-step samples use the same random seed.
>
> ***Q4: Additional elaboration regarding the implication of having unit norm.***
>
> ***A4:*** Thanks for the suggestion. We’ve added more discussions in the appendix accordingly.
>
> ***Q5: Typos of $c_{skip}$ and $c_{out}$; typo in Eq(20) in appendix; typo in line 126; typo in line 122.***
>
> ***A5:*** Thanks for pointing it out. We’ve fixed them in the revision.
>
> ***Q6: Are there any indications that the continuous consistency models will still face instability issues for even larger scales, in particular as compared to diffusion/flow models? This question mostly relates to the gap between sCT and sCD.***
>
> ***A6:*** In our experiments, we find that the training of sCM is stable at both small and large scales. We did not even apply any gradient clipping and the network activations are stable during the course of training. That said, we mostly focus on CNN-based architectures (with self-attention blocks), and leave the explorations of other architectures for future work.
>
> We believe the gap between sCT and sCD on ImageNet 512x512 is mainly caused by the higher training variance of CT, particularly in the complex latent spaces defined by the pretrained encoder. We agree with reviewer 2i3a’s hypothesis that the current encoder/decoder may not be optimal for consistency models. Theoretically, since the ground truth mapping in consistency models aims to transform a Gaussian distribution into a multimodal data distribution with potentially disconnected supports, its tangent can become ill-conditioned at boundary points, resulting in worse optimization dynamics. If we could develop a better encoder/decoder to create a more well-conditioned ground truth mapping in the latent space, the training of consistency models would likely become significantly easier. This is a promising research direction, and we plan to explore it in future work.
>
>
> ***References:***
>
> [1] Tong et al. 2024. Improving and Generalizing Flow-Based Generative Models with Minibatch Optimal Transport.
>
> [2] Kornilov et al 2024. Optimal Flow Matching: Learning Straight Trajectories in Just One Step.

---

> > ### Comment · Reviewer_QC4D · 2024-11-25
> >
> > Thank you for your response. Many of my main concerns have been addressed, and I believe this paper deserves to be highlighted at the conference.

---

### Official Review · Reviewer_DGp7 · 2024-11-03

**Soundness:** 3
**Presentation:** 4
**Contribution:** 3
**Rating:** 8
**Confidence:** 3

**Summary:**

The paper presents a unified perspective on diffusion-based and flow-based generative models and introduces a comprehensive set of techniques aimed at improving the training stability and overall performance of continuous-time consistency models for large-scale image generation. The techniques include: 1) enhancing time transformation and embeddings, 2) replacing the AdaGN layer with Adaptive Double Normalization, 3) normalizing the tangent function and applying tangent warm-up, 4) implementing an adaptive weighting function in the training objective, and 5) optimizing forward-mode differentiation. These techniques mitigate the numerical instability issues in continuous-time consistency models and enable the model to achieve highly competitive performance in class-conditioned image generation.

**Strengths:**

S1 - The paper provides a comprehensive analysis and set of solutions addressing the numerical instability issues in continuous-time consistency models, significantly improving performance and enabling the model to achieve competitive results on selected benchmarks.

S2 - Many of the enhancements are supported by detailed theoretical justification and experimental results.

S3 - The unified perspective on previous diffusion and flow-matching parameterizations is thorough, complete, and well-grounded, offering novel insights that could benefit the community.

S4 - The paper is well-structured and easy to follow.

**Weaknesses:**

W1 - Several design choices appear arbitrary and lack supporting evidence.

For example, in Section 4.1, the authors discuss the preference for Adaptive Double Normalization over AdaGN, but there is no experimental evidence supporting this choice. It would be more insightful to add a Figure similar to Figure 5 show experimental comparison between Adaptive Double Norm and AdaGN.

Similarly, in Section 4.2, the authors propose training with linear warm-up w.r.t the model's time derivative, yet no evidence is provided to demonstrate this choice’s effectiveness. Again, including an ablation study or comparative analysis showing the impact of the linear warm-up on training stability or performance metrics would provide more concrete evidence for the effectiveness of this specific choice

Furthermore, Figure 5(b) suggests that incorporating adaptive weighting in a two-step setting may lead to worse performance, while in the one-step setting, it only yields marginal improvement. Have the authors considered alternative designs for the two step case?

W2 - In Sections 4.1 and 5.2, the paper discusses the training compute of sCM. However, including a comparison of compute efficiency with other models (e.g., ECT [1]) would be more insightful, maybe a table or figure comparing the compute efficiency (e.g., FLOPs or training time) of sCM against ECT and other relevant baselines for a given performance level.

Additionally, given that the model is trained on a large-scale dataset (ImageNet 512) under latent setting, it would be beneficial to include discussions related to text-to-image generation.

[1] Geng, Zhengyang, Ashwini Pokle, William Luo, Justin Lin, and J. Zico Kolter. "Consistency Models Made Easy." arXiv preprint arXiv:2406.14548 (2024).

**Questions:**

Q1 - In Section 4.1, the authors emphasize the importance of time transformation in mitigating numerical instability. Have the authors considered other potential candidates for time transformation?

Q2 - In Figure 6(a) and Section 5.2, the authors mention that sCT is less effective at higher resolutions. Do the authors have any insights into why this might be?

Q3 - In Figure 6(b), it appears that the performance of sCD-XL under the two-step sCD setting is better than that of sCD-XXL, which contradicts the results reported in Table 2. Could the authors clarify the specific settings used in Figure 6(b)?

---

> ### Author Response · Authors · 2024-11-22
> **Thank you for the detailed review and thoughtful feedback!**
>
> Thank you for the detailed review and thoughtful feedback. Below we address specific questions.
>
> ***Q1: Experimental comparison between Adaptive Double Normalization and AdaGN.***
>
> ***A1:*** As observed in [1] and confirmed by our experiments, AdaGN leads to divergence when used for CM training. It results in FIDs ranging from 200 to 300, with highly unrealistic samples. Given this significant performance gap, it is impractical to include the results of AdaGN alongside Adaptive Double Normalization in a plot like Figure 5. We have updated Lines 304-305 to clarify this point in the revision.
>
>
> ***Q2: Ablation study for tangent linear warmup.***
>
> ***A2:*** Empirically, tangent warmup is optional. Removing it generally does not impact model performance, although gradients tend to have more spikes without it. While we do not consider tangent warmup a major technical contribution of our paper, we nonetheless included it in the main text for better reproducibility.
>
> ***Q3: Adaptive weighting performs worse for 2-step sampling in Figure 5b.***
>
> ***A3:*** Thank you for pointing this out! The figure initially had incorrect legends—we mistakenly labeled the runs with adaptive weighting as "w/o adaptive weighting," leading to the confusion. We have since updated the paper to correct this. The revised Figure 5(b) now clearly shows that adaptive weighting improves upon fixed weighting functions for 2-step sampling and achieves comparable results for 1-step sampling.
>
> ***Q4: Compute efficiency comparison with ECT.***
>
> ***A4:*** Please refer to our [sumary of updates, part C.](https://openreview.net/forum?id=LyJi5ugyJx&noteId=LDtCNvtiA5)
>
> ***Q5: Discussions related to text-to-image generation.***
>
> ***A5:*** We believe our method can be readily generalized to such models. Similar to our setup for ImageNet 512×512, these text-to-image diffusion models are also trained in the latent space, and are often designed for generating images of resolution around 512×512. We plan to explore this in future work.
>
>
> ***Q6: Other potential candidates for time transformation.***
>
> ***A6:*** We explored various other time transformations during our initial experiments, such as those defined by trigonometric functions. Empirically, their results were very similar to the identity transformation, which we decided to use for simplicity.
>
> ***Q7: Why is sCT less effective at higher resolutions?***
>
> ***A7:*** We believe the higher training variance of CT is the main issue, particularly in the complex latent spaces defined by the pretrained encoder. We agree with the reviewer 2i3a’s hypothesis that the current encoder/decoder may not be optimal for consistency models. Theoretically, since the ground truth mapping in consistency models aims to transform a Gaussian distribution into a multimodal data distribution with potentially disconnected supports, its tangent can become ill-conditioned at boundary points, resulting in worse optimization dynamics. If we could develop a better encoder/decoder to create a more well-conditioned ground truth mapping in the latent space, the training of consistency models would likely become significantly easier. This is a fascinating research direction, and we plan to explore it in future work.
>
>
> ***Q8: Confusion about sCD-XXL vs. sCD-XL in Fig 6(b).***
>
> ***A8:*** To clarify, sCD-XXL outperforms sCD-XL under the two-step sCD setting. In Fig 6(b), we show the relative FID ratio (FID_CM / FID_DM), where FID_CM is the consistency model's FID, and FID_DM is the diffusion model's FID. What the reviewer observed is that sCD-XXL has a larger **FID ratio** than sCD-XL, this is because FID_DM of XXL is much smaller. Therefore, the actual FID of sCD-XXL (ratio × FID_DM) is still lower than that of sCD-XL.
>
> ***References:***
>
> [1] Song, Yang, and Prafulla Dhariwal. "Improved techniques for training consistency models." arXiv preprint arXiv:2310.14189 (2023).

---

> > ### Comment · Reviewer_DGp7 · 2024-11-25
> >
> > Thank you for your thorough response. Many of my concerns have been addressed, and I'm willing to increase my rating.

---

### Official Review · Reviewer_Fdt4 · 2024-11-03

**Soundness:** 4
**Presentation:** 4
**Contribution:** 4
**Rating:** 10
**Confidence:** 4

**Summary:**

The authors propose improvements to the consistency models generative paradigm and named their new method sCM. Specifically they -vastly- improve the FID for consistency models with the introduction of several new ideas to both stabilize and simplify continuous consistency models.

My understanding is the main claim of simplification for sCM comes from the simplification of EDM (Kerras et al.) normalizing design, resulting in $c_{in}=c_{skip}=1$ which in turns simplifies the continuous expression of consistency models. Another simplification is the combination of both EDM and Flow Matching concepts into their method which they call TrigFlow. Yet another simplification, not claimed as such by authors, is the use of vanilla L2 loss compared to Huber/LPIPS use in previous iterations of consistency models. This last simplification has the additional benefit to be more probabilistically grounded.

There are 3 main proposed ideas to stabilize the training of consistency models:
1. Identity-time transformation as a replacement to the log-transformation from EDM
2. Fourier embedding of the time dimension are replaced by positional embeddings
3. AdaGN is modified to also normalize the conditioning inputs for scale and bias.
More ideas are also proposed in the training objective to stabilize training, namely: tangent normalization and tangent warmup. It is my understanding that the adaptative weighing is the same as in EDM.

The paper also provides ample ablations to demonstrate the effects and the reasoning motivating these 3 proposed improvements.

**Strengths:**

The paper is very well grounded mathematically and experimentally. The analysis is based on understanding the causes of training instabilities by decomposing the loss, validating each component experimentally and proposing changes to solve the root causes.
The mathematics while greatly simplified are still pretty complex and the paper shines in its clarity to make the logical reasoning easy to follow.
The experimental results are also outstanding resulting in very significant gains, essentially taking consistency models within 10% of the SOTA for diffusion models.

**Weaknesses:**

1. The limitations of the method are not very clear to me besides the fact that it's still 10% worse than SOTA for diffusion.
2. The section on positional embeddings (line 269 and on) lacks details to be fully understandable without having to read another paper. Maybe beefing up that section would make the paper more self-contained.
3. I found all figure very useful with the exception of Figure (3) which I felt did not add much value.

I also noticed a typo (definitely not affecting my score), just leaving it there for authors to fix their manuscript:
- Line 362: "cause instability" => "causes instability"

**Questions:**

These are mostly from the weakness section:
1. Are there more limitations other than the 10% worse than diffusion SOTA?
2. Is the method really fully stable?

---

> ### Author Response · Authors · 2024-11-22
> **Official response**
>
> Thank you for the detailed review and thoughtful feedback. Please find our responses to your questions below.
>
>
> ***Q1: More Limitations.***
>
> ***A1***: Thanks for the suggestion. Due to the page limit, we’ve included an additional section in the appendix for further discussions on limitations.
>
> ***Q2: Elaborate positional embedding.***
>
> ***A2***: The positional embedding for timesteps is a standard layer initially used in DDPM [1] and widely adopted in diffusion models. We’ve added further clarifications around Line 259-261.
>
>
> ***Q3: Fig 3 does not add much value.***
>
> ***A3***: Figure 3 highlights the key difference between discrete-time and continuous-time training, specifically that continuous-time training avoids discretization errors. We are happy to revise the illustration based on reviewer feedback.
>
> ***Q4: Typos.***
>
> ***A4***: Thanks for pointing it out! We’ve fixed it in the updated paper.
>
> ***Q5: Is the method really fully stable?***
>
> ***A5***: We believe so. Combining all the proposed techniques together, we found that the training of sCM is stable at both small and large scales, requiring no gradient clipping, and exhibiting smooth variation of network activations during the course of optimization. That said, we have only examined CNN architectures (with self-attention blocks) for CMs so far. We leave the exploration of other architectures, such as DiTs and SSMs, for future work.
>
> ***References:***
>
> [1] Ho, Jonathan, Ajay Jain, and Pieter Abbeel. "Denoising diffusion probabilistic models." Advances in neural information processing systems 33 (2020): 6840-6851.

---

### Official Review · Reviewer_2i3a · 2024-11-04

**Soundness:** 4
**Presentation:** 3
**Contribution:** 4
**Rating:** 10
**Confidence:** 5

**Summary:**

This paper investigates a fundamental topic in consistency models (CMs), specifically the challenges of discretization errors and the resulting training stability issue. Consistency Models can be trained in discrete or continuous time, either from scratch using a dataset or distilled from pretrained teacher scores. CMs' theoretical foundation elucidates the importance of controlling the discretization error and eventually achieving consistency in continuous time. While continuous-time CMs eliminate the discretization errors present in their discrete-time counterparts, they suffer from training instability, a problem that is not yet well understood in the research community.

This work conducts a comprehensive study into continuous-time CMs, covering forward process parameterization, network architecture, and training techniques. The authors first develop a simplified diffusion process formulation called TrigFlow, which unifies EDM and Flow Matching for the first time. Building upon this foundation, they analyze the gradient flow of continuous-time CMs, identify the root cause of training instability, and mitigate this issue through modifications to time embeddings and adaptive group normalization. Additional training techniques, such as adaptive weighting functions and annealing, further contribute to improved training stability and scalability.

The resulting method, sCT/sCD, allows continuous-time CMs to be trained at an unprecedented scale, scaling up to 1.5B parameters on ImageNet 512x512. These results significantly narrow the performance gap between CMs and state-of-the-art diffusion models to less than 10% in FID, while matching or even surpassing adversarial methods and discrete/continuous autoregressive models in both performance and efficiency.

**Strengths:**

This is a very strong paper in analysis, practical techniques, writing, and experiment results.

- Novelty. This paper's novelty is evident in several aspects.
    - First, it studies an important but less studied problem: consistency models in continuous time, together with the training stability and discretization error of consistency models.
    - The proposed TrigFlow, as a novel unification of EDM and Flow Matching, substantially simplifies the analysis presented later and the practical techniques.
    - The gradient analysis of continuous-time objective reveals the root cause of instability. To the best of my knowledge, this is the first paper to establish the gradient analysis for CMs.
    - Model architecture modifications are original since existing works are mostly inherited from Diffusion Models' design and focus on the training techniques and formulations, leaving the architectural design underexplored.

- Soundness. Its technical claims are well backed up by both theoretical analysis and empirical results. I particularly appreciate the in-depth investigation into the training dynamics and gradient analysis of continuous-time CMs.

- Presentation. The logical flow of this paper is well structured and smooth. The problem statement is clearly defined, and the explanation of why discretization errors matter for CMs and the motivation toward continuous-time formulation is crystal clear. The gradient analysis into continuous-time CMs is thoughtfully motivated and carefully organized. Even the appendix is well-written, offering useful insights into the proposed techniques.

- Experiments. Proposed techniques allow for training continuous-time Consistency Models (sCMs) at an unprecedented scale. Experiment results are impressive, matching/outperforming adversarial approaches, score distillation, and recent autoregressive models.
    - Gradient variances have been carefully controlled via adaptive weighting and normalization techniques.
    - Comprehensively studying the scaling behaviors of sCMs under continuous-time training.
    - Comparisons with improved score distillation baseline using many methods developed in this work confirm the mode coverage of CMs.
    - Additionally, the paper discusses efficient and stable implementation strategies for continuous-time CMs.

Given the potential impact of this paper, I strongly recommend acceptance with conference highlights. It was a great pleasure to read through the manuscript!

**Weaknesses:**

I did not find any apparent weaknesses in the analysis or experiments (including both ablation studies and performance evaluation). There are research questions worth further investigation, as discussed below.

**Questions:**

1. I appreciate the scaling study from ImageNet 64x64 to 512x512, where the former operates directly in the image space, while the latter relies on additional image compression models. The difference between sCT and sCD at the 512x512 resolution is intriguing, as even increasing model size and batch size cannot easily close the gap, while at the 64x64 resolution, scaling is sufficient to compensate for the variance induced by Monte Carlo estimation.
    - Data complexity at 512x512 and the lack of effective mode decomposition definitely contribute to this discrepancy. However, I am curious about the authors' thoughts on the extent to which the increased variance could be caused by the pretrained image encoder/decoder. In other words, could data modes become *more dispersed* in the latent space, making it harder for sCT to learn effectively? Would it be better to train CMs directly in the image space (with some special architecture design and weighting schemes) or to find a latent space that is more suitable for CMs? In some sense, modern autoregressive models focus on tokenizer design for visual generation. Could latent space compression for CMs require properties distinct from those used in DMs? While this is more hypothetical, I would be happy to hear more thoughts.

2. If I understand correctly, there are three weighting functions applied, namely learnable adaptive weighting, tangent normalization (per-sample basis weighting), a prior weighting $w(t) = \frac{1}{\sigma_d \tan(t)} = \frac{1}{\sigma_t}$. To what extent does this prior weighting contribute to variance reduction? Is it helpful to stabilize the learnable adaptive weighting layer? I assume the time embeddings of learnable adaptive weighting could be either positional embedding or Fourier embedding since it is not directly involved in $\frac{\mathrm{d} \boldsymbol{f}}{\mathrm{d}t}$.

As an additional comment, could the authors consider conducting distillation experiments using the data-free formulation in [1]? I am curious how continuous-time CMs would scale and how stable they would be without using an extra dataset. No hurry to complete these experiments during the limited rebuttal period!

[1] Consistency Models Made Easy

### Minor

1. In Line 213 and in Line 227, both diffusion models and consistency models are denoted as $f_\theta(\mathbf{x}_t, t)$ but with different equations.

2. The Adaptive Double Normalization is less explained. Is it the same as local response normalization applied to the modulation layer?

---

> ### Author Response · Authors · 2024-11-22
> **Official response**
>
> Thank you for the detailed review and thoughtful feedback. Please find our responses to your questions below.
>
> ***Q1: Why is sCT less effective than sCD on ImageNet 512?***
>
> ***A1***: We believe the higher training variance of CT is the main issue, particularly in the complex latent spaces defined by the pretrained encoder. We agree with the reviewer’s hypothesis that the current encoder/decoder may not be optimal for consistency models. Theoretically, since the ground truth mapping in consistency models aims to transform a Gaussian distribution into a multimodal data distribution with potentially disconnected supports, its tangent can become ill-conditioned at boundary points, resulting in worse optimization dynamics. If we could develop a better encoder/decoder to create a more well-conditioned ground truth mapping in the latent space, the training of consistency models would likely become significantly easier. This is a fascinating research direction, and we plan to explore it in future work.
>
> ***Q2: To what extent does this prior weighting contribute to variance reduction? Is it helpful to stabilize the learnable adaptive weighting layer?***
>
> ***A2***: We can express the relationship between prior weighting and tangent normalization as:$\frac{w_t \cdot \|\frac{df}{dt}\|}{w_t \cdot \|\frac{df}{dt}\| + c} = \frac{\|\frac{df}{dt}\|}{\|\frac{df}{dt}\| + \frac{c}{w_t}}$,
> where $w_t$ is the prior weighting and $c$ is the tangent normalization constant. This shows that the prior weighting acts as a time-dependent tangent normalization with $c'_t = \frac{c}{w_t}$, effectively balancing the variance of the tangent across $x_t$ and $t$.
>
> Empirically, we observe that $\|\frac{df}{dt}\|$ becomes quite large as $t \to T$ (i.e., the pure noise distribution). Prior weighting helps reduce variance and improves the numerics of tangent computation at these time steps. Prior weighting also provide slight benefits for the training stability of learnable adaptive weighting, but it is not strictly necessary.
>
>
> ***Q3: Data-free distillation of continuous-time CMs.***
>
> ***A3***: Thanks for the insightful suggestion! This is an interesting research direction and we plan to explore it in future work.
>
> ***Q4: Conflicting notations for both diffusion models and consistency models.***
>
> ***A4***: Thanks for pointing it out! We have changed the notation of diffusion models to $f^{\text{DM}}$.
>
> ***Q5: More explanations about the Adaptive double normalization.***
>
> ***A5***: Pixel normalization is quite similar to local response normalization. Since our network is a convolutional network, the shapes of the modulation layers $s(t)$ and $b(t)$ match the hidden feature $x$ (i.e., $C \times H \times W$). We normalize the $C$ values at each pixel position, effectively performing RMS normalization for each pixel.

---

> > ### Comment · Reviewer_2i3a · 2024-12-03
> >
> > Thank you for your thoughtful responses to my questions. I am pleased to see the further improvements made to the paper and have updated my score to 10.

---

### Author Response · Authors · 2024-11-22
**A summary of updates**

We would like to thank all reviewers for providing high quality reviews and constructive feedback that have improved the paper. We are encouraged that reviewers acknowledged our novelty and contributions, including “the first gradient analysis for CMs” (2i3a, QC4D), "Trigflow, a novel unification of EDM and Flow Matching" (2i3a, QC4D, pgsr), and "several new ideas to stabilize and simplify continuous consistency models" (Fdt4). Our analysis and methods were recognized as “well-backed by theory and experiments” (2i3a), “comprehensive” (DGp7), and offering “significant insights” (pgsr). We are happy that reviewers found our experimental results “impressive, matching/outperforming adversarial approaches” (2i3a), with “ample ablations” and “significant gains” (Fdt4), demonstrating scalability and competitiveness with state-of-the-art models (QC4D, pgsr), and the paper was deemed “very well-written, clear, and easy to follow” (2i3a, Fdt4, DGp7, QC4D, pgsr).


We have updated our draft to further improve the writing and incorporate suggestions from reviewers, extended the appendix with more details. Below, we summarize changes made in the updated submission.

## A. Improved writing and fixed typos.

We have fixed all the typos pointed out by the reviewers and added more explanations of several techinical parts, including positional embedding, adaptive double normalization, trigflow formulation and tangent warmup.

## B. More discussions.

We have updated the appendix to add a new section for further discussions and limitations. We cover the potential reason for why sCT is less effective than sCD on ImageNet 512x512, the computation costs of sCM, and the limitations of our proposed method.

## C. Comparison with ECT.

We compare the 1-step FID scores at different training iterations between ECT and sCT on CIFAR-10. As shown below, our sCT significantly outperforms ECT throughout training, demonstrating superior compute efficiency and faster convergence.

For a fair comparison, we use the same network architecture as ECT on CIFAR-10, specifically the DDPM++ network proposed by [1], which does not include the AdaGN layer. We also adopt the same dropout rate of 0.20 and batch size of 128 as ECT, differing from our default setting of 512 reported in Table 1. For sCT, we set $P_{mean}=-1.0$ and $P_{std}=1.8$, using the TrigFlow parameterization (with $c_{noise}=t$). All other hyperparameters match those used in the experiments reported in Table 1.


| Iters    | 100k     | 200k     | 400k |
| -------- | -------- | -------- | -----|
| ECT      |   4.54 | 3.86 |3.60 |
| sCT     | **3.97**| **3.51** | **3.09**     |


Additionally, since JVP computation, also known as “forward differentiation,” does not require saving intermediate activations, its GPU memory cost is nearly identical to that of a single forward pass. The effective FLOPS are roughly equivalent to twice the forward computation for most network layers. Consequently, each iteration of our continuous-time CM has a comparable computational cost to previous discrete-time CMs, as the latter also require two forward passes per iteration.

## D. Clarifying the ablation study of adaptive weighting in Fig 5(b).

We thank reviewers DGp7 (R3) and pgsr (R5) for identifying this error. The figure initially contained incorrect legends, where we mistakenly labeled the runs with adaptive weighting as "w/o adaptive weighting," causing the confusion. We have since corrected this in the paper. The updated Figure 5(b) clearly demonstrates that adaptive weighting outperforms fixed weighting functions for 2-step sampling and achieves comparable results for 1-step sampling.

## E. Adding uncurated 1-step samples (using the same random seed as 2-step samples).

We have added uncurated 1-step samples in Appendix H, generated by our sCD-XXL model trained on ImageNet 512. For a clear comparison, the 1-step and 2-step samples are generated using the same random seed. As illustrated in the figures, our 1-step sCD can produce high-quality samples.

***References:***

[1] Song, Yang, et al. "Score-based generative modeling through stochastic differential equations." arXiv preprint arXiv:2011.13456 (2020).

[2] Karras, Tero, et al. "Elucidating the design space of diffusion-based generative models." Advances in neural information processing systems 35 (2022): 26565-26577.

---

### Meta-Review · Area_Chair_2mto · 2024-12-22

**Metareview:**

Consistency models (CMs) are specializations of diffusion models for the purpose of faster sampling. While existing CMs are faster, they are difficult to train, and the quality lags behind other diffusion models. This work provides a close study and several improvements that theoretically re-connect CM and other classes of diffusion, simplify this theory, stabilize training as a result of the theoretical analysis, and then scale up the model, training, and data. The result is a CM that narrows the gap with the state-of-the-art for diffusion on ImageNet 512x512. All told, this work is strong on multiple fronts with theoretical, empirical, and engineering contributions that significantly improve the state-of-the-art for CMs.

This a clear accept with all ratings >= 8 and 3/5 reviewers giving the highest rating. The meta-reviewer agrees because this submission delivers on theoretical and empirical contributions while improving results on a popular and significant topic that is unarguably well-received by the expert reviewers with backgrounds on diffusion modeling, consistency models, and large-scale generative modeling.

**Additional Comments On Reviewer Discussion:**

The authors provided a general summary and review-wise responses. 4/5 reviewers engaged in the discussion and maintained or raised their positive ratings. The points raised were largely about clarification or ablation, and not about correctness, and have been satisfactorily resolved by the rebuttal and the summarized updates that can be incorporated into the revision and appendix.

---

### Decision · Program_Chairs · 2025-01-22

Accept (Oral)